# A Review of the System-Intrinsic Nonequilibrium Thermodynamics in Extended Space (MNEQT) with Applications

**DOI:** 10.3390/e23121584

**Published:** 2021-11-26

**Authors:** Purushottam D. Gujrati

**Affiliations:** 1Department of Physics, The University of Akron, Akron, OH 44325, USA; pdg@uakron.edu; 2Department of Polymer Science, The University of Akron, Akron, OH 44325, USA

**Keywords:** unique-nonunique macrostate, system-intrinsic and medium-intrinsic properties, internal equilibrium, extended state space, entropy with and without memory, entropy generation, global temperature, generalized macroheat and macrowork, steady state, microstate probabilities, Brownian motion

## Abstract

The review deals with a *novel approach* (MNEQT) to nonequilibrium thermodynamics (NEQT) that is based on the concept of internal equilibrium (IEQ) in an enlarged state space $S_{Z}$ involving *internal variables as additional state variables*. The IEQ macrostates are unique in $S_{Z}$ and have no memory just as EQ macrostates are in the EQ state space $S_{X}\subset S_{Z}$. The approach provides a clear strategy to identify the internal variables for any model through several examples. The MNEQT deals directly with system-intrinsic quantities, which are very useful as they fully describe irreversibility. Because of this, MNEQT solves a long-standing problem in NEQT of identifying a unique global temperature *T* of a system, *thus fulfilling Planck’s dream of a global temperature for any system*, even if it is not uniform such as when it is driven between two heat baths; *T* has the conventional interpretation of satisfying the Clausius statement that the *exchange macroheat*$d_{e}Q$*flows from hot to cold*, and other sensible criteria expected of a temperature. The concept of the generalized macroheat $dQ=d_{e}Q+d_{i}Q$ converts the Clausius inequality $dS\geq d_{e}Q/T_{0}$ for a system in a medium at temperature $T_{0}$ into the *Clausius equality*dS≡dQ/T, which also covers macrostates with memory, and follows from the extensivity property. The equality also holds for a NEQ isolated system. The novel approach is extremely useful as it also works when no internal state variables are used to study nonunique macrostates in the EQ state space $S_{X}$ at the expense of explicit time dependence in the entropy that gives rise to memory effects. To show the usefulness of the novel approach, we give several examples such as irreversible Carnot cycle, friction and Brownian motion, the free expansion, etc.

## 1. Introduction

Thermodynamics of a system out of equilibrium (EQ) [1,2,3,4,5,6,7,8] is far from a complete science in contrast to the EQ thermodynamics based on the original ideas of Carnot, Clapeyron, Clausius, Thomson, Maxwell, and many others [9,10,11,12,13,14,15,16] that has by now been firmly established in physics, thanks to Boltzmann [17,18] and Gibbs [19]. Therefore, it should not be a surprise that there are currently many schools of nonequilibrium (NEQ) thermodynamics (NEQT), among which are the most widely known schools of local-EQ thermodynamics, rational thermodynamics, extended thermodynamics, and GENERIC thermodynamics [20,21]. This pedagogical review and various applications in different contexts deal with a recently developed NEQT, which we have termed MNEQT, with M referring to a macroscopic treatment in terms of *system-intrinsic* (SI) quantities of the system Σ at each instant. These quantities are normally taken to be *extensive* SI-quantities, and are used as state variables to describe a macrostate M of Σ. The MNEQT has met with success as we will describe in this review so it is desirable to introduce it to a wider class of readers and supplement it with many nontrivial applications.

We take Σ as a *discrete* system in that it is separated from its surrounding medium $\tilde{\Sigma }$ (if it exists) with which it interacts; see Figure 1. Such a system is also called a Schottky system [21,22,23] Because of the use of SI-quantities, the MNEQT differs from all other existing approaches to the NEQT in that the latter invariably deal with exchange quantities with $\tilde{\Sigma }$, which are *medium-intensive*(MI) quantities that differ from SI-quantities in important ways in a NEQ process as we will see. We will use $\overset{{}^{\circ}}{M}$NEQT to refer to the latter approaches, with $\overset{{}^{\circ}}{M}$ referring to the use of macroscopic exchange quantities. The corresponding NEQ statistical mechanics of the MNEQT is termed μNEQT, in which μ refers to the treatment of Σ in terms of microstates, which form a countable set $\left\{m_{k}\right\}$, with *k* counting various microstates. The existence of the μNEQT is possible only because of the use of SI-quantities in the MNEQT. These quantities are easily associated with $\left\{m_{k}\right\}$ as will become clear here. This ability in the MNEQT immediately distinguishes it from the $\overset{{}^{\circ}}{M}$NEQT as the latter cannot lead directly to a statistical mechanical treatment with $\left\{m_{k}\right\}$. Therefore, we believe that the MNEQT and μNEQT will prove very useful. All quantities pertaining to M are called *macroquantities*, while those pertaining to microstates contain an index *k* and are called *microquantities* for simplicity in this review.

While most of the review deals with an isolated system or an interacting system in a medium, we will occasionally also consider a system interacting with two different media such as in Figure 2, to study driven and steady macrostates [24,25,26,27] at $\tau ∼\tau_{\mathrm{st}\;}$for which there is no EQ macrostate having unique values of the temperature, pressure, etc. as long as we do not allow the media to come to EQ with each other, which takes much longer time $\tau_{\mathrm{EQ}\;}>>\tau_{\mathrm{st}\;}$. A steady or an unsteady macrostate always gives rise to irreversible entropy generation so it truly belongs to the realm of the NEQT. What makes the MNEQT a highly desirable approach is that it can also deal with unsteady processes easily as we will do.

### 1.1. Unique Macrostates in Extended State Space

The firm foundation of EQ statistical mechanics is accomplished by using the concept of microstates $m_{k}$ of Σ and their EQ probabilities $p_{k\mathrm{eq}}$. This is feasible as the EQ macrostate $M_{\mathrm{eq}}$ is *unique* in the EQ state space $S_{X}$ spanned by the set of observables (see Definition 1) X=(E,V,⋯), where E,V,⋯ are the energy, volume, etc. using standard notation (we do not show the number of particles *N* as we keep it fixed throughout this review; see later, however). But the same cannot be said about its extension to describe NEQT, since NEQ macrostates M in $S_{X}$ are not unique [12] even if they appear in a process between two EQ macrostates, which we will always denote by $\bar{P}$ and use P for any general process including $\bar{P}$. It is clear that unless we can specify the microstates for M uniquely, we cannot speak of their probabilities $p_{k}$ in a sensible way, but this is precisely what we need to establish a rigorous *NEQ statistical mechanics* of thermodynamic processes [28,29,30,31,32,33,34,35,36,37,38]. The system is usually surrounded by an external medium $\tilde{\Sigma }$, which we always take to be in EQ; see Figure 1b. The combination $\Sigma_{0}$ as the union $\Sigma \cup \tilde{\Sigma }$ forms an isolated system, which we assume to be stationary.

The lack of uniqueness of M is handled in the MNEQT by using a well-established practice [3,16,39,40,41,42,43,44,45] by considering a properly extended state space $S_{Z}$ spanned by Z≐X∪ξ, by including a set ξ of *internal variables*, in which NEQ macrostates M and microstates of interest can be uniquely specified during the entire process P. Here, ξ is internally generated within Σ so it *cannot* be controlled by the observer. The use of internal variables in glasses, prime examples of NEQ systems, is well known, where they give rise to distinct relaxations of the glassy macrostate [42,43,46,47,48,49]. Their justification is based on the ideas of chemical reactions [50], and has been formalized recently by us [45] to any NEQ macrostate M. It is well known that internal variables contribute to irreversibility in P, which justifies their important role in the NEQT. We give several examples for their need later in the review and a clear strategy to identify them for computation under different conditions. In $S_{Z}$, the *unique* M’s are specified by the collection $\left\{m_{k},p_{k}\right\}$ of two *independent* quantities, which form a probability space P. We can then pursue any P followed by M(t) as the latter evolves in time *t* to another (EQ or NEQ) unique macrostate. A major simplification occurs when this independence is maintained at each instant so that during the evolution, each microstate $m_{k}$ follows a trajectory (such as a Brownian trajectory) $\gamma_{k}$ whose characteristics do not depend on $p_{k}\left(t\right)$ as a function of time *t* ([32] (for example)); the latter, of course, determines the trajectory probability $p_{\gamma_{k}}$. Thus, $\left\{\gamma_{k},p_{\gamma_{k}}\right\}$ uniquely specifies P in P. For the same collection $\gamma ≐\left\{\gamma_{k}\right\}$, different choices of $\left\{p_{\gamma_{k}}\right\}$ describe different processes.

### 1.2. Layout

The review is divided into two distinct parts. The first part consisting of Section 3, Section 4, Section 5, Section 6, Section 7, Section 8 and Section 9 deals with the up-to-date foundation of the MNEQT for Σ, regardless of whether it is isolated or interacting (in the presence of one or more a external sources). We have tried to make the new concepts and their physics as clear as possible so a reader can appreciate the foundation of the MNEQT, which can be complex at times. The most important one is that of the NEQ temperature *T* as anticipated by Planck that is required to be defined globally over the system so that it can satisfy the Clausius statement about macroheat flow from hot to cold. The concepts of the *generalized macroheat* dQ and the *generalized macrowork* dW are directly and uniquely defined in terms of SI-quantities that pertain to the system alone. Thus, they are capable of describing the irreversibilty in the system. A clear strategy to identify internal variables is discussed for carrying out thermodynamic computation. The other part consisting of Section 10, Section 11, Section 12, Section 13 and Section 14 deals with various applications of the MNEQT, many of which cannot be studied within the $\overset{{}^{\circ}}{M}$NEQT without imposing additional requirements. This part provides an abundant evidence of successful implementation of the MNEQT.

The layout of the paper is as follows. In the next section, we introduce our notation and give some useful definitions and new concepts without any explanation. This section is only for bookkeeping so that readers can come back to it to refresh the concepts in the manuscript later when they are not sure of their meanings. The next six sections deal with various new concepts and theory behind the MNEQT. Section 3 introduces the central concept of internal variables that are required for arbitrary NEQ macrostates M. Many examples are given to highlight their importance for M. They form the extended state space $S_{Z}$, which contains the state space $S_{X}$ as a proper subspace. The internal variables are irrelevant for EQ macrostates in $S_{X}$. Section 4 is also very important, where we introduce the concept of NEQ entropies based on the original ideas of Boltzmann. In this sense, the derivation of this entropy is thermodynamic in nature, and gives rise to an expression of *S* that generalizes the Gibbs formulation of the entropy to NEQ macrostates. Using this formulation, we reformulate a previously given proof of the second law. In Section 5, we formulate the statistical mechanics of the MNEQT, and discuss the statistical significance of dW and dQ that provide a reformulation of the first law in terms of SI-quantities for any *arbitrary process* between any two arbitrary macrostates. The SI-quantities are determined by Σ alone, even if it is interacting with its exterior, and its usage has neither been noted nor has been appreciated by other workers in the field. These generalized macroquantities are different from exchange macrowork and macroheat. In this reformulation, the first law includes the second law in that it contains all the information of the irreversibility encoded in M. This formulation applies equally well to the exchange energy change $d_{e}E$ and the internally generated energy change $d_{i}E$, which shows the usefulness of the formulation. In Section 6, which is the most important section for the foundation of the MNEQT, we discuss the conditions for M to be uniquely specified in $S_{Z}$, and introduce the concept of the internal equilibrium (IEQ) to specify $M_{\mathrm{ieq}}$ in $S_{Z}$. A parallel is drawn between $M_{\mathrm{ieq}}$ and $M_{\mathrm{eq}}$ so that many results valid for $M_{\mathrm{eq}}$ also apply to $M_{\mathrm{ieq}}$, except that the latter has nonzero entropy generation ($d_{i}S\geq 0$). The entropy of $M_{\mathrm{ieq}}$ is a state function in $S_{Z}$, while that of a macrostate $M_{\mathrm{nieq}}$ that lies outside of $S_{Z}$ is not a state function. see later. The entropy of M that lies outside $S_{X}$ is similarly not a state function of X. We show that the NEQ entropy in Section 4 reduces to the thermodynamic EQ entropy for $M_{\mathrm{eq}}$ and to the thermodynamic IEQ entropy for $M_{\mathrm{ieq}}$. We introduce the concept of a NEQ thermodynamic temperature *T* as an inverse entropy derivative (∂S/∂E). We show that this concept satisfies various sensible requirements (C1–C4) of a thermodynamic temperature, which is global over the entire system even if it is inhomogeneous. This, we believe, solves a long-standing problem of a NEQ temperature. In terms of *T*, we show that the Clausius inequality in the $\overset{{}^{\circ}}{M}$NEQT is turned into an equality in the MNEQT as shown in Section 7. In Section 9, which is the last section of the first part, we use the idea of chemical equilibrium to show how entropy is generated in an isolated system. We now turn to the second part of the review. In Section 10, we consider various applications of the MNEQT ranging from a simple system to composite systems under various conditions. This section is very important in that we establish here that we can treat a system either (i) as a "black box" $\Sigma_{B}$ of temperature *T* but without knowing anything about its interior, or (ii) as a composite system $\Sigma_{C}$ for which we have a detailed information about its interior inhomogeneity. Both realizations give the same irreversible entropy generation. Thus, we can always treat a system as $\Sigma_{B}$ of temperature *T*, whose study then becomes simpler. In Section 11, we apply our approach to a glassy system and derive the famous Tool-Narayanaswamy equation for the glassy temperature *T*. In Section 12, we apply the MNEQT to study an irreversible Carnot cycle and determine its efficiency in terms of $\Delta_{i}S$. In Section 13, we apply the MNEQT to a very important problem of friction and the Brownian motion. In Section 14, we consider a classical and a quantum expansion. In the classical case, we study the expansion in $S_{X}$, where M is a non-IEQ macrostate, with an explicit time-dependence, and in $S_{Z}$, where M is a an IEQ macrostate, with no explicit time-dependence, and show that we obtain the same result. The quantum expansion is only studied in $S_{Z}$. The last section provides an extensive discussion of the MNEQT and draws some useful conclusions.

## 2. Notation, Definitions and New Concepts

### 2.1. Notation

Before proceeding further, it is useful to introduce in this section our notation to describe various systems and their behavior and new concepts for their understanding without much or any explanation (that will be offered later in the review where we discuss them) so that a reader can always come back here to be reminded of their meaning in case of confusion. In this sense, this section plays an important role in the review for the purpose of bookkeeping.

Even though Σ is macroscopic in size, it is extremely small compared to the medium $\tilde{\Sigma }$; see Figure 1b. The medium $\tilde{\Sigma }$ consists of two parts: A work source ${\tilde{\Sigma }}_{w}$ and a macroheat source ${\tilde{\Sigma }}_{h}$, both of which can interact with the system Σ directly but not with each other. This separation allows us to study macrowork and macroheat exchanges separately. We will continue to use $\tilde{\Sigma }$ to refer to both of them together. The collection $\Sigma_{0}=\Sigma \cup \tilde{\Sigma }$ forms an isolated system, which we assume to be stationary. The system in Figure 1a is an isolated system, which we may not divide into a medium and a system. Each medium in Figure 2, although not interacting with each other, has a similar relationship with Σ, except that the collection $\Sigma_{0}=\Sigma \cup {\tilde{\Sigma }}_{1}\cup {\tilde{\Sigma }}_{2}$ forms an isolated system. In case they were mutually interacting, they can be treated as a single medium. In the following, we will mostly focus on Figure 1 to introduce the notation, which can be easily extended to Figure 2.

We will use the term "body" to refer to any of $\Sigma ,\tilde{\Sigma }$, and $\Sigma_{0}$ in this review and use $\Sigma_{b}$ to denote it. However, to avoid notational complication, we will use the notation suitable for Σ for $\Sigma_{b}$ if no confusion would arise in the context. As the mechanical aspect of a body is described by the Hamiltonian H, whose value determines its macroenergy *E*, it plays an important role in thermodynamics. Therefore, it is convenient to introduce
(1)w≐X\E=(V,⋯),W≐Z\E=(V,⋯,ξ),
where \E means to delete *E* from the set, and ⋯ refers to the rest of the elements in X besides *V*. We use x to denote the collection of coordinates and momenta of the *N* particles in the phase space of Σ. The variable W appears as a parameter set in the Hamiltonian H(xW) of Σ that can be varied in a process with a concomitant change in H. As internal variables play no role in EQ, W=w in Equation We will normally employ a discretization of the phase space in which we divide it into cells δx, centered at x and of some small size, commonly taken to be 2πℏ3N. The cells cover the entire phase space. To account for the identical nature of the particles, the number of cells and the volume of the phase space is assumed to be divided by N! to give distinct arrangements of the particles in the cells, which are indexed by *k*=1,2,⋯ and write them as δxk; the center of δxk is at xk. These cells represents the microstates mk. The energy and probability of these cells are denoted by Ek,pk in which Ek(W) is a function of W. Different choices of pk for the same set mk,Ek describes different macrostates for a given W, one of which corresponding to pkeq uniquely specifies an EQ macrostate Meq; all other states are called NEQ macrostates M. Among M are some special macrostates Mieq that are said to be in internal equilibrium (IEQ); the rest are nonIEQ macrostates Mnieq. An arbitrary macrostate Marb refers to either an EQ or a NEQ macrostate.

We use a suffix 0 to denote all quantities pertaining to $\Sigma_{0}$, a tilde $\left(\tilde{}\right)$ for all quantities pertaining to $\tilde{\Sigma }$, and no suffix for all quantities pertaining to Σ even if it is isolated. Thus, the set of observables are denoted by $X_{0},\tilde{X}$ and X, respectively, and the set of state variables by $Z_{0},\tilde{Z}$ and Z, respectively, in the state space $S_{Z}$; the set of internal variables are $\xi_{0},\tilde{\xi }$ and ξ, respectively. As $\tilde{\Sigma }$ is taken to be in EQ, weakly interacting with and is extremely large compared to Σ, all its fields can be safely taken to be the fields associated with $\Sigma_{0}$ so can be denoted by using the suffix 0.

In the discrete approach, Σ and $\tilde{\Sigma }$ are spatially disjoint so
$$V_{0}=V+\tilde{V}.$$

They are weakly interacting so that their energies are *quasi-additive*
$$E_{0}=E+\tilde{E}+E_{\mathrm{int}}≃E+\tilde{E},$$
where $E_{\mathrm{int}}$ is the weak interaction energy between Σ and $\tilde{\Sigma }$ and can be neglected to a good approximation. We also take them to be *quasi-independent* [41] so that their entropies also become quasiadditive:(2)$$\begin{array}{cc} S_{0}\left(X_{0},t\right) & =S\left(X\left(t\right),t\right)+\tilde{S}\left(\tilde{X}\left(t\right)\right)+S_{\mathrm{corr}}\left(t\right)\\ \; & ≃S\left(X\left(t\right),t\right)+\tilde{S}\left(\tilde{X}\left(t\right)\right); \end{array}$$
here, $S_{\mathrm{corr}}\left(t\right)$ is a negligible contribution to the entropy due to quasi-independence between Σ and $\tilde{\Sigma }$, and can also be neglected to a good approximation. The entropy $\tilde{S}$ has no explicit time dependence as $\tilde{\Sigma }$ is always assumed to be in equilibrium, and $X_{0}$ remains constant for the isolated system $\Sigma_{0}$. The discussion of quasi-independence and its distinction from weak interaction has been carefully presented elsewhere ([41] ($S_{\mathrm{corr}}$ was called $S_{\mathrm{int}}$ there; however, $S_{\mathrm{corr}}$ seems to be more appropriate)) for the first time, which we summarize as follows. The concept of quasi-independence is determined by the thermodynamic concept of *correlation length*$\lambda_{\mathrm{corr}}$, which is a property of macrostates, and can be much larger than the interaction length between particles. A simple well-known example is of the correlation length of a nearest neighbor Ising model, which can be extremely large near a critical point than the nearest neighbor distance between the spins. This distinction is usually not made explicit in the literature. For quasi-independence between Σ and $\tilde{\Sigma }$, we require their sizes to be larger than $\lambda_{\mathrm{corr}}$. Throughout this review, we will think of the above *approximate equalities* as equalities to make the energies to be additive by neglecting the interaction energy between Σ and $\tilde{\Sigma }$, which is a standard practice in the field, but also assuming quasi-independence between them to make the entropies to be additive, which is not usually mentioned as a requirement in the literature.

For a reversible process, the entropy of each macrostate $M_{\mathrm{eq}}\left(t\right)\in S_{X}$ of a body along the process is a state function of X(t), but not for an irreversible process for which $M\left(t\right)\notin S_{X}$. Their entropies are written as S(X(t),t) [51,52] with an explicit time dependence. In general [14,51,52,53],
(3)S(X(t),t)≤S(X(t));fixedX(t).

The equilibrium values of various entropies are always denoted with no explicit time dependence such as by $S_{0}\left(X_{0}\right)$ for $\Sigma_{0}$. These entropies represent the maximum possible values of the entropies of a body as it relaxes and comes to equilibrium for a given set of observables. Once in equilibrium, the body will have no memory of its original macrostate. The set $X_{0}$, which includes its energy $E_{0}$ among others, remains constant for $\Sigma_{0}$ as it relaxes. This notion is also extended to a body in internal equilibrium.

**Notation** **1.**
*We use modern notation [3,16] and its extension, see Figure 1, that will be extremely useful to understand the usefulness of our novel approach. Any infinitesimal and extensive system-intrinsic quantity dY(t) during an arbitrary process dP can be partitioned as*

(4)
$$dY\left(t\right)\equiv d_{e}Y\left(t\right)+d_{i}Y\left(t\right),$$

*where $d_{e}Y\left(t\right)$ is the change caused by exchange (e) with the medium and $d_{i}Y\left(t\right)$ is its change due to internal or irreversible (i) processes going on within the system.*


Throughout the review, we assume that there is only one species of stable particles, whose number *N* is an observable, and is held fixed to fix the size of Σ. We can list *N* in X if we keep another observable such as *V* fixed to fix the size of the system. Here, we will keep *N* fixed for the size. If there are several species k=1,2,⋯,r of particles that undergo *l* distinct chemical reactions among themselves, then the individual numbers $N_{k},k\in \left\{1,2,\cdots ,r\right\}$ of the species are not constant, only their total *N* remains constant. In this case, we need distinct $l^{\prime }≐l-1$ extents of reaction [3,16] as internal variables in Z as has been discussed later. If the species do not undergo chemical reactions among themselves, then $N_{k}$’s are individually observables. In this case, we can choose $l^{\prime }$ independent numbers that are contained in X. In this review, we only consider a single species for simplicity.

### 2.2. Some Definitions and New Concepts

**Definition** **1.**
*Observables X=(E,V,N,⋯) of a system are quantities that can be controlled from outside the system, and internal variables $\xi =(\xi_{1},\xi_{2},\xi_{3},\cdots )$ are quantities that cannot be controlled. Their collection Z=X∪ξ is called the set of state variables in the state space S.*


**Definition** **2.**
*A system-intrinsic quantity is a quantity that pertains to the system alone and can be used to characterize the system. A medium–intrinsic quantity is a quantity that is solely determined by the medium alone and can be used to characterize the exchange between the system and the medium.*


**Definition** **3.**
*A macrostate in $S_{X}$ or $S_{Z}$ is a collection $\left\{m_{k},p_{k}\right\}$ of microstates $m_{k}$ and their probabilities $p_{k},k=1,2,\cdots$. In general, $p_{k}$ are functions of X or Z, depending on the state space. They are implicit function of time t through them; they may also depend explicitly on time t if not unique in the state space.. For an EQ or an IEQ macrostate, $p_{k}$ have no explicit dependence on t. For EQ states, $p_{k\;}$have no time-dependence. It is through the microstate probabilities that thermodynamics gets its stochastic nature.*


**Definition** **4.**
*The collection $\left\{m_{k},p_{k}\right\}$ provides a complete microscopic or statistical mechanical description of thermodynamics for $M_{\mathrm{arb}}$ in some state space S in which one deals with macroscopic or ensemble averages, see Definition 12, over $\left\{m_{k}\right\}$ of microstate variables. The same collection $\left\{m_{k},p_{k}\right\}$ also provides a microscopic description of a microstate and its probability in any arbitrary process.*


**Definition** **5.**
*The nonequilibrium macrostates can be classified into two classes:*
(a)
*Internal-equilibrium macrostates (IEQ): The nonequilibrium entropy S(X,t) for such a macrostate is a state function S(Z) in the larger nonequilibrium state space $S_{Z}$ spanned by Z; $S_{X}$ is a proper subspace of $S_{Z}$: $S_{X}\subset S_{Z}$. As there is no explicit time dependence, there is no memory of the initial macrostate in IEQ macrostates.*
(b)
*Non-internal-equilibrium macrostates (NIEQ): The nonequilibrium entropy for such a macrostate is not a state function of the state variable Z. Accordingly, we denote it by S(Z,t) with an explicit time dependence. The explicit time dependence gives rise to memory effects in these NEQ macrostates that lie outside the nonequilibrium state space $S_{Z}$. A NIEQ macrostate in $S_{Z}$ becomes an IEQ macrostate in a larger state space $S_{Z^{\prime }},Z^{\prime }⊃Z$, with a proper choice of $Z^{\prime }$.*



**Definition** **6.***An arbitrary macrostate*(ARB) *of a system**refers to all possible thermodynamic states, which include EQ macrostates, and NEQ macrostates with and without the memory of the initial macrostate. We denote an arbitrary macrostate by $M_{\mathrm{arb}}$, NEQ macrostates by M, EQ macrostates by $M_{\mathrm{eq}}$, and IEQ macrostates by $M_{\mathrm{ieq}}$.*

**Definition** **7.**
*Thermodynamic entropy S is defined by the Gibbs fundamental relation for a macrostate.*


**Definition** **8.**
*Statistical entropy S for $M_{\mathrm{arb}}$ is defined by its microstates by Gibbs formulation.*


**Definition** **9.**
*Changes in quantities such as S,E,V,⋯ in an infinitesimal processes δP are denoted by dS,dE,dV,⋯; changes during a finite process P are denoted by ΔS,ΔE,ΔV,⋯.*


**Definition** **10.**
*The path $\gamma_{P}$ of a macrostate M is the path it takes in S during a process P. The trajectory $\gamma_{k}$ is the trajectory a microstate $m_{k}$ takes in time in S during the process P.*


As $m_{k}$ evolves due to Hamilton’s equations of motion for given W, the variation of $x_{k}$ has no effect on $E_{k}$. Therefore, we will no longer exhibit x and simply use H(W) for the Hamiltonian. The microenergy $E_{k}$ changes isentropically as W changes without changing $p_{k}$ [54]. Accordingly, the generalized macrowork dW does not generate any stochasticity. The latter is brought about by the generalized macroheat dQ, which changes $p_{k}$ but without changing $E_{k}$. In the MNEQT,
(5a)dQ≡TdS
in terms of the temperature
(5b)T=∂E/∂S
and dS of Σ. The Equation ([Disp-formula FD5a-entropy-23-01584]) is a general result in the MNEQT.

It is convenient to introduce φ=(S,Z) as the set of all thermodynamic macrovariables, which takes the microvalue $\varphi_{k}$ on $m_{k}$.

**Definition** **11.**
*Macropartition: As suggested in Figure 1 and Notation 1, the change*

(6)
$$d\varphi ≐d_{e}\varphi +d_{i}\varphi$$

*in the SI-macrovariable **φ** of Σ consists of two parts: the MI-change $d_{e}\varphi$ is the change due to exchange with $\tilde{\Sigma }$, and $d_{i}\varphi$ is the irreversible change occurring within Σ.*


This is an extension of the standard partition for the entropy change [3,16]
(7)$$dS≐d_{e}S+d_{i}S.$$

For *E* and *V*, the partitions are
(8)$$dE≐d_{e}E+d_{i}E,dV≐d_{e}V+d_{i}V,$$
except that
(9)$$d_{i}E\equiv 0,d_{i}V\equiv 0,$$
for the simple reason that internal processes cannot change *E* and *V*, respectively. For *N*, the partition is
$$dN≐d_{e}N+d_{i}N,$$
with $d_{i}N$ present when there is chemical reaction. We will find the shorthand notation
(10)$$d_{\alpha }=\left(d,d_{e},d_{i}\right)$$
quite useful in the following for the various infinitesimal contributions. These linear operators satisfy
(11)$$d\equiv d_{e}+d_{i}.$$

**Definition** **12.**
*Ensemble Average: In NEQT, any thermodynamic macroquantity **φ** is obtained by the instantaneous ensemble average*

(12)
$$\varphi \equiv \langle \varphi \rangle =\sum_{k}p_{k}\varphi_{k},$$

*where **φ** takes microvalues $\varphi_{k}$ on $m_{k}$ at that instant with probability $p_{k}$.*


We have used the standard convention to write φ for $\langle \varphi \rangle$. For example, the internal energy *E* is given by
(13)$$E\equiv \langle E\rangle =\sum_{k}p_{k}E_{k},$$
while the statistical entropy, often called the Gibbs entropy, is given by
(14)$$S\equiv \langle S\rangle =\sum_{k}p_{k}S_{k}=-\sum_{k}p_{k}\ln p_{k}$$
where the *microentropy*$S_{k}$ is
(15)$$S_{k}\equiv -\eta_{k}≐-\ln p_{k};$$
in terms of Gibbs’ *index of probability* $\eta_{k}≐\ln p_{k}$ ([19] (p. 16)).

**Definition** **13.**
*Micropartition:The macropartition in Equation (Equation 6) is extended to microvariable $\varphi_{k}$:*

(16a)
$$d\varphi_{k}≐d_{e}\varphi_{k}+d_{i}\varphi_{k}.$$


*Thus,*

(16b)
$$dE_{k}≐d_{e}E_{k}+d_{i}E_{k},dS_{k}≐d_{e}S_{k}+d_{i}S_{k}.$$


*The micropartition also applies to $dp_{k}$:*

(17a)
$$dp_{k}≐d_{e}p_{k}+d_{i}p_{k},$$


*We define*

(17b)
$$d_{\alpha }\eta_{k}≐\frac{d_{\alpha }p_{k}}{p_{k}}.$$



In a process, φ undergoes infinitesimal changes $d_{\alpha }\varphi_{k}$ at fixed $p_{k}$, or infinitesimal changes $d_{\alpha }p_{k}$ at fixed $\phi_{k}$. The changes result in two distinct ensemble averages or process quantities.

**Definition** **14.**
*Infinitesimal macroquantities $\langle d_{\alpha }\varphi \rangle$ are ensemble averages*

(18a)
$$d_{\alpha }\varphi_{m}\equiv \langle d_{\alpha }\varphi \rangle =\sum_{k}p_{k}d_{\alpha }\varphi_{k}$$

*at fixed $\left\{p_{k}\right\}$ so they are isentropic. We identify them as mechanical macroquantity and write it as $d_{\alpha }\varphi_{m}$. Infinitesimal macroquantities*

(18b)
$$d_{\alpha }\varphi_{s}≐\langle \varphi_{k}d_{\alpha }\eta \rangle$$

*that are ensemble averages involving $\left\{d_{\alpha }p_{k}\right\}$ are identified as stochastic macroquantities and written as $d_{\alpha }\varphi_{s}$. Together, they determine the change $d_{\alpha }\varphi$:*

(19)
$$d_{\alpha }\varphi \equiv d_{\alpha }\langle \varphi \rangle ≐d_{\alpha }\varphi_{m}+d_{\alpha }\varphi_{s}.$$



We must carefully distinguish $d_{\alpha }\langle \varphi \rangle$ and $\langle d_{\alpha }\varphi \rangle$. For *E*, we will use instead the following notation:(20)$$d_{\alpha }Q=d_{\alpha }E_{s},d_{\alpha }W=-d_{\alpha }E_{m},$$
from which follows
(21)$$d_{\alpha }E=d_{\alpha }Q-d_{\alpha }W.$$

Using Equation (Equation 9) for $d_{i}E$, we have the following thermodynamic identity:(22)$$d_{i}Q\equiv d_{i}W.$$

For $d_{\alpha }=d,d_{e}$, we have the following SI- and MI- formulation of the first law:
(23a)$$\begin{array}{cc} dE & =dQ-dW \end{array}$$
(23b)$$\begin{array}{cc} dE & =d_{e}Q-d_{e}W, \end{array}$$
where we have used the identity $dE=d_{e}E$. The top equation is also known as the *Gibbs fundamental relation*.

We can use the operator identity in Equation (Equation 11) to introduce the following important identities following Notation 1
(24)$$\begin{array}{cc} dW & =d_{e}W+d_{i}W,dQ=d_{e}Q+d_{i}Q, \end{array}$$
(25)$$\begin{array}{cc} dW_{k} & =d_{e}W_{k}+d_{i}W_{k},dQ_{k}=d_{e}Q_{k}+d_{i}Q_{k}, \end{array}$$
that will be very useful in the MNEQT. For an isolated system, $d_{e}W\equiv 0,d_{e}Q\equiv 0$. Note that dW,dQ, etc. do not represent changes in any SI-macrovariable.

**Definition** **15.**
*We simply call dQ and dW macroheat and macrowork, respectively, unless clarity is needed and use exchange macroheat for $d_{e}Q$ and exchange macrowork for $d_{e}W$, irreversible macroheat for $d_{i}Q$ and irreversible macrowork for $d_{i}W$, respectively.*


Manipulating w such as the “volume” *V* from the outside through ${\tilde{\Sigma }}_{w}$ requires some external “force” $F_{w0}$, such as the external pressure $P_{0}$ to do some “exchange macrowork” $d\tilde{W}$ on Σ. We have $d\tilde{w}=d_{e}\tilde{w}=-d_{e}w$, and
(26)$$d\tilde{W}≐F_{w0}\cdot d\tilde{w}=-d_{e}W≐-F_{w0}\cdot d_{e}w,$$
where $F_{w0}=\left(P_{0},...,A_{0}=0\right)$; see Figure 1. We use $F_{w0}=\left(f_{w0},A_{0}=0\right)$.

In a NEQ system, the generalized force $F_{w}$ in Σ differs from $F_{w0}$. The resulting macrowork done by Σ is
(27)$$dW≐F_{w}\cdot dW.$$

This is the SI-macrowork and differs from the MI-macrowork $d\tilde{W}$ $=-d_{e}W$. Here,
(28)$$F_{w}≐-\partial E/\partial W=\left(P\left(t\right),...,A\left(t\right)\right)=\left(f\left(t\right),A\left(t\right)\right);$$
see Figure 1. The SI-affinity A corresponding to ξ [16,50] is nonzero, except in EQ, when it vanishes: $A_{\mathrm{eq}}\equiv A_{0}=0=0$ [3,16]. The " SI-macrowork" $dW_{\xi }$ done by Σ as ξ varies is
(29)$$dW_{\xi }≐A\cdot d\xi .$$

Even for an isolated NEQ system, $dW_{\xi }$ will not vanish; it vanishes only in EQ, since ξ does no work when $A_{0}=0$; however, $F_{w0},d\tilde{W}$ and $d_{e}W$ are unaffected by the presence of ξ.

The macroforce *imbalance* is the difference
(30a)$$\Delta F^{w}≐\left(F_{w}-F_{w0}\right)=\left(f_{w}-f_{w0},A\right).$$

In general, A controls the behavior of ξ in M [16,50] and vanishes when EQ is reached [3,16]. Here, we will take a more general view of A, and extend its definition to X also. In particular, $\Delta F^{h}≐T_{0}-T$ also plays the role of an affinity [55] so we can include it with $\Delta F^{w}$ to form set of *thermodynamic macroforces* or of *macroforce imbalance*:
(30b)$$\Delta F≐(T_{0}-T,f_{w}-f_{w0},A).$$

The same reasoning also shows that ΔF plays the role of an activity.

The *irreversible macrowork*$d_{i}W≐dW-d_{e}W\equiv dW+d\tilde{W}$ is given by
(31)$$d_{i}W≐\left(f_{w}-f_{w0}\right)\cdot d_{e}w+f_{w}\cdot d_{i}w+A\cdot d\xi \geq 0.$$

For the sake of clarity, we will take *V* as a symbolic representation of X, and a single ξ as an internal variable in many examples. Then W=(V,ξ) is the macrowork parameter. In this case, we have
(32a)$$\begin{array}{cc} dW & =PdV+Ad\xi ,d_{e}W=P_{0}dV, \end{array}$$
(32b)$$\begin{array}{cc} d_{i}W & =(P-P_{0})dV+Ad\xi , \end{array}$$
provided $d_{i}V=0$.

The microanalogue of $\Delta F^{w}$ is the internal microforce imbalance
(33)$$\Delta F_{k}^{w}≐\left(f_{wk}-f_{w0},A_{k}\right),$$
which determines the internal microwork
(34)$$d_{i}W_{k}≐\left(f_{wk}-f_{w0}\right)\cdot d_{e}w+f_{wk}\cdot d_{i}w+A_{k}\cdot d\xi ,$$
as the exchange microwork is
(35)$$d_{e}W_{k}≐f_{w0}\cdot d_{e}w=d_{e}W,\forall k.$$

**Remark** **1.**
*It should be warned that dQ in Equations (15) and (16) in [55] and Sec. IVB in [41] refers only to the exchange macroheat; recall Definition 15. Thus, the usage there is different from the generalized macroheat in the review.*


## 3. Internal Variables

We should emphasize that the concept of internal variables and their usefulness in NEQT has a long history. We refer the reader to an excellent exposition of this topic in the monograph by Maugin ([40] (see Ch. 4)). We consider a few simple examples to justify why internal variables are needed to uniquely specify a M, and how to identify them for various systems.

It should be stated that in order to capture a NEQ process, internal variables are usually *necessary*. Another way to appreciate this fact is to realize that

**Remark** **2.**
*For an isolated system, all the observables in $X_{0}$ are fixed so if the entropy is a function of $X_{0}$ only, it cannot change [41,51,52,55] even if the system is out of Equation*


Thus, we need additional independent variables to ensure the law of increase of entropy for a NEQ isolated system. A point in $S_{X}$ represents $M_{\mathrm{eq}}$, but a point $S_{Z}$ represents M. In EQ, internal variables are no longer independent of the observables. Consequently, their affinities (see later) vanish in Equation It is common to define the internal variables so their EQ values vanish. We now discuss various scenarios where they are needed for a proper consideration.

### 3.1. A Two-Level System

Consider a NEQ system of *N* particles such as Ising spins, each of which can be in two levels, forming an isolated system $\Sigma_{0}$ of volume *V*. Let $\rho_{l}$ and $e_{l}\left(V\right),l=1,2$ denote the probabilities and energies of the two levels of a particle in a NEQ macrostate so that $\rho_{1},\rho_{2}$ keep changing. We have assumed that $e_{l}\left(V\right)$ depends on the observable *V* only, which happens to be constant for $\Sigma_{0}$. We have $e=\rho_{1}e_{1}\left(V\right)+\rho_{2}e_{2}\left(V\right)$ for the average energy per particle, which is also a constant for $\Sigma_{0}$, and
$$d\rho_{1}+d\rho_{2}=0$$
as a consequence of $\rho_{1}+\rho_{2}=1$. Using de=0, we get
$$d\rho_{1}+d\rho_{2}e_{2}/e_{1}=0,$$
which, for $e_{1}\neq e_{2}$, is inconsistent with the first equation (unless $d\rho_{1}=0=d\rho_{2}$, which corresponds to EQ). Thus, $e_{l}\left(V\right)$ cannot be treated as constant in evaluating de. In other words, there must be an extra dependence in $e_{l}$ so that
$$e_{1}d\rho_{1}+d\rho_{2}e_{2}+\rho_{1}de_{1}+\rho_{2}de_{2}=0,$$
and the inconsistency is removed. This extra dependence must be due to *independent* internal variables that are not controlled from the outside (isolated system) so they continue to relax in $\Sigma_{0}$ as it approaches Equation Let us imagine that there is a single internal variable ξ so that we can express $e_{l}$ as $e_{l}\left(V,\xi \right)$ in which ξ continues to change as the system comes to equilibrium. The above equation then relates $d\rho_{1}$ and dξ; they both vanish simultaneously as EQ is reached. We also see that without any ξ, the isolated system cannot equilibrate; see Remark 2.

### 3.2. A Many-Level System

The above discussion is easily extended to a Σ with many energy levels of a particle with the same conclusion that at least a single internal variable is required to express $e_{l}=e_{l}\left(V,\xi \right)$ for each level *l*. We can also visualize the above system in terms of microstates. A microstate $m_{k}$ refers to a particular distribution of the *N* particles in any of the levels with energy $E_{k}=\sum_{l}N_{l}e_{l}$, where $N_{l}$ is the number of particles in the *l*th level, and is obviously a function of N,V,ξ so we will express it as $E_{k}\left(V,\xi \right)$; we suppress the dependence on *N*. This makes the average energy of the system also a function of V,ξ, which we express as E(V,ξ).

### 3.3. Disparate Degrees of Freedom

In classical statistical mechanics, the kinetic and potential energies *K* and *U*, respectively, are functions of independent variables. Only their sum K+U=E can be controlled from the outside, but not individually. Thus, one of them can be treated as an internal variable. In a NEQ macrostates, each term can have its own temperature. Only in EQ, do they have the same temperature.

This has an important consequence for glasses, where the vibrational degrees of freedom (dof${}_{v}$) come to EQ with the heat bath at $T_{0}$ faster than the configurational degrees of freedom (dof${}_{c}$), which have a different temperature than $T_{0}$. The disparity in dof${}_{v}$ and dof${}_{c}$ cannot be controlled by the observer so it plays the role of an internal variable. A well-known equation, the Tool-Narayanaswamy equation is concerned with this disparity and is discussed in Section 11.

Consider a collection of semiflexible polymers in a solution on a lattice. The interaction energy *E* consists of several additive terms as discussed in ([41], Equation (Equation 40)): the interaction energy $E_{\mathrm{ps}}$ between the polymer and the solvent, the interaction energy $E_{\mathrm{ss}}$ between the solvent, the interaction energy $E_{\mathrm{pp}}$ between polymers. Only the total *E* can be controlled from the outside so the remaining terms determine several internal variables.

In the examples above, the internal variables are not due to spatial inhomogeneity. An EQ system is uniform. Thus, the presence of ξ suggests some sort of nonuniformity in the system. To appreciate its physics, we consider a slightly different situation below as a possible example of nonuniformity.

### 3.4. Nonuniformity

(a) We consider as a simple NEQ example a composite isolated system Σ, see Figure 3, consisting of two subsystems $\Sigma_{1}$ and $\Sigma_{2}$ of identical volumes and numbers of particles but at different temperatures $T_{1}$ and $T_{2}$ at any time $t<\tau_{\mathrm{eq}}$ before EQ is reached at $t=\tau_{\mathrm{eq}}$ so the subsystems have different time-dependent energies $E_{1}$ and $E_{2}$, respectively. We assume a diathermal wall separating $\Sigma_{1}$ and $\Sigma_{2}$. Treating each subsystem in EQ at each *t*, we write their entropies as $S_{1}\left(E_{1},V/2,N/2\right)$ and $S_{2}\left(E_{2},V/2,N/2\right)$, which we simply show as $S_{1}\left(E_{1}\right)$ and $S_{2}\left(E_{2}\right)$ as we will not let their volumes and particles numbers change. The entropy $S≐S_{1}\left(E_{1}\right)+S_{2}\left(E_{2}\right)$ of Σ is a function of $E_{1}$ and $E_{2}$. Obviously, Σ is in a NEQ macrostate at each $t<\tau_{\mathrm{eq}}$. As $E_{1}$ and $E_{2}$ do not refer to Σ, we form two independent combinations from $E_{1}$ and $E_{2}$
(36)$$E=E_{1}+E_{2},\xi =E_{1}-E_{2},$$
that refer to Σ so that we can express the entropy as S(E,ξ) for Σ treated as a blackbox $\Sigma_{B}$; we do not need to know about its interior (its inhomogeneity) anymore. Here, ξ plays the role of an internal variable, which continues to relax towards zero as Σ approaches Equation For given *E* and ξ, S(E,ξ) has the maximum possible values since both $S_{1}$ and $S_{2}$ have their maximum value. As we will see below, this is the idea behind the concept of *internal equilibrium* in which S(E,ξ) is a state function of state variables and continues to increase as ξ decreases and vanishes in Equation In this macrostate, S(E,ξ=0) has the maximum possible value for fixed *E* so it becomes a state function; see Definition 16. This case and its various extensions are investigated in MNEQT in Section 10.3.

(b) We can easily extend the model to include four identical subsystems of fixed and identical volumes and numbers of particles, but of different energies $E_{1},E_{2},E_{3}$, and $E_{4}$. Instead of using these 4 independent variables, we can use the following four independent combinations
(37)$$\begin{array}{cc} E & =E_{1}+E_{2}+E_{3}+E_{4}=\mathrm{constant},\\ \xi & =E_{1}+E_{2}-E_{3}-E_{4},\\ \;\xi^{\prime } & =E_{1}-E_{2}+E_{3}-E_{4},\\ \xi^{\prime \prime } & =E_{1}-E_{2}-E_{3}+E_{4}, \end{array}$$
to express the entropy of Σ as $S(E,\xi ,\xi^{\prime },\xi^{\prime \prime })$. The pattern of extension for this simple case of energy inhomogeneity. is evident.

(c) We make the model a bit more interesting by allowing the volumes $V_{1}$ and $V_{2}$ to also vary as Σ equilibrates. Apart from the internal variable ξ, we require another internal variable $\xi^{\prime }$ to form two independent combinations
(38)$$V=V_{1}+V_{2}=\mathrm{constant},\;\xi^{\prime }=V_{1}-V_{2}$$
so that we can use $S\left(E,V,\xi ,\xi^{\prime }\right)≐S_{1\mathrm{eq}}\left(E_{1},V_{1}\right)+S_{2\mathrm{eq}}\left(E_{2},V_{2}\right)$ for the entropy of Σ in terms of the entropies of $\Sigma_{1}$ and $\Sigma_{2}$.

(d) In the above examples, we have assumed the subsystems to be in Equation We now consider when the subsystems are in IEquation We consider the simple case of two subsystems $\Sigma_{1}$ and $\Sigma_{2}$ of identical volumes and numbers of particles. Each subsystem is in different IEQ macrostates described by $E_{1},\xi_{1}$ and $E_{2},\xi_{2}$. We now construct four independent combinations
(39)$$\begin{array}{cc} E & =E_{1}+E_{2}=\mathrm{constant},\xi =E_{1}-E_{2},\\ \xi^{\prime } & =\xi_{1}+\xi_{2},\xi^{\prime \prime }=\xi_{1}-\xi_{2}, \end{array}$$
which can be used to express the entropy of Σ as $S(E,\xi ,\xi^{\prime },\xi^{\prime \prime })$.

(e) The example in (a) can be easily extended to the case of expansion and contraction by replacing $E,E_{1}$, and $E_{2}$ by $N,N_{L}$, and $N_{R}$, see Figure 6, to describe the diffusion of particles [56]. The role of β and *E*, etc. are played by βμ and *N*, etc.

### 3.5. Relative Motion in Piston-Gas System

We now consider the motion of the piston in Figure 4a because of the pressure difference across it. The discussion also shows how the Hamiltonian becomes dependent on internal variables, and how the system is maintained *stationary* despite motion of its parts.

Let $P_{p}$ denote the momentum of the piston. The gas, the cylinder and the piston constitute the system Σ. We have a gas of mass $M_{g}$ in the cylindrical volume $V_{g}$, the piston of mass $M_{p}$, and the rigid cylinder (with its end opposite to the piston closed) of mass $M_{c}$. However, we will consider the composite subsystem $\Sigma_{\mathrm{gc}}=\Sigma_{g}\cup \Sigma_{c}$ so that with $\Sigma_{p}$ it makes up Σ. The Hamiltonian H of the system is the sum of $H_{\mathrm{gc}}$ of the gas and cylinder, $H_{p}$ of the piston, the interaction Hamiltonian $H_{\mathrm{int}}$ between the two subsystems $\Sigma_{\mathrm{gc}}\;$ and $\Sigma_{p}$, and the interaction Hamiltonian $H_{\mathrm{sm}}$ between Σ and $\tilde{\Sigma }$. As is customary, see the discussion in Section 2, we will neglect $H_{\mathrm{sm}}$ here. We assume that the centers-of-mass of $\Sigma_{\mathrm{gc}}$ and $\Sigma_{p}$ are moving with respect to the medium with linear momentum $P_{\mathrm{gc}}$ and $P_{p}$, respectively. We do not allow any rotation for simplicity. We assume that
(40)$$P_{\mathrm{gc}}+P_{p}=0,$$
so that Σ is at rest with respect to the medium. Thus,
(41)$$H\left(\left.x\right|V,P_{\mathrm{gc}},P_{p}\right)=\sum_{\lambda }H_{\lambda }\left(\left.x_{\lambda }\right|V_{\lambda },P_{\lambda }\right)+H_{\mathrm{int}},$$
where λ= gc, p, $x_{\lambda }=\left(r_{\lambda },p_{\lambda }\right)$ denotes a point in the phase space $\Gamma_{\lambda }$ of $\Sigma_{\lambda }$; $V_{\lambda \;}$is the volume of $\Sigma_{\lambda }$, and $V=V_{\mathrm{gc}}+V_{p}$ is the volume of Σ. We do not exhibit the number of particles $N_{g},N_{c},N_{p}$ as we keep them fixed. We let x denotes the collection ($x_{\mathrm{gc}},x_{p}$). Thus, $H(\left.x\right|V,P_{\mathrm{gc}},P_{p})$ and the average energy *E* depend on the parameters $V,P_{\mathrm{gc}},P_{p}$. As the relative motion cannot be controlled from the outside, one of the momenta plays the role of an internal variable.

We discuss the example of the spring in Figure 4b will be discussed in Section 13.

### 3.6. Extended State Space

It should be clear from above that we can identify the entropy

If we divide Σ into many subsystems $\left\{\Sigma_{i}\right\}$ so that they are *all* quasi-independent, then the entropy additivity gives
$$S\left(X\left(t\right),t\right)=\sum_{i}S_{i}\left(X_{i}\left(t\right),t\right).$$

As we will be dealing with the Hamiltonian of the system, it is useful to introduce the notation in Equation (Equation 1) with W=(w,ξ). Then, *E* and $E_{k}$ become a function of W as we will show in Section 5. Here, W appears as a *parameter* in the Hamiltonian, which we will write as $H(\left.x\right|W)$, where x is a point (collection of coordinates and momenta of the particles) in the phase space Γ(W) specified by W. As an example, $V,P_{\mathrm{gc}},P_{p}$ are the parameters in Section 13. When the system moves about in the phase space Γ(W), x changes but W as a parameter remains fixed in a state subspace $S_{W}\subset S_{Z}$; see the discussion of Equation ([Disp-formula FD55a-entropy-23-01584]).

It is important to draw attention to the following important distinction between the Hamiltonian H and the ensemble average energy *E*; see Equation (Equation 44). While *E* accounts for the stochasticity through microstate probabilities, the use of the Hamiltonian is going to be restricted to a particular microstate. In other words, the Hamiltonian depends on x and W but the energy depends on the entropy *S* and W. The energy $E_{k}$ of $m_{k}$, on the other hand, depends only on W and denotes the value of H for $m_{k}$. In the following, we will always treat Hamiltonians and microstate energies as equivalent description, which does not depend on knowing $\left\{p_{k}\right\}$; the average energies depend on $\left\{p_{k}\right\}$ for their definition; see Equation (Equation 13).

## 4. NEQ Entropy

### 4.1. Determination of S

The uniqueness issue about the NEQ macrostate says nothing about the entropy of an arbitrary (so it may be nonunique) macrostate $M:\left\{m_{k},p_{k}\right\}$, which is *always* given by the Gibbs entropy in Equation (Equation 14); see also [57]. The ensemble averaging implies that the entropy is a statistical concept, as is the energy $E=\langle E\rangle$, Equation (Equation 13).

We now justify the Gibbs’ statistical formulation of *S* for any arbitrary M in thermodynamics. The demonstration follows a very simple combinatorial argument [52] using Boltzmann concept of thermodynamic entropy. In the demonstration, M is not required to be uniquely identified. This entropy satisfies the *law of increase of entropy* as is easily seen by the discussion by Landau and Lifshitz for a NEQ ideal gas in $S_{X}$ to derive the equilibrium distribution [14] (Equation (7.9) here shows how this entropy formulation emerges in statistical physics. It is applicable to both EQ and NEQ macrostates as is clear from Section 40 (see Equation (40.7) in particular) dealing with NEQ ideal gas). Thus, the form in Equation (Equation 14) is not restricted to only uniquely identified M’s. Hopefully, this will become clear below.

**Proposition** **1.**
***The Second Law:** The NEQ Gibbs entropy $S_{0}\left(X_{0},t\right)$ of an isolated system $\Sigma_{0}$ is bounded above by its equilibrium entropy $S_{0}\left(X_{0}\right)$ and continuously increases towards it so that [14]*

(42)
$$dS_{0}\left(X_{0},t\right)/dt\geq 0.$$



### 4.2. General Formulation of the Statistical Entropy

We focus on a macrostate M(t)of some body Σ at a given instant *t*, which refers to the set $m=\;\left\{m_{k}\right\}$ of microstates and their probabilities $p=\left\{p_{k}\right\}$. The microstates are specified by $\left(E_{k}\left(t\right),W\left(t\right)\right)$, and may not uniquely specify the macrostate M(t). Thus, even the set m need not be uniquely specified. In the following, we will use the set Z(t)=(E(t),W(t)) for the set m for simplicity. We will also denote Z(t) by $\bar{Z}$ so that we can separate out the explicit variation due to *t*. For simplicity, we suppress *t* in M in the following. For the computation of combinatorics, the probabilities are handled in the following abstract way. We consider a large number $N=CW(\bar{Z})$ of independent *replicas* or *samples* of Σ, with C some large integer constant and $W(\bar{Z})$ the number of distinct microstates $m_{k}$. We will see that $W(\bar{Z})$ is determined by $m_{k}$’s having nonzero probabilities. We will call them *available* microstates. The samples should be thought of as identically prepared experimental samples [53].

Let $\Gamma (\bar{Z})$ denote the sample space spanned by $\left\{m_{k}\right\}$, and let $N_{k}\left(t\right)$ denote the number of *k*th samples (samples specified by $m_{k}$) so that
(43)$$0\leq p_{k}\left(t\right)=N_{k}\left(t\right)/N\leq 1;\;\;\sum_{k=1}^{W(\bar{Z})}N_{k}\left(t\right)=N.$$

The above sample space is a generalization of the *ensemble* introduced by Gibbs, except that the latter is restricted to an equilibrium body, whereas $\Gamma (\bar{Z})$ refers to the body in any arbitrary macrostate so that $p_{k}$ may be time-dependent, and need not be unique. The *ensemble average* of some quantity Z over these samples is given by Equation (Equation 12). Thus,
(44)$$\langle Z\rangle \equiv \sum_{k=1}^{W(\bar{Z})}p_{k}\left(t\right)Z_{k},\;\;\sum_{k=1}^{W(\bar{Z})}p_{k}\left(t\right)\equiv 1,$$
where $Z_{k}$ is the value of Z in $m_{k}$.

The samples are, by definition, *independent* of each other so that there are no correlations among them. Because of this, we can treat the samples $\left\{m_{k}\right\}$ to be the outcomes of some random variable, the macrostate M(t). This independence property of the outcomes is crucial in the following. They may be equiprobable but not necessarily. The number of ways W to arrange the N samples into $W(\bar{Z})$ distinct microstates is
(45)$$W\equiv N!/\prod_{k}N_{k}\left(t\right)!.$$

Taking its natural log, as proposed by Boltzmann, to obtain an *additive* quantity per sample as
(46)S≡lnW/N,
and using Stirling’s approximation, we see easily that it can be written as the average of the negative of Gibbs’ index of probability:(47)$$S\left(\bar{Z},t\right)\equiv -\langle \eta (t)\rangle \equiv -\sum_{k=1}^{W(\bar{Z})}p_{k}\left(t\right)\ln p_{k}\left(t\right),$$
where we have also shown an explicit time-dependence, which merely reflects the fact that it is not a state function in $S_{\bar{Z}}$, a reflection of the fact that M is not uniquely specified in $S_{\bar{Z}}$. We have put *t* back above for clarity. Thus, Equation (Equation 47) is nothing but Equation (Equation 14) in form, and thus justifies it for an arbitrary M.

The above derivation is based on fundamental principles and does not require the body to be in equilibrium; therefore, it is always applicable for any arbitrary macrostate M(t). To the best of our knowledge, even though such an expression has been extensively used in the literature for NEQ entropy, it has been used by simply appealing to the information entropy [57].

The distinction between the Gibbs’ statistical entropy S and the thermodynamic entropy *S* should be emphasized. The latter appears in the Gibbs fundamental relation that relates the energy change dE with the entropy change dS as is well known in classical thermodynamics, and as we will also demonstrate below; see also Equation ([Disp-formula FD23a-entropy-23-01584]). The concept of microstates is irrelevant for this, which is a purely thermodynamic relation. On the other hand, S is solely determined by $\left\{m_{k}\right\}$ so its a statistical quantity. It then becomes imperative to show their equivalence, mainly because S is based on the Boltzmann idea. This equivalence has been justified elsewhere [51,52], and will be briefly summarized below.

**Remark** **3.**
*Because of this equivalence, we will no longer make any distinction between the statistical Gibbs entropy and the thermodynamic entropy and will use the standard notation S for both of them.*


**Remark** **4.**
*The Gibbs entropy S appears as an instantaneous ensemble average, see Definition 12. This average should be contrasted with a temporal average in which a macroquantity φ is considered as the average over a long period $\tau_{0}$ of time*

$$\varphi =\frac{1}{\tau_{0}}\int_{0}^{\tau_{0}}\varphi \left(t\right)dt,$$

*where φ(t) is the value of φ at time t [14]. For an EQ macrostate $M_{\mathrm{eq}}$, both definitions give the same result provided ergodicity holds. The physics of this average is that φ(t) at t represents a microstate of $M_{\mathrm{eq}}$. As $M_{\mathrm{eq}}$ is invariant in time, these microstates belong to $M_{\mathrm{eq}}$, and the time average is the same as the ensemble average if ergodicity holds. However, for a NEQ macrostate M(t), which continuously changes with time, the temporal average is not physically meaningful as the microstate at time t corresponds to M(t) and not to M(t=0) in that the probabilities and Z are different in the two macrostates. Only the ensemble average makes any sense at any time t as was first pointed out in [58]. Because of this, we only consider ensemble averages in this review.*


The maximum possible value of S(t) for given $\bar{Z}\in S_{\bar{Z}}$ occurs when $m_{k}$ are *uniquely* specified in $S_{\bar{Z}}$. This makes S(t) a state function of $\bar{Z}\left(t\right)$ with no explicit time dependence, which we write as $S(\bar{Z})$. Thus,
(48)$${\left.S_{\max}\left(\bar{Z},t\right)\right|}_{\bar{Z}\;\mathrm{fixed}}=S\left(\bar{Z}\right).$$

The simplest way to understand the physical meaning is as follows: Consider $\bar{Z}\in S_{Z}$ at some time *t*. As S(t) may not be a unique function of $\bar{Z}$, we look at all possible entropy functions for this $\bar{Z}$. These entropies correspond to all possible sets of $\left\{p_{k}\left(t\right)\right\}$ for a fixed $\bar{Z}$, and define different possible macrostates $\left\{M\right\}$. We pick that particular $\bar{M}\in \left\{M\right\}$ among these that has the *maximum possible value* of the entropy, which we denote by $S(\bar{Z})$ or S(Z(t)) without any explicit *t*-dependence. This entropy is a *state function* $S(\bar{Z})$. For a macroscopic system, this occurs when the corresponding microstate probabilities for $\bar{M}$ are
(49a)$${\bar{p}}_{k}\left(t\right)=1/W\left(\bar{Z}\right)>0,\;\;\;\;\forall {\bar{m}}_{k}\in \Gamma \left(\bar{Z}\right),$$
so that
(49b)$$S\left(\bar{Z}\right)=\ln W\left(\bar{Z}\right).$$

We wish to point out the presence of nonzero probabilities in Equation ([Disp-formula FD49a-entropy-23-01584]) that explains the comment above of available microstates. Including microstates with zero probabilities will not correcting account for the number of microstates with given $\bar{Z}$.

There is an alternative to the above picture in which we can imagine the Σ for which $\bar{Z}$ has been fixed, which essentially "isolates" Σ and converts it into a $\Sigma_{0}$. Then, as *t* varies, its entropy increases until it reaches its maximum value $S(\bar{Z})$ in accordance with Proposition 1.

**Remark** **5.**
*We emphasize that $\bar{Z}=\left(E,W\right)$ so $p_{k}$ above in Equation ([Disp-formula FD49a-entropy-23-01584]) is determined by the average energy E and not by the microstate energy $E_{k}$ as derived later in Section (Section 8.2). The $p_{k}$ in Equation ([Disp-formula FD49a-entropy-23-01584]) basically replaces the actual probability distribution in Equation (Equation 101) by a flat distribution of height $1/W(\bar{Z})$ and width $W(\bar{Z})$, a common practice in the thermodynamic limit of statistical mechanics [14]. Despite this modification, the entropy has the same value for a macroscopic system, for which β and $F_{w}$ are given by Equations (Equation 72) and (Equation 73), respectively; see also Section 8.2.*


Let us consider a different formulation of the entropy for a macrostate $\bar{M}\left(t\right)\in$$S_{\bar{X}}$ specified by some $\bar{X}=X\left(t\right)\subset Z$ at some instance *t*. This macrostate provides a more incomplete specification than in $S_{\bar{Z}}$. Applying the above formulation to $\bar{M}\in$$S_{\bar{X}}$, and consisting of microstates $\left\{{\bar{m}}_{k}\right\},$ forming the set $\bar{m}\equiv m\left(\bar{X}\right),$ with probabilities $\left\{{\bar{p}}_{k}\left(t\right)\right\}$, we find that
(50)$$S\left(\bar{X},t\right)\equiv -\sum_{k=1}^{W(\bar{X})}{\bar{p}}_{k}\left(t\right)\ln{\bar{p}}_{k}\left(t\right),\sum_{k=1}^{W(\bar{X})}{\bar{p}}_{k}\left(t\right)\equiv 1,$$
is the entropy of $\bar{M}$; here $W(\bar{X})$ is the number of distinct microstates ${\bar{m}}_{k}$. It should be obvious that
$$W\left(\bar{X}\right)\equiv \sum_{\xi (t)}W\left(\bar{Z}\right).$$

Again, under the equiprobable assumption
$${\bar{p}}_{k}\left(t\right)\to {\bar{p}}_{k,\mathrm{eq}}=1/W\left(\bar{X}\right),\;\;\;\;\forall {\bar{m}}_{k}\in \Gamma \left(\bar{X}\right),$$

$\Gamma (\bar{X})$ denoting the sample space spanned by $\bar{m}=\left\{{\bar{m}}_{k}\right\}$, the above entropy takes its maximum possible value
(51)$$S_{\max}\left(\bar{X},t\right)=S\left(\bar{X}\right)=\ln W\left(\bar{X}\right),$$
which is the well-known value of the Boltzmann entropy for a body in equilibrium
(52)$$S\left(\bar{X}\right)=\ln W\left(\bar{X}\right),$$
and provides a statistical definition of, and hence connects it with the, thermodynamic entropy of the body proposed by Boltzmann. The maximization again has the same implication as in Equation (Equation 48): For given $\bar{X}$, we look for the maximum entropy at all possible times. It is evident that
(53)$$S\left(\bar{Z},t\right)\leq S\left(\bar{Z}\right)\leq S\left(\bar{X}\right).$$

Thus, the NEQ entropy $S(\bar{Z},t)$ as $t\to \tau_{\mathrm{eq}}$, the equilibration time, reduces to $S(\bar{X})$ in EQ, as expected. Before equilibration, $S(\bar{Z})$ in $S_{\bar{Z}}$ remains a nonstate function $S(\bar{X},t)$ in $S_{\bar{X}}$ where we do not invoke ξ. It is the variation in ξ that is responsible for the time variation in $S(\bar{X},t)$. A simple proof of this conclusion is given in Section 8.3; see Remark 15 also. We can summarize this conclusion as

**Conclusion** **1.**
*The variation in time in $S(\bar{X},t)$ in $S_{\bar{X}}$ is due to the missing set of internal variables ξ.*


We now revert back to the standard use of X, and Z. Let us consider a body Σ, which we take to be isolated and out of equilibrium so that its macrostate M spontaneously relaxes towards $M_{\mathrm{eq}}$ at fixed X. Its entropy S(X,t) in $S_{X}$ has an explicit time dependence, which continue to increase towards S(X). For such NEQ states, the explicit time dependence in S(X,t) is explained by introducing ξ to make their entropies a state function in an appropriately chosen larger state space $S_{Z}$ [41]. It is also shown there that a NIEQ macrostate with a nonstate function entropy S(Z,t) may be converted to an IEQ macrostate with a state function entropy $S(Z^{\prime })$ by going to an appropriately chosen larger state space $S_{Z^{\prime }}$ spanned by $Z^{\prime }$ with $S_{Z}$ its proper subspace. Therefore, in most cases of interest here, we would be dealing with a state function and usually write it as S(Z), unless a choice for Z has been made based on the experimental setup. In that case, we must deal with a pre-determined state space $S_{Z}$ so that some NEQ states that lie outside $S_{Z}$ can become a state function in some $S_{Z^{\prime }}⊃S_{Z}$.

We have discussed above that the explicit time dependence in a NEQ macrostate with a nonstate function entropy $S_{\mathrm{neq}}\left(t\right)≐S\left(X,t\right)$ is due to additional state variables in ξ and that this NEQ macrostate may be converted into an IEQ macrostate with a macrostate function entropy $S_{\mathrm{ieq}}\left(Z\right)$ by going from $S_{X}$ to an appropriately chosen larger state space $S_{Z}$. Similarly, it has been shown [41] that a NIEQ macrostate $M_{\mathrm{nieq}}$ in $S_{Z}$ with a nonstate function entropy $S_{\mathrm{nieq}}\left(t\right)≐S\left(Z,t\right)$ may be converted to an IEQ macrostate $M_{\mathrm{ieq}}^{\prime }$ in an appropriately chosen larger state space $S_{Z^{\prime }}$ with a state function entropy $S_{\mathrm{ieq}}\left(Z^{\prime }\right)$.The additional internal variables $\xi^{\prime }$ that are over and above ξ in $Z^{\prime }$ give rise to additional entropy generation as they relax for fixed Z. This results in the following inequality:(54)$$S_{\mathrm{ieq}}\left(Z\right)\geq S_{\mathrm{ieq}}\left(Z^{\prime }\right)=S_{\mathrm{nieq}}\left(Z,t\right).$$

However, if the choice for Z has been made based on the experimental setup and the observation time $\tau_{\mathrm{obs}}$, see Section 8.1, we must restrict our discussion to $S_{Z}$ so that we must consider $M_{\mathrm{nieq}}$ in $S_{Z}$ the following. This will be done in Section 8.3; see Remarks 13 and 15.

### 4.3. A Proof of the Second Law

The second law has been proven so far under different assumptions ([53,58,59,60,61], (among others)). Here, we provide a simple proof of it based on the postulate of the flat distribution; see Remark 5. The current proof is an extension of the proof given earlier see also ([53] (Theorem 4)). We consider an isolated system $\Sigma_{0}$ for which the second law is expressed by Equation (Equation 42). However, for simplicity, we will suppress the subscript 0 from all the quantities in this section. As the law requires considering the instantaneous entropy as a function of time, we need to focus on the sample space at each instant to determine its entropy *S* as a function of time. At each instance, it is an ensemble average over the instantaneous sample space Γ(t) formed by the instantaneous set m(t) of available microstates, see Equations (Equation 14) or (Equation 47). We will use the flat distributions for the microstates at each instance, see Remark 5, so that the entropy is given by Equation ([Disp-formula FD49b-entropy-23-01584]).

To prove the second law, see Proposition 1, we proceed in steps by considering a sequence of sample spaces belonging to Γ as follows [53,58]. At a given instant, a system happens to be in some microstate. We start at $t=t_{1}=0$, at which time Σ happens to be in a microstate, which we label $m_{1}$. It forms a sample space $\Gamma_{1}$ containing $m_{1}$ with probability $p_{1}^{\left(1\right)}=1$, with the superscript denoting the sample space. We have $S^{\left(1\right)}=0$. At some $t=t_{2}$, the sample space is enlarged from $\Gamma_{1}$ to $\Gamma_{2}$, which contains $m_{1}$ and $m_{2}$, with probabilities $p_{1}^{\left(2\right)}$ and $p_{2}^{\left(2\right)}$. Using the flat distribution, the entropy is now $S_{2}=\ln2$. We just follow the system in a sequence of time so that at $t=t_{n}$, we have a sample space $\Gamma_{n}$ with $m_{1},m_{2},\cdots ,m_{n}$ so that $S_{n}=\ln n$. Continuing this until all microstates in Γ have appeared, we have $S_{\max}=\ln W$.

Thus, we have proven that the entropy continues to increase until it reaches its maximum in accordance with Proposition 1.

## 5. Hamiltonian Trajectories in $S_{Z}$

### 5.1. Generalized Microforce and Microwork for Σ

Traditional formulation of statistical thermodynamics [14,19,53] takes a mechanical approach in which $m_{k}$ follows its classical or quantum mechanical evolution dictated by its SI-Hamiltonian $H(\left.x\right|W)$. The quantum microstates are specified by a set of good quantum numbers, which we have denoted by *k* above as a single quantum number for simplicity; we take k∈N,N denoting the set of natural numbers. We will see below that *k* does not change as W changes. In the classical case, we use a small cell $\delta x_{k}$ around $x_{k}=x$ as discussed above as the microstate $m_{k}$. The Hamiltonian gives rise to a purely mechanical evolution of individual $m_{k}$’s, which we will call the *Hamiltonian evolution*, and suffices to provide their mechanical description. The change in $H(\left.x\right|W)$ in a process is
(55a)$$dH=\frac{\partial H}{\partial x}\cdot dx+\frac{\partial H}{\partial W}\cdot dW.$$

The first term on the right vanishes identically due to Hamilton’s equations of motion for any $m_{k}$. Thus, for fixed W, the energy $E_{k}=H_{k}≐H\left(\left.x_{k}\right|W\right)$ remains constant as $m_{k}$ moves about in Γ(W). Only the variation dW in $S_{Z}$ generates any change in $E_{k}$. Consequently, we do not worry about how $x_{k}$ changes in $H(\left.x\right|W)$ in the phase space, and focus, instead, on the state space $S_{Z}$, in which can write
(55b)$$dE_{k}=\frac{\partial E_{k}}{\partial W}\cdot dW=-dW_{k},$$
where $dW_{k}$ denotes the *generalized microwork* produced by the *generalized microforce* $F_{wk}$:
(55c)$$dW_{k}=F_{wk}\cdot dW,\;F_{wk}≐-\partial E_{k}/\partial W.$$

For the case W=(V,ξ), the corresponding microforce $F_{wk}$ is ($P_{k},A_{k})$, where
(56)$$P_{k}=-\partial E_{k}/\partial V,A_{k}=-\partial E_{k}/\partial \xi .$$

The corresponding microwork is
(57)$$dW_{k}=P_{k}dV+A_{k}d\xi .$$

### 5.2. Statistical Significance of dW and dQ

Before proceeding further, let us see how the generalized macrowork and macroheat could be understood from a statistical point of view so that we can identify them using the Hamiltonian. Once W has been identified, the Hamiltonian must be expressed in terms of it. Thus, $m_{k}$ and $E_{k}$ are functions of W in $S_{Z}$. We now prove

**Theorem** **2.**
*E(t) is a function of W(t) and S(t) for any $M_{\mathrm{arb}}$, even though $E_{k}\left[W\left(t\right)\right]$’s are functions of W(t) only.*


**Proof.** We consider the differential
(58)$$dE\left(t\right)\equiv \sum_{k}p_{k}\left(t\right)dE_{k}\left(t\right)+\sum_{k}E_{k}\left(t\right)dp_{k}\left(t\right).$$As $p_{k}\left(t\right)$’s are unchanged in the first sum, this sum is evaluated at *constant entropy* so this is purely mechanical macroquantity $dE_{m}$; see Equation (Equation 20). This sum is a function of W(t) as is seen clearly in Equation ([Disp-formula FD55b-entropy-23-01584]). The second contribution is at fixed microstate energies $E_{k}$ so W(t) is held fixed, but require changes in the probabilities so it is the stochastic contribution $dE_{s}$, see Equation ([Disp-formula FD55b-entropy-23-01584]). The changes $\left\{dp_{k}\left(t\right)\right\}$ result in is dS. As $dE_{s}$ and
(59)$$dS=-\sum_{k}\left(\eta_{k}\left(t\right)+1\right)dp_{k}\left(t\right)=-\sum_{k}\eta_{k}\left(t\right)dp_{k}\left(t\right)$$
are both extensive, they must be linearly related with an intensive constant of proportionality. This proves that E(t) is a function of S(t) and W(t) in general for any $M_{\mathrm{arb}}$. □

Note that we have used the identity $\sum_{k}dp_{k}=0$ above; see also Equation (Equation 108).

We introduce a special process, to be called a generalized *isometric* process, which is a process at fixed W(t) and is a generalization of an *isochoric* process. In this process, the work done by each mechanical variables in W(t) remains zero so $dE_{m}\equiv 0$. We now prove the following theorem that establishes the physical significance of the two contributions.

**Theorem** **3.**
*The isentropic contribution represents the generalized macrowork dW(t) and the stochastic contribution represents the generalized macroheat dQ(t) for any $M_{\mathrm{arb}}$.*


**Proof.** We follow Landau and Lifshitz [14] and rewrite the first term in Equation (Equation 58) as
$$\begin{array}{cc} dE_{m}\left(t\right) & \equiv \sum_{k}p_{k}\left(t\right)\frac{\partial E_{k}}{\partial W}\cdot dW\left(t\right)\\ \; & =-\sum_{k}p_{k}\left(t\right)F_{wk}\left(t\right)\cdot dW\left(t\right) \end{array}$$
where we have used Equation ([Disp-formula FD55c-entropy-23-01584]). The use of Equations ([Disp-formula FD64a-entropy-23-01584]) and ([Disp-formula FD64b-entropy-23-01584]) proves that
(60)$$dW\left(t\right)\equiv -dE_{m}\left(t\right)$$
is the isentropic contribution, making macrowork a mechanical concept as we have already pointed out. This identification then also proves that the macroheat in the first law, see Equation ([Disp-formula FD23a-entropy-23-01584]), must be properly identified with dQ(t). Accordingly,
(61)$$dQ\left(t\right)\equiv dE_{s}\left(t\right)\equiv \sum_{k}E_{k}\left(t\right)dp_{k}\left(t\right),$$
is purely stochastic. □

The linear proportionality between $dQ=dE_{s}$ and dS mentioned above in the proof of Theorem 2 results in
(62)$$dQ\left(t\right)/dS\left(t\right)=T_{\mathrm{arb}}\left(t\right),$$
which is a statistical proof of the identity in Equation ([Disp-formula FD5a-entropy-23-01584]) relating dQ(t) and dS(t) for any $M_{\mathrm{arb}}$. We also note that the ratio $T_{\mathrm{arb}}\left(t\right)$ is related to the ratio of two SI-macroquantities. Thus, it can be used to characterize the instantaneous macrostate $M_{\mathrm{arb}}$. This should be contrasted with the $\overset{{}^{\circ}}{M}$NEQT, in which the ratio
(63)$$d_{e}Q\left(t\right)/d_{e}S\left(t\right)=T_{0}$$
does not characterize the instantaneous macrostate $M_{\mathrm{arb}}$. In Equation (Equation 104), we provide a general procedure for a thermodynamic identification of $T_{\mathrm{arb}}$.

**Remark** **6.**
*It is worth emphasizing that dQ(t) and dS(t) in Equation (Equation 61)–(Equation 59) are defined as instantaneous quantities in terms of the instantaneous changes $\left\{dp_{k}\left(t\right)\right\}$, regardless of the speed of the segmental process $dP_{\mathrm{arb}}≐dP\left(t\right)$, and instantaneous values $\left\{E_{k}\left(t\right)\right\}$ and $\left\{p_{k}\left(t\right)\right\}$. Therefore, the generalized macroheat and entropy change are defined regardless of the speed of the arbitrary process. As dE(t) in Equation (Equation 58) is also defined instantaneously, it is clear from Equation ([Disp-formula FD23a-entropy-23-01584]) that the generalized work dW is also defined instantaneously regardless of the speed of the arbitrary process. This is consistent with our above derivation of dW in terms of generalized forces. The observation is very important as it shows that the existence of all SI-quantities does not depend on the speed of the arbitrary process $dP_{\mathrm{arb}}$. However, see also Section 8.1 further clarification on the importance of $\tau_{\mathrm{obs}}$. From now onward, we will not make a distinction between T and $T_{\mathrm{arb}}$.*


We should point out that, as W(t) is a parameter, dW(t) is the same for all microstates. The statistical nature of $dE_{m}$ is reflected in the statistical nature of $F_{w}\left(t\right)$,such as P(t) and A(t), of the system. Thus, the SI-fields $F_{wk}\left(t\right)$ are *fluctuating* quantities from microstate to microstate as expected in any averaging process.

We can now identify W as the *macrowork parameter*, and the variation dZ(t)≐(dE(t),dW(t)) in $S_{Z}$ defines not only the microwork $\left\{dW_{k}\right\}$, but also a thermodynamic process P. The trajectory $\gamma_{k}$ in $S_{Z}$ followed by $m_{k}$ as a function of time will be called the *Hamiltonian trajectory* during which W varies from its initial (in) value $W_{\mathrm{in}}$ to its final (fin) value $W_{\mathrm{fin}}$ during P, the the path $\gamma_{P}$ denotes the path the macrostate follows during this process; see Definition 10. The variation produces the generalized microwork $dW_{k}$. As $p_{k}$ plays no role in $dW_{k}$, its determination is simplifies in the MNEQT. The microwork $dW_{k}$ also does not change the index *k* of $m_{k}$ as said above. The ensemble average of $F_{wk}$ is $F_{w}$, see Equation (Equation 28),
(64a)$$F_{w}=\langle F_{w}\rangle ≐\sum_{k}p_{k}F_{wk};$$
that of $dW_{k}$ is dW given by
(64b)$$dW=\langle dW\rangle ≐\sum_{k}p_{k}dW_{k}≐F_{w}\cdot dW,$$
as given earlier in Equation (Equation 27). It is based on using the mechanical definition (force X displacement) of work. The macroforce corresponding to W=(V,ξ) is $F_{w}=\left(P,A\right)$, where $P=\langle P\rangle$, and $A=\langle A\rangle$. The corresponding SI-macrowork is given earlier in Equation ([Disp-formula FD32a-entropy-23-01584]).

The above discussion proves that the definition of macroheat and macrowork is valid for any $M_{\mathrm{arb}}$. It is useful to compare the above approach with the traditional formulation of the first law in terms of $d_{e}Q\left(t\right)$ and $d_{e}W\left(t\right)$: *both formulations are valid in all cases*. It should be mentioned that the above identification is well known in equilibrium statistical mechanics, but its extension to irreversible processes and our interpretation is, to the best of our knowledge, novel. While the instantaneous average $F_{w}\left(t\right)$ such as the pressure P(t) is mechanically defined under all circumstances, it will only be identified with the thermodynamic definition of the instantaneous pressure
(65)$$P\left(t\right)=-{\left(\partial E/\partial V\right)}_{S,\xi }$$
for a uniquely identified macrostate in $S_{Z}$.

Being purely mechanical in nature, a trajectory is completely *deterministic* and cannot describe the evolution of a macrostate M during P unless supplemented by thermodynamic stochasticity, which requires $p_{k}\left(M\right)$ as discussed above [14]. Thermodynamics emerges when quantities pertaining to the trajectories are averaged over the trajectory ensemble $\left\{\gamma_{k}\right\}$ with appropriate probabilities that will usually change during the process.

**Conclusion** **4.**
*The change dE consists of two independent contributions- an isentropic change $dE_{m}=-dW$, and an stochastic change $dE_{s}=T_{\mathrm{arb}}dS$. On the other hand, the MI-macroheat and the MI-macrowork suffer from ambiguity; see, for example, Kestin [12].*


**Remark** **7.**
*It is clear from the above discussion that it is the macroheat and not the macrowork that causes $p_{k}\left(t\right)$, and therefore the entropy to change. This is the essence of the common wisdom that heat is random motion. But we now have a mathematical definition: macroheat is the isometric part of dE(t) that is directly related to the change in the entropy through changes in $p_{k}\left(t\right)$. Macrowork is that part of the energy change caused by isentropic variations in the "mechanical" state variables W(t). This is true no matter how far the system is from equilibrium. Thus, our formulation of the first law and the identification of the two terms is the most general one, and applicable to any $M_{\mathrm{arb}}$.*


**Remark** **8.**
*The relationship between the macroheat and the entropy becomes simple only when M happens to be in internal equilibrium, see Section 6.1, in which case $T_{\mathrm{arb}}\left(t\right)$ is replaced by T(t), which has a thermodynamic significance; see Equation (Equation 72) and we have the thermodynamic identity, called the Clausius Equality in Equation ([Disp-formula FD5a-entropy-23-01584]) dQ(t)=T(t)dS(t) for $M_{\mathrm{ieq}}$, which is very interesting in that it turns the well-known Clausius inequality $d_{e}Q=T_{0}d_{e}S\leq$$T_{0}dS$ into an equality.*


For the sake of completeness, we briefly discuss the various attempts to the study of the microanalogs $dW_{k}$ and $dQ_{k}$ of the dW and dQ, respectively, that has flourished into an active field in diverse branches of NEQT at diverse length scales from mesoscopic to macroscopic lengths [33,34,36,37,38,62,63,64,65,66]; see also some recent reviews [67,68,69]. Unfortunately, this endeavor is apparently far from complete [12,33,34,36,37,62,63,64,65,66,67,68,69,70,71,72,73,74,75,76,77,78,79,80,81,82,83,84,85,86,87,88,89,90]. This is because of the confusion about the meaning of macrowork and macroheat even in classical NEQT [9,12] involving SI- or MI- description, which has only recently been clarified [32,41,51,52,55,91,92] in the MNEQT, where a clear distinction is made between the generalized macrowork (macroheat) dW (dQ) and the exchange macrowork (macroheat) $d_{e}W$ ($d_{e}Q$). In an EQ process, both macroworks (macroheats) have the same magnitude, but not in a NEQ process, where the difference determines $d_{i}W\geq 0$ ($d_{i}Q\geq 0$).

It is important to draw attention to the following important fact. We first recognize that the first law in Equation ([Disp-formula FD23b-entropy-23-01584]) refers to the change in the energy caused by exchange quantities. Therefore, dE on the left truly represents $d_{e}E$. Accordingly, we write Equation ([Disp-formula FD23b-entropy-23-01584]) as
(66)$$d_{e}E=d_{e}Q-d_{e}W,$$
which justifies Equation (Equation 21) for $d_{e}$. Subtracting this equation from Equation ([Disp-formula FD23a-entropy-23-01584]), we obtain the identity
(67)$$d_{i}E=d_{i}Q-d_{i}W\equiv 0,$$
which not only justifies Equation (Equation 21) for $d_{i}$ but also Equation (Equation 22) for which we have used Equation (Equation 9).

**Remark** **9.**
*The above analysis demonstrates the important fact that the first law can be applied either to the exchange process ($d_{e}$) or to the interior process ($d_{i}$). The last formulation is also applicable to an isolated system.*


### 5.3. Medium $\tilde{\Sigma }$

The above discussion can be easily extended to the medium (the suffix $\tilde{k}$ denotes its microstates) with the following results
(68)$$\begin{array}{cc} d\tilde{W}\left(t\right) & =-d{\tilde{E}}_{m}\equiv -\sum_{\tilde{k}}{\tilde{p}}_{\tilde{k}}\frac{\partial {\tilde{E}}_{\tilde{k}}}{\partial \tilde{w}}\cdot d\tilde{w}\\ \; & =f_{w0}\cdot d\tilde{w}=-d_{e}W,\\ d\tilde{Q}\left(t\right) & =d{\tilde{E}}_{s}\equiv \sum_{\tilde{k}}{\tilde{E}}_{\tilde{k}}d{\tilde{p}}_{\tilde{k}}=-d_{e}Q, \end{array}$$
where all the quantities including $\tilde{k}$ refer to the medium, except $d_{e}W$ and $d_{e}Q$, and have their standard meaning. The analog of Equation (Equation 62) is $d\tilde{Q/}d\tilde{S}=T_{0}$ as expected; see Equation (Equation 63). We clearly see that
(69a)$$dW_{0}≐dW+d\tilde{W}=d_{i}W\geq 0$$
such as when mechanical equilibrium is not present. In this case, we also have
(69b)$$dQ_{0}≐dQ+d\tilde{Q}=d_{i}Q\geq 0,$$
with $dW_{0}=dQ_{0}$ in view of Equation (Equation 22). In a finite process P, all infinitesimal quantities are replaced by their net changes
(69c)$$\Delta W_{0}≐\Delta W+\Delta \tilde{W}=\Delta Q_{0}=\Delta_{i}W\geq 0,$$
where $\Delta_{i}W$ is obtained by integrating $d_{i}W$ in Equation ([Disp-formula FD77c-entropy-23-01584]) over P; the result is given in Equation (Equation 82), where it is discussed.

### 5.4. Irreversible Macrowork and Macroheat

We can now identify $d_{i}W\left(t\right)$ and $d_{i}Q\left(t\right):$(70)$$d_{i}W\left(t\right)\equiv -\left(dE_{m}+d{\tilde{E}}_{m}\right),d_{i}Q\left(t\right)\equiv \left(dE_{s}+d{\tilde{E}}_{s}\right),$$
satisfying Equation (Equation 22), which follows from $d_{i}E=0$; see Equation (Equation 9). It is easy to see that $d_{i}W\left(t\right)$ reproduces Equation (Equation 31), where we must use $d\tilde{w}=-d_{e}w$, and $dw=d_{e}w+d_{i}w$.

## 6. Unique Macrostates

### 6.1. Internal Equilibrium

We now revert back to the original notation X and Z. We will refer to S(Z(t)) in terms of microstate number W(Z(t)) in Equation ([Disp-formula FD49b-entropy-23-01584]) as the *time-dependent Boltzmann formulation* of the entropy or simply the Boltzmann entropy [65], whereas S(X) in Equation (Equation 51) represents the equilibrium (Boltzmann) entropy. It is evident that the Gibbs formulation in Equations (Equation 47) and (Equation 50) supersedes the Boltzmann formulation in Equations (Equation 48) and (Equation 51), respectively, as the former contains the latter as a special limit. However, it should be also noted that there are competing views on which entropy is more general [65,93]. We believe that the above derivation, being general, makes the Gibbs formulation more fundamental. The continuity of S(Z,t) follows directly from the continuity of $p_{k}\left(t\right)$. Its existence follows from the observation that it is bounded above by lnW(Z) and bounded below by 0, see Equation ([Disp-formula FD49b-entropy-23-01584]).

We now introduce the central concept of the MNEQT, which is based on the existence of S(Z) above; see Definition 5, which we now expand.

**Definition** **16.**
*A NEQ macrostate M whose entropy is a state function S(Z) in $S_{Z}$ is said to be an internal equilibrium (IEQ) macrostate $M_{\mathrm{ieq}}$ [41,55]; if not, its entropy S(Z,t) is an explicit function of time t in $S_{Z}$. An IEQ-macrostate in $S_{Z}$ is a unique macrostate in $S_{Z}$.*


We clarify this point. If we do not use ξ for M, which is not unique in $S_{X}$, then its entropy cannot be a state function in $S_{X}$, and must be expressed as S(X,t). Thus, the importance of ξ is to be able to deal with a state function entropy S(Z) by choosing an *appropriate* number of internal variables. Throughout this work, we will only deal with IEQ macrostates. However, as we will see, our discussion of NEQ macrowork will cover all states.

Being a state function, S(Z) shares many of the properties of EQ entropy S(X), see Definition 5:(1)*Maximum*: S(Z) is the maximum possible value of the NEQ entropy in $S_{Z}$ for a given Z [41].(2)*No memory* -Its value also does not depend on how the system arrives in $M_{\mathrm{ieq}}\equiv M\left(Z\right)$, i.e., whether it arrives there from another IEQ macrostate or a non-IEQ macrostate [41]. Thus, it has no memory of the earlier macrostate.

There are some macrostates that emerge in fast changing processes such as the free expansion that possess memory of the initial states so that their entropy will no longer be a state function in $S_{X}$. In this case, we need to enlarge the state space to $S_{Z}$ by including internal variables as done in Section 14.

**Remark** **10.**
*It may appear to a reader that the concept of entropy being a state function is very restrictive. This is not the case as this concept, although not recognized by several workers, is implicit in the literature where the relationship of the thermodynamic entropy with state variables is investigated. To appreciate this, we observe that the entropy of a body in internal equilibrium [41,55] is given by the Boltzmann formula in Equation ([Disp-formula FD49b-entropy-23-01584]) in terms of the number of microstates corresponding to Z(t). In classical nonequilibrium thermodynamics [3], the entropy is always taken to be a state function. In the Edwards approach [94] for granular materials, all microstates are equally probable as is required for the above Boltzmann formula. Bouchbinder and Langer [42,43] assume that the nonequilibrium entropy is given by Equation ([Disp-formula FD49b-entropy-23-01584]). Lebowitz [65] also takes the above formulation for his definition of the nonequilibrium entropy. As a matter of fact, we are not aware of any work dealing with entropy computation that does not assume the nonequilibrium entropy to be a state function. This does not, of course, mean that all states of a system are internal equilibrium states. For states that are not in internal equilibrium, the entropy is not a state function so that it will have an explicit time dependence. But, as shown elsewhere, Ref. [41] this can be avoided by enlarging the space of internal variables. The choice of how many internal variables are needed will depend on experimental time scales and cannot be answered in generality just as is the case in EQ thermodynamics for the number of observables; the latter depends on the experimental setup. A detailed discussion is offered elsewhere [45].*


### 6.2. Gibbs Fundamental Relation

Being a state function, S(Z) in $S_{Z}$ for $M_{\mathrm{ieq}}$ results in the following Gibbs fundamental relation for the entropy
(71a)$$dS=\frac{\partial S}{\partial Z}\cdot dZ=\frac{\partial S}{\partial E}dE+\frac{\partial S}{\partial W}\cdot dW,$$
which can be inverted to express the Gibbs fundamental relation for the energy as
(71b)$$dE=TdS-F_{w}\cdot dW,$$
where we have introduced
(72)$$\begin{array}{cc} \beta & =1/T≐\partial S/\partial E=1/(\partial E/\partial S), \end{array}$$
(73)$$\begin{array}{cc} F_{w} & ≐T\partial S/\partial W=-\partial E/\partial W \end{array}$$
as the inverse temperature of the system (we set the Boltzmann constant $k_{B}=1$ throughout the review), and have used Equation (Equation 28) for the generalized macroforce $F_{w}$. Recalling Equation ([Disp-formula FD64b-entropy-23-01584]), we see that the second term in Equation ([Disp-formula FD71b-entropy-23-01584]) is nothing but the SI-macrowork dW. Comparing Equation ([Disp-formula FD71b-entropy-23-01584]) with Equation ([Disp-formula FD23a-entropy-23-01584]), we can identify the generalized macroheat dQ with TdS, which then proves Equation ([Disp-formula FD5a-entropy-23-01584]).

It should be stated here that the choice and the number of state variables included in X or Z is not so trivial and must be determined by the nature of the experiments [40]. *We will simply assume here that they have been specified*. Just as S=S(X) is a state function of X for $M_{\mathrm{eq}}$ in $S_{X}$, there are $M_{\mathrm{ieq}}$ in $S_{Z}$ for which S(Z) is a state function of Z.

The possibility of a Gibbs fundamental relation for $M_{\mathrm{nieq}}$ is deferred to Section 8.3.

### 6.3. A Digression on the NEQ-Temperature

While the concept of the macrowork is quite familiar from mechanics, the concept of the macroheat is peculiar to thermodynamics in view of Equation ([Disp-formula FD5a-entropy-23-01584]). In EQ thermodynamics, the macroheat dQ is directly proportional to the change dS, and the constant of proportionality determines the EQ temperature *T*. Indeed, the concepts of entropy and of temperature are unique to thermodynamics and are well established in EQ thermodynamics. A Σ in thermal equilibrium with a $\tilde{\Sigma }$ at $T_{0}$ obviously has the same temperature $T_{0}$. The temperature for an isolated system in equilibrium is also well defined; its inverse is identified with the energy derivative of the *equilibrium* entropy [14]. The definition is valid for *all* EQ systems, even those containing gravitational interaction. This is confirmed by the fact that Bekenstein used it to identify the temperature of an isolated black hole [95,96]. The formulation is valid both classically and quantum mechanically [14].

The EQ definition of the temperature is formally identical to that in Equation (Equation 72), which is valid in NEQT [41,51,52,53,55,91,92]. In this, we have a general thermodynamic definition of a temperature for any M. It is important to realize that the notion of a NEQ temperature is an absolute necessity for the *Clausius statement* of the second law that the exchanged macroheat flows spontaneously from hot to cold to be meaningful.

It is clear from the above discussion that macrowork is the isentropic change in the energy, while macroheat is the energy change due to the entropy change. This is not as surprising a statement as it appears, since a mechanical system is usually thought of as a system for which the entropy concept is not meaningful. A different way to state this is that the entropy remains constant (isentropic) in any mechanical process as we have done above. Planck [97] had already suggested that the temperature should be defined for NEQ macrostates just as the entropy should be defined for them if we need to carry out a thermodynamic investigation of a NEQ system. Such a temperature was apparently first introduced by Landau [98] for partial set of the degrees of freedom (dof). This then allows the possibility that the notion of temperature can be separately applied, for example, to vibrational and configurational dof in glasses that are known to be out of equilibrium with each other [46] in that they are ascribed different temperatures. This means that macroheat would be exchanged between them until they come to equilibrium, but this is internally exchanged. But there seems to be a lot of confusion about the meaning of the entropy and temperature in NEQT ([8,24,40,93,99,100,101,102,103,104,105] (for example)), where different definitions lead to different results. In contrast, the meaning of entropy and fields in equilibrium thermodynamics has no such problem.

We agree with Planck and believe that there must exist a unifying approach to identify the temperature for $M_{\mathrm{arb}}$; see Definition 6, with or without memory effects in $S_{Z}$. The inverse temperature defined above in Equation (Equation 72) is not directly applicable to nonIEQ-macrostates in $S_{Z}$ for which *S* is not a state function, but can be extended to them so as to accommodate memory effects as we do in Section 8.3. However, we will not consider them in detail in this review.

**Criterion** **5.**
*The identification of temperature in $M_{\mathrm{arb}}$ must satisfy some stringent but obvious criteria:*


C1It must be intensive and must reduce to the temperature determined by Equation (Equation 72) for $M_{\mathrm{eq}}$ and $M_{\mathrm{ieq}}$ even for an isolated system.C2It must cover negative temperatures [106] that are commonly observed for some dof such as nuclear spins in a system. As these dof are not involved in any macroscopic motion ([14] (Section 73)), there is no kinetic energy involved. Most common occurrence of a negative temperature is when the above spin dof are out of equilibrium with the other dof such as lattice vibrations in the system.C3It must satisfy the Clausius statement that macroheat between two objects always flows spontaneously from hot to cold for positive temperatures. When negative temperatures are considered, macroheat must flow from a system at a negative temperature to a system at a positive temperature.C4It must be a global rather than a local property of the system so that we can differentiate hot and cold between two different systems.

The first criterion ensures that the new temperature is an extension of the conventional notion of the temperature that is valid when the entropy is a state function. This means that the new notion of temperature is valid for any arbitrary macrostate. In addition, it must exist even for an isolated system. The second criterion ensures that our formalism includes negative temperatures that may occur in a lattice system. The third criterion ensures compliance with the second law for interacting systems. This is a very important criterion, which *every notion of temperature must satisfy*. We will come back to this issue again in Section 6.5 where we prove it in the MNEQT. The last criterion ensures that the temperature is associated with the entire system, whether the system is homogeneous or not. This will be explained by direct calculations of inhomogeneous systems in Section 10. By extension, the concept of a NEQ temperature can be also applied to different dof of a system such as a glass under the assumption that they are weakly interacting in accordance with the approach taken by Landau [98]. This results in the Tool-Narayanaswamy relation derived in Section 11.

Before we close this discussion, we wish to point out major differences between the NEQ temperature *T* in the MNEQT and its other definitions. We first consider the $\overset{{}^{\circ}}{M}$NEQT. The most important theories belonging to this class are the classical local irreversible thermodynamics (LNEQT) [3], the rational thermodynamics (RNEQT) [107], and the extended irreversible thermodynamics (ENEQT) [104] as we had mentioned earlier. We refer the reader to [40,104] for excellent reviews on these theories that use local densities of energy *e* and entropy *s*. They are continuum theories, and can all be classified as continuum $\overset{{}^{\circ}}{M}$NEQT to be denoted by the CNEQT here. We consider them critically later in Section 15. They differ in the choice of their state spaces. Considering the local entropy and energy densities *s* and *e*, the inverse local temperature is defined as ∂s/∂e, and differs from the global temperature in the MNEQT.

(1)In the LNEQT, each local volume element is in EQ so the local temperature is the EQ temperature of the volume element, and differs from *T*, which is a global temperature.(2)In the RNEQT, the temperature is taken as a primitive quantity along with the entropy. Because of the memory effect, the temperature at any time depends on the entire history. Thus, it is a local analog of the global temperature of $M_{\mathrm{nieq}}$ in the MNEQT, but the latter is defined thermodynamically.(3)In the ENEQT, the fluxes are part of the state variables so the local temperature also depends on them. Assuming the total entropy to also depend on the fluxes ([20], (see Equation (5.66), for example)), one can identify the global analog of the temperature in the ENEQT. However, as fluxes are MI-quantities, this temperature cannot be compared with the SI-temperature in the MNEQT.

There is a recent attempt [108] to introduce another NEQ temperature by using fluctuation theorems to determine the entropy generation, which is then related to the Gouy-Stodola theorem derived later (see Equation ([Disp-formula FD84b-entropy-23-01584])). It is limited to a interacting Σ in a medium $\tilde{\Sigma }$ so does not apply to an isolated system. In addition, its validity is limited to the situation when the Gouy-Stodola theorem is valid as seen from the derivation of Equation ([Disp-formula FD84b-entropy-23-01584]).

### 6.4. Uniqueness of $S_{\mathrm{ieq}}\left(Z\right)$ and *T*

We now give an alternative demonstration of the uniqueness of the entropy of $M_{\mathrm{ieq}}$ in $S_{Z}$, which is based on the discussion of the internal variables in Section 3. Let us assume that we divide Σ into a finite number of nonoverlapping EQ subsystems $\left\{\Sigma_{i}\right\}$ such that $\cup_{i}\Sigma_{i}=\Sigma$. Without loss of generality, we assume that the subsystems are not in EQ with each other (their fields are not identical) so that Σ is in a NEQ macrostate. Let $\lambda_{\mathrm{corr}}^{\left(i\right)}$ denote the correlation length of $\Sigma_{i}$, and we define $\lambda_{\mathrm{corr}}=\max\left\{\lambda_{\mathrm{corr}}^{\left(i\right)}\right\}$ to denote the maximum correlation length determining quasi-independence required for entropy additivity as discussed in Section 2; see Equation (Equation 2). For this, we need to take the linear size $\Delta l_{i}≳\lambda_{\mathrm{corr}}$ of $\Sigma_{i}$. The EQ microstate $M_{\mathrm{eq}}^{\left(i\right)}$ of $\Sigma_{i}$ is uniquely described in $S_{X}$. The additivity of entropy gives $S_{\mathrm{ieq}}$ that must be a function of $\left\{X_{i}\right\}$. Moreover, since each $M_{\mathrm{eq}}^{\left(i\right)}$ has a unique entropy $S_{i}\left(X_{i}\right)$, $S_{\mathrm{ieq}}$ also has a unique value
(74a)$$S_{\mathrm{ieq}}\left(\left\{X_{i}\left(t\right)\right\}\right)=\sum_{i}S_{i}\left(X_{i}\left(t\right)\right).$$

As we need to express $S_{\mathrm{ieq}}$ in terms of $X\left(t\right)=\sum_{i}X_{i}\left(t\right)$, we need additional independent linear combinations $\xi \left(t\right)=\cup_{i}\xi_{i}\left(t\right)$ made from the set $\left\{X_{i}\right\}$ as already discussed in Section 3 to ensure that S(Z(t)) depends on the *same* number $n^{*}$ of state variables as there are in $S_{\mathrm{ieq}}\left(\left\{X_{i}\left(t\right)\right\}\right)$. This *uniquely* defines
(74b)$$S\left(Z\left(t\right)\right)=\sum_{i}S_{i}\left(X_{i}\left(t\right)\right)$$
in $S_{Z}$ in terms of the unique values$\;S_{i}\left(X_{i}\left(t\right)\right)$. It is a mathematical identity between the left side for $\Sigma_{B}$ and the right side for $\Sigma_{C}$. We can also take $\Sigma_{i}$’s to be in $M_{i\mathrm{ieq}}$’s in $S_{Z_{i}}$’s so that $S_{i}\left(Z_{i}\left(t\right)\right)$ are also uniquely defined. Then, the same reasoning as above also proves that
(74c)$$S\left(Z\left(t\right)\right)=\sum_{i}S_{i}\left(Z_{i}\left(t\right)\right)$$
is unique by ensuring that the number of arguments $n^{*}$ are the same on both sides between the entropies for $\Sigma_{B}$ and $\Sigma_{C}$.

We now prove the following central theorem on the existence of NEQ entropy for any $M_{\mathrm{ieq}}$ with $n^{*}$ independent state variables.

**Theorem** **6.**
*Existence: The SI-entropy S(Z(t)) for any $M_{\mathrm{ieq}}$ exists, has a unique thermodynamic temperature T, and is additive in $S_{Z}$.*


**Proof.** According to the postulates of classical thermodynamics, EQ entropies exist in $S_{X}$ and are continuous. Therefore, Equation ([Disp-formula FD74a-entropy-23-01584]) proves the existence of the entropy S(Z(t)) for any $M_{\mathrm{ieq}}\;$in the state space $S_{Z}$, and is continuous. It follows from the existence of S(Z(t)) that $M_{\mathrm{ieq}}\;$has a unique thermodynamic temperature *T*. In addition, S(Z(t)) is also additive as follows from Equation ([Disp-formula FD74c-entropy-23-01584]). This proves the theorem. □

**Corollary** **1.**
*The state space $S_{Z}$ contains $S_{X}$ as a proper subspace because of the presence of the internal variables, except when Σ is in EQ, when they become the same.*


**Proof.** For any $M_{\mathrm{ieq}}$, $S_{Z}$ contains all possible linear combinations of ξ(t) made from the set $\left\{X_{i}\right\}$. Hence, it contains $S_{X}$ as a proper subspace. In EQ, internal variables become superfluous as they are no longer independent so their affinity vanishes. Thus, $S_{Z}$ reduces to $S_{X}$ in Equation □

### 6.5. Irreversibility Inequalities in $M_{\mathrm{ieq}}$

We consider the Hamiltonian $H(\left.x\right|w,\xi )$ in $S_{Z}$. We only consider the case of extensive macrowork parameters. As $m_{k}$ evolves under the variation in W, its energy $E_{k}$ changes by $dE_{k}=-dW_{k}\;$without changing $p_{k}$; see Equation ([Disp-formula FD55b-entropy-23-01584]). The change determines the isentropic generalized macrowork $dW=F_{w}\cdot dW=-dE_{m}$. The stochasticity appears from the generalized macroheat $dQ=dE_{s}=TdS$. Recalling that for $\tilde{\Sigma }$, $T=T_{0},f_{w0}=\left(P_{0},\cdots \right),A_{0}=0$, we have in general,
(75a)$$\begin{array}{cc} d_{e}W & =-d\tilde{W}=f_{w0}\cdot d_{e}w=P_{0}dV+\cdots , \end{array}$$
(75b)$$\begin{array}{cc} d_{e}Q & =-d\tilde{Q}=T_{0}d_{e}S, \end{array}$$
where the missing terms in the top equation refires to other elements in w. The irreversible macrowork $d_{i}W≐dW-d_{e}W$ due to the thermodynamic macroforce $\Delta F^{w}$ has been given in Equation (Equation 31).

Using $d_{e}Q\;$in dQ, we find
(76a)$$d_{i}Q=TdS-T_{0}d_{i}S=\left\{\begin{array}{c} \left(T-T_{0}\right)d_{e}S+Td_{i}S\\ \left(T-T_{0}\right)dS+T_{0}d_{i}S \end{array}\right.\geq 0.$$

Equating this with $d_{i}W$ from Equation (Equation 31), we obtain for the irreversible entropy generation
(76b)$$d_{i}S=\left\{\begin{array}{c} \left\{\left(T_{0}-T\right)d_{e}S+\Delta F^{w}\cdot dW\right\}/T\\ \left\{\left(T_{0}-T\right)dS+\Delta F^{w}\cdot dW\right\}/T_{0} \end{array}\right.\geq 0;$$
see Equation (Equation 31) for $\Delta F^{w}\cdot dW$. Each term on the right side must be *nonnegative* for the second law to be valid. Thus, in terms of $\Delta F^{h}=T_{0}-T$, we see that the first term
(77a)$$\Delta F^{h}d_{e}S\geq 0,$$
in the first equation, which proves the Clausius statement of the macroheat flow from “hot” to “cold,” thus making sure that *T* indeed can be thought of as a "thermodynamic temperature" of the entire system, even if the latter is inhomogeneous. This is the requirement C4 for a thermodynamic temperature. Another important consequence of the second law comes from the first term in the second equation [55]:
(77b)$$\Delta F^{h}dS\geq 0.$$

Similarly, the second term results in the inequality
(77c)$$d_{i}W\equiv \Delta F^{w}\cdot dW\geq 0$$
due to macroforce imbalance, and consists of three separate inequalities
(77d)$$\left(f_{w}-f_{w0}\right)\cdot d_{e}w\geq 0,f_{w}\cdot d_{i}w\geq 0,A\cdot d\xi \geq 0.$$

This thus proves the inequality for $d_{i}W$ in Equation (Equation 31). Using the inequality for $d_{i}W$ in $d_{i}Q=d_{i}W$ also proves the inequality for $d_{i}Q$ in Equation ([Disp-formula FD76a-entropy-23-01584]). All these inequalities help drive the system towards EQ in accordance with the second law. We summarize the result in the following corollary.

**Corollary** **2.**
*The irreversible macrowork $d_{i}W\left(t\right)$ or macroheat $d_{i}Q\left(t\right)$ is nonnegative.*


**Proof.** From Equation ([Disp-formula FD77c-entropy-23-01584]), we find that
(78)$$d_{i}W\left(t\right)=d_{i}Q\left(t\right)>0\;\;\mathrm{for}\;T>0$$
in accordance with the second law. □

For example, if $d_{i}W$ corresponds to the irreversible macrowork done by pressure imbalance only (so that we omit the last term in $d_{i}W$ in Equation ([Disp-formula FD32b-entropy-23-01584])), then
(79)$$d_{i}W\left(t\right)=\left(P\left(t\right)-P_{0}\right)dV\left(t\right)>0.$$

If the system’s pressure $P\left(t\right)>P_{0}$, the pressure of the medium, the volume of the system increases so that dV(t)>0. In the opposite case, dV(t)<0. In both cases, $d_{i}W\left(t\right)>0$ out of equilibrium. When $d_{i}W\left(t\right)$ consists of several independent contributions, each contribution must be nonnegative in accordance with the second law and Corollary 2. The significance of the irreversible macrowork in Equation (Equation 79) has been discussed in Refs [91,92], where it is shown that this macrowork results in raising the kinetic energy of the center-of-mass of the surface separating Σ and $\tilde{\Sigma }$ by $dK_{S}$ and overcoming macrowork $dW_{\mathrm{fr}}$ done by all sorts of viscous or frictional drag. Because of the stochasticity associated with any statistical system, both energies dissipate among the particles in the system and appear in the form of macroheat $d_{i}Q\left(t\right)$.

In the absence of any heat exchange ($d_{e}S=0$) or for an isothermal system ($T=T_{0}$),we have
(80)$$d_{i}Q=Td_{i}S=d_{i}W,$$
where $d_{i}W$ is given by Equation (Equation 31).

### 6.6. Internal Variables and the Isolated System

The above formulation of MNEQT is perfectly suited for considering an isolated system Σ ($d_{e}W=d_{e}Q\equiv 0$) so that Equation (Equation 22) or $d_{i}E=0$ in Equation (Equation 9) becomes the most important thermodynamic equality. For an isolated system, dX=0 so that $d_{i}W=A\cdot d\xi$.

**Theorem** **7.**
*The irreversible entropy generated within an isolated system is still related to the dissipated macrowork performed by the internal variables.*


**Proof.** As *E* remains fixed for an isolated system ($dQ=Td_{i}S$), we have from Equation ([Disp-formula FD23a-entropy-23-01584])
(81)$$d_{i}Q=Td_{i}S=d_{i}W=A\cdot d\xi \geq 0$$
in accordance with the second law. □

Note that the above equation, though it is identical to Equation (Equation 80) in form, is very different in that $d_{i}W$ here is simply A·dξ. Same conclusion is also obtained when we apply Equation ([Disp-formula FD76b-entropy-23-01584]) to an isolated system.

**Corollary** **3.**
*Neither the entropy can increase nor will there be any dissipated work unless some internal variables are present in an isolated system. If no internal variables are used to describe an isolated system, then thermodynamics requires it to be in Equation*


**Proof.** The proof follows trivially from Equation (Equation 81). □

### 6.7. Dissipation and Thermodynamic Forces

As the inequality $d_{i}W\geq 0$, see Equations (Equation 31) and ([Disp-formula FD77c-entropy-23-01584]), or $\Delta_{i}W\geq 0$, see ([Disp-formula FD69c-entropy-23-01584]), for the irreversible macrowork for $M_{\mathrm{ieq}}$ in $S_{Z}$ follows from the second law, it is natural to identify it as the *dissipation* or the *dissipated work*; recall that $\Delta_{i}W$ is obtained by integrating $d_{i}W$ in Equation (Equation 31) over P
(82)$$\Delta_{i}W=\int_{P}\left[\left(f_{w}-f_{w0}\right)\cdot d_{e}w+f_{w}\cdot d_{i}w+A\cdot d\xi \right].$$

**Definition** **17.***The irreversible macrowork $d_{i}W\geq 0$ or $\Delta_{i}W\geq 0$ for $M_{\mathrm{ieq}}$ belonging to $S_{Z}$ along P, is identified as the dissipation or the dissipated work in the MNEQT*.

The definition is applicable regardless of Z, and has contributions from macroforce imbalance in $S_{Z}$ as given in Equation ([Disp-formula FD30a-entropy-23-01584]) at each point in P. All microstates along the path $\gamma_{P}$ of P denote IEQ-macrostates $M_{\mathrm{ieq}}$ belonging to $S_{Z}$. In this sense, the definition is a generalization of the definition of the *lost work* $\Delta W_{\mathrm{lost}}$ in the $\overset{{}^{\circ}}{M}$NEQT [11,12,14,16,25,26] in an irreversible process $\bar{P}$ between EQ macrostates $A_{\mathrm{eq}}$ and $B_{\mathrm{eq}}$ to any process P containing $M_{\mathrm{ieq}}$ between $A_{\mathrm{ieq}}$ and $B_{\mathrm{ieq}}$. The overbar in $\bar{P}$ is for EQ macrostates $A_{\mathrm{eq}}$ and $B_{\mathrm{eq}}$. The lost work is well known in the $\overset{{}^{\circ}}{M}$NEQT; see for example, p. 12 in Woods [11] or Section 20 in Landau and Lifshitz [14]. As $\Delta_{i}S$ does not directly appear in the $\overset{{}^{\circ}}{M}$NEQT, $\Delta W_{\mathrm{lost}}$, which is given by
(83)$$\Delta W_{\mathrm{lost}}={\left(\Delta_{e}W\right)}_{\mathrm{rev}}-\Delta_{e}W,$$
where ${\left(\Delta_{e}W\right)}_{\mathrm{rev}}$ is the exchange work during the reversible process ${\bar{P}}_{\mathrm{rev}}$ associated with $\bar{P}$, is used to determine $\Delta_{i}S$ indirectly as we now explain. We take $\tilde{\Sigma }$$={\tilde{\Sigma }}^{\prime }\cup {\tilde{\Sigma }}_{w}^{\prime \prime }$, where ${\tilde{\Sigma }}^{\prime }$ at constant $T_{0},P_{0}$ is thermally insulated from another working medium ${\tilde{\Sigma }}_{w}^{\prime \prime }$, with $\Sigma_{0}=\Sigma \cup \tilde{\Sigma }$. Let $\Delta_{e}Q^{\prime }$ and $\Delta_{e}W^{\prime }$ be the exchange macroquantities from ${\tilde{\Sigma }}^{\prime }$, which are well defined, and $\Delta_{e}W^{\prime \prime }=-\Delta_{e}{\tilde{W}}^{\prime \prime }$ the exchange macrowork from ${\tilde{\Sigma }}_{w}^{\prime \prime }$. We will closely follow Landau and Lifshitz ([14] (where $\Delta_{e}{\tilde{W}}^{\prime \prime }$ is denoted by *R* and $d_{e}{\tilde{W}}^{\prime \prime }$ by dR)). We first consider an infinitesimal process $\delta \bar{P}$. In the $\overset{{}^{\circ}}{M}$NEQT,
$$dE=d_{e}Q^{\prime }-d_{e}W^{\prime }-d_{e}W^{\prime \prime },$$
so that $d_{e}{\tilde{W}}^{\prime \prime }=dE-T_{0}d_{e}S^{\prime }+P_{0}dV=dE-T_{0}dS+P_{0}dV+T_{0}d_{i}S$, where we have used $dS=d_{e}S^{\prime }+d_{i}S$. We thus have
$$d_{e}{\tilde{W}}^{\prime \prime }=dG-SdT_{0}+VdP_{0}+T_{0}d_{i}S,$$
from which we obtain for $\bar{P}$
$$\Delta_{e}{\tilde{W}}^{\prime \prime }=\Delta G-\int_{\bar{P}}SdT_{0}+\int_{\bar{P}}VdP_{0}+\int_{\bar{P}}T_{0}d_{i}S,$$
which is the generalization of the known result in [14] to an arbitrary process in $S_{X}$. For fixed and constant $T_{0}$ and $P_{0}$, the first two integrals vanish and the third integral reduces to $T_{0}\Delta_{i}S$ over $\bar{P}$. Thus, as the minimum of $\Delta_{e}{\tilde{W}}^{\prime \prime }$ is given by ΔG, we derive the result in [14]:$$\Delta_{e}{\tilde{W}}^{\prime \prime }-\Delta G=T_{0}\Delta_{i}S\;\mathrm{for}\;\mathrm{fixed}\;T_{0},P_{0}.$$

Using $\Delta_{e}W=-\left(\Delta_{e}{\tilde{W}}^{\prime }+\Delta_{e}{\tilde{W}}^{\prime \prime }\right)$, and recognizing that ${\left(\Delta_{e}W\right)}_{\min}={\left(\Delta_{e}W\right)}_{\mathrm{rev}}=-\Delta F$, we have proved not only Equation (Equation 83) but also
(84a)$$\Delta W_{\mathrm{lost}}=-\Delta F-\Delta_{e}W.$$

We now show that we obtain the same result in the MNEQT, where we assume that the temperature of Σ remains equal to $T_{0}$ as assumed by Landau and Lifshitz [14], use $dE=T_{0}dS-dW$, and recognize that $d_{e}W=d_{e}W^{\prime }+d_{e}W^{\prime \prime }$. Comparing it with the dE in the $\overset{{}^{\circ}}{M}$NEQT above, we immediately obtain
(84b)$$\Delta_{i}W=\Delta W-\Delta_{e}W=T_{0}\Delta_{i}S.$$

Thus, both theories give the same result in this simple example. But our general expression for $d_{i}W$ or $\Delta_{i}W$ is not restricted to EQ terminal macrostates $A_{\mathrm{eq}}$ and $B_{\mathrm{eq}}$ of $\bar{P}$; they refer to any two end macrostates $A_{\mathrm{ieq}}$ and $B_{\mathrm{ieq}}$ of any arbitrary process P. The procedure described by Landau and Lifshitz [14] or by Woods [11] is not general enough to make $\Delta W_{\mathrm{lost}}$ useful in all cases.

We now turn to our approach and relate dissipation with the entropy generation $d_{i}S$ for Σ in Equation ([Disp-formula FD76b-entropy-23-01584]). The strategy is simple. We use Equation (Equation 22) and express $d_{i}Q$ using Equation ([Disp-formula FD76a-entropy-23-01584]). Let us use the top equation, which gives
(85a)$$Td_{i}S=\frac{\left(T_{0}-T\right)}{T_{0}}d_{e}Q+d_{i}W\geq 0.$$

For W=(V,ξ), it reduces to
(85b)$$Td_{i}S=\frac{\left(T_{0}-T\right)}{T_{0}}d_{e}Q+\left(P-P_{0}\right)dV+Ad\xi \geq 0;$$
see, for example, Ref. [16]. The first term in both equations is due to macroheat exchange $d_{e}Q$ with $\tilde{\Sigma }$ at different temperatures, which is not considered part of dissipation as we have defined above.

It is clear that the root cause of dissipation is a "*force imbalance*" $P\left(t\right)-P_{0},A\left(t\right)-A_{0}\equiv A\left(t\right)$, etc. [11,12,41,51,52,54,55,91,92] between the external and the internal forces performing macrowork, giving rise to an internal macrowork $d_{i}W$ due to all kinds of force imbalances in $\Delta F^{w}$, which is not properly captured by $d\tilde{W}-dF$ in the $\overset{{}^{\circ}}{M}$NEQT in all cases as discussed above. The force imbalance are commonly known as *thermodynamic forces* driving the system towards equilibrium.

The irreversible macrowork is present even if there is no temperature difference such as in an isothermal process as long as there exists some nonzero thermodynamic force. The resulting irreversible entropy generation is then given by $Td_{i}S=d_{i}W\geq 0$; see Equation ([Disp-formula FD76b-entropy-23-01584]). We summarize this as a conclusion [16]:

**Conclusion** **8.**
*To have dissipation, it is necessary and sufficient to have a nonzero thermodynamic force. In its absence, there can be no dissipation.*


We now prove one of the central results in the MNEQT in the following theorem.

**Theorem** **9.**
*The proportionality parameter T in Equation ([Disp-formula FD5a-entropy-23-01584]) or (Equation 62) satisfies all the criteria (C1–C4) of a sensible temperature. Therefore, we identify T as the temperature of the system in any arbitrary macrostate $M_{\mathrm{arb}}$.*


**Proof.** As dQ and dS scale the same way with the size of Σ, *T* is an intensive quantity. When the entropy is a state function in $S_{Z}$ or $S_{Z^{\prime }}⊃S_{Z}$, we have a Gibbs fundamental relation given in Equation ([Disp-formula FD71a-entropy-23-01584]). So the temperature is defined by a derivative in $S_{Z}$ or $S_{Z^{\prime }}$, the latter giving $T_{\mathrm{arb}}$. This shows that C1 is satisfied for any $M_{\mathrm{arb}}$. As we have not imposed any restrictions on the signs of dQ(t) and dS(t), the parameter T(t) can be of any sign, which shows that C2 is satisfied. To demonstrate consistency with the second law, we rewrite the top equation in Equation ([Disp-formula FD76b-entropy-23-01584]) to express $d_{i}S$ as a sum of two independent contributions
(86)$$\begin{array}{cc} d_{i}S^{Q}\left(t\right) & ≐\left(1/T-1/T_{0}\right)d_{e}Q\left(t\right),\\ d_{i}S^{W}\left(t\right) & ≐d_{i}W\left(t\right)/T\equiv d_{i}Q\left(t\right)/T, \end{array}$$
so that
(87)$$d_{i}S\left(t\right)=d_{i}S^{Q}\left(t\right)+d_{i}S^{W}\left(t\right);$$
here, $d_{i}S^{Q}\left(t\right)$ is generated solely by exchange macroheat $d_{e}Q\left(t\right)$ at different temperatures, and $d_{i}S^{W}\left(t\right)$ by the irreversible macrowork or macroheat $d_{i}W\left(t\right)\equiv d_{i}Q\left(t\right)$. The two contributions are *independent* of each other. Accordingly, both contributions individually must be nonnegative in accordance with the second law. In particular, the inequality
(88)$$d_{i}S^{Q}\left(t\right)\geq 0.$$For $T\left(t\right)>T_{0}$, $d_{e}Q\left(t\right)<0$ so that the macroheat flows from the system to the medium. For $T\left(t\right)<T_{0}$, $d_{e}Q\left(t\right)>0$ so that the macroheat flows from the medium to the system. This establishes that C3 is satisfied. As dQ(t) and dS(t) are global quantities, the parameter T(t) is also a global parameter, which means that C4 is also satisfied. This proves the theorem. □

Because of the importance of C4, we give many example in Section 10 to justify that *T* acts as a global temperature of the system even if it is composite with different temperatures. These examples leave no doubt that C4 is satisfied.

We conclude this subsection by considering a special case, also studied by Landau and Lifshitz ([14] (Section 13 and specifically Equation (13.4))): it deals with the irreversibility generated *only by macroheat exchange at different temperatures but no internal (macrowork) dissipation*. It follows from Equation (Equation 87) that $d_{i}W=0$ in this case, even though there is irreversibility ($d_{i}S\left(t\right)>0$) due to the macroheat exchange. If there are internal variables also, then $A=A_{0}=0$ to ensure $d_{i}W=0$. This example is important in that it shows that just because there is irreversibility in the system, we do not have $d_{i}W\left(t\right)\equiv d_{i}Q\left(t\right)\neq 0$. We see from Equation (Equation 87) that $d_{i}S\left(t\right)=d_{i}S^{Q}\left(t\right)$, which can be rewritten, using $d_{e}Q\left(t\right)=T_{0}d_{e}S\left(t\right)$, as
(89)$$d_{e}Q\left(t\right)=T\left(t\right)dS\left(t\right)=T_{0}d_{e}S\left(t\right),$$
a result also derived by Landau and Lifshitz; note that they use dQ for $d_{e}Q$. For $d_{i}Q=0$, we have $dQ\left(t\right)=d_{e}Q\left(t\right)$ so that Equation (Equation 89) is consistent with Equation ([Disp-formula FD5a-entropy-23-01584]), as it must. In the presence of nonzero $d_{i}Q$, Equation (Equation 89) gets modified: one must subtract $d_{i}Q$ from the right side.

### 6.8. Cyclic Process

For a general body that is not isolated, the concept of its internal equilibrium state plays a very important role in that the body can come back to this macrostate $M_{\mathrm{ieq}}$ several times in a nonequilibrium process. In a cyclic nonequilibrium process, such a macrostate can repeat itself in time after some cycle time $\tau_{c}$ so that all state variables and functions including the entropy repeat themselves:$$Z\left(t+\tau_{c}\right)=Z\left(t\right),\;M\left(t+\tau_{c}\right)=M\left(t\right),\;S\left(t+\tau_{c}\right)=S\left(t\right).$$

This ensures
$$\Delta_{c}S\equiv S\left(t+\tau_{c}\right)-S\left(t\right)=0$$
in a cyclic process. All that is required for the cyclic process to occur is that the body must start and end in the same internal equilibrium state; however, during the remainder of the cycle, the body need not be in internal equilibrium.

The same argument also applies to a cyclic process that starts and returns to $M_{\mathrm{eq}}$ after some cycle time $\tau_{c}$. However, the body need not be in EQ macrostates during the rest of the cycle. We will consider such a case when we consider a NEQ Carnot cycle in Section 12.

### 6.9. Steady State

Consider a system between two different media as shown in Figure 2. For example, we can consider Σ between two heat baths ${\tilde{\Sigma }}_{h}$ and ${\tilde{\Sigma }}_{h}$ replacing the two media in the figure. We will study this example in the MNEQT in Section 10.3 using a composite system Σ between the two heat sources. In the presence of two media, it is possible for Σ to reach a steady state, in which it satisfies conditions similar to that for a cyclic process above in terms of MI-macroquantities:
(90a)dZ=0,dS=0,
where the changes are over the system Σ. The above conditions in the MNEQT lead to important relations between exchange and irreversible macroquantities:
(90b)$$d_{i}Z=-d_{e}Z,\;d_{i}S=-d_{e}S;$$
as usual, the irreversible contributions satisfy the second law inequalities. For *E*, we have from Equation ([Disp-formula FD23a-entropy-23-01584])
dQ=dW=0,
which follows from dQ=TdS=0 for $M_{\mathrm{ieq}}$ in $S_{Z}$ or for $M_{\mathrm{nieq}}$ in $S_{Z^{\prime }}$. Therefore, in the MNEQT,
$$d_{i}Q=-d_{e}Q\geq 0,d_{i}W=-d_{e}W\geq 0.$$

As a consequence,
$$d_{e}Q=d_{e}W\leq 0,$$
a result that cannot be derived in the $\overset{{}^{\circ}}{M}$NEQT by using the first law in Equation ([Disp-formula FD23b-entropy-23-01584]).

It should be noted, as said earlier in Section 1, that the steady state occurs only over a short period $\tau ∼\tau_{\mathrm{st}}$ compared to the time $\tau_{\mathrm{EQ}}$ required for the two media to equilibrate with each other. The latter time period is extremely large compared to $\tau_{\mathrm{st}}$ because of their extreme sizes. For a time period longer than $\tau_{\mathrm{st}}$, the steady state cannot be treated as steady as Σ will begin the equilibrium process between them so that eventually at $\tau_{\mathrm{EQ}}$, $d_{i}S$ will vanish as $d_{e}Q\to 0$. We will not consider this possibility here, but can be studied in the MNEQT.

### 6.10. Intrinsic Adiabaticity Theorem

We now have a clear statement of the generalization of the adiabatic theorem [14] for nonequilibrium processes going on in a body in an arbitrary macrostate in terms of the intrinsic quantity dS. We will call it the *intrinsic adiabatic theorem*.

**Definition** **18.***Intrinsic Adiabatic Process: An intrinsic adiabatic process is an isentropic process (dS(t)=0 and not necessarily $d_{e}S\left(t\right)=0$*).

Such a process also includes the stationary limit, i.e. the steady macrostate of a non-equilibrium process discussed in the previous section. However, the extension goes beyond the conventional notion of an adiabatic process commonly dealt with in the $\overset{{}^{\circ}}{M}$NEQT, according to which an adiabatic process [14] is one for which $d_{e}Q\left(t\right)=0$, which is equivalent to $d_{e}S\left(t\right)=0$. If $d_{i}S\left(t\right)=0$, it also represents a reversible process in a thermally isolated system so that $d_{e}Q\left(t\right)=0$. One can also have dS(t)=0 in an irreversible process during which
(91)$$d_{i}S\left(t\right)=-d_{e}S\left(t\right)>0;$$
as usual, Equation (Equation 22) always remains satisfied. If the system is in a $M_{\mathrm{arb}}$, then we must also have
(92)$$d_{i}Q\left(t\right)=-d_{e}Q\left(t\right)=T_{0}d_{i}S\left(t\right)>0;$$
recall Equation (Equation 62) for a $M_{\mathrm{arb}}$.

**Theorem** **10.**
*In an intrinsic adiabatic process, the sets of microstates and of their probabilities $p_{k}$ do not change, but $d_{e}p_{k}=-d_{i}p_{k}\neq 0$ for all k.*


**Proof.** In terms $d_{e}p_{k}$ and $d_{i}p_{k}$, Equations (Equation 91) and (Equation 92) become
(93a)$$\begin{array}{cc} \sum_{k}\eta_{k}d_{i}p_{k} & =-\sum_{k}\eta_{k}d_{e}p_{k}, \end{array}$$
(93b)$$\begin{array}{cc} \sum_{k}E_{k}d_{i}p_{k} & =-\sum_{k}E_{k}d_{e}p_{k}. \end{array}$$Recognizing that there is only macrowork in dE, which requires $p_{k}$ not to change, we conclude that
$$dp_{k}=0\;\;\mathrm{for}\;\forall k$$
in an adiabatic process. As $d_{i}S\left(t\right)$ does not vanish in an irreversible process, $d_{i}p_{k}\left(t\right)$ cannot vanish. Accordingly, $d_{e}p_{k}=-d_{i}p_{k}\neq 0,\forall k$ for an irreversible adiabatic process. The conditions in Equation (Equation 108) remain valid as expected. As $p_{k}$’s do not change, no microstate can appear or disappear. This proves the theorem. □

## 7. Clausius Equality

We recall that Equation ([Disp-formula FD5a-entropy-23-01584]), which we call the *Clausius equality*, follows from the Gibbs fundamental equation for $M_{\mathrm{ieq}}$ in $S_{Z}$ or $M_{\mathrm{nieq}}$ in $S_{Z^{\prime }}$, see Equation ([Disp-formula FD71b-entropy-23-01584]). It is merely is a consequence of the state function *S* for a $M_{\mathrm{ieq}}$ or $M_{\mathrm{nieq}}$ in respective state spaces so the equality is also valid for any $M_{\mathrm{arb}}$, see Equation (Equation 62). Here, we are only concerned with some $M_{\mathrm{ieq}}$. The equality is very interesting, and should be contrasted with the *Clausius inequality*
(94)$$d_{e}Q\leq T_{0}dS.$$

First, it follows from Equation ([Disp-formula FD5a-entropy-23-01584]) that dQ/T is nothing but the *exact differential*dS for $M_{\mathrm{ieq}}$ so that
(95)∮dQ(t)/T(t)≡0
forany cyclic process; here we have added the time argument for clarity. It is only because of the use of dQ(t) in place of $d_{e}Q\left(t\right)$ that the Clausius inequality has become an equality. The equality should not be interpreted as the absence of irreversibility ($\Delta_{i}S>0$) as is clear from Equation (Equation 96) obtained by using $d_{i}S\left(t\right)\equiv dS\left(t\right)-d_{e}S\left(t\right)$ for a cyclic process taking time τ:(96)$$N\left(t,\tau \right)\equiv \oint d_{i}S\left(t\right)=-\oint d_{e}Q\left(t\right)/T_{0}\geq 0,$$
which is the second law for a cyclic process, and represents the irreversible entropy generated in a cycle. This is the original Clausius inequality. The quantity N(t) is the *uncompensated transformation* of Clausius [16] that is directly related to $d_{i}S\left(t\right)$ [109]; in contrast, $N_{0}\left(t,\tau \right)$
(97)$$N_{0}\left(t,\tau \right)\equiv \oint d_{i}Q\left(t\right)/T\left(t\right)\equiv \oint d_{i}W\left(t\right)/T\left(t\right)\geq 0,$$
where we have used the fundamental identity in Equation (Equation 22), is determined by the irreversible macroheat $d_{i}Q\left(t\right)$ or the irreversible macrowork $d_{i}W\left(t\right)$, and represents a different quantity as is evident. In terms of the two macroheats, we have
(98)$$\oint d_{i}Q\left(t\right)/T\left(t\right)=-\oint d_{e}Q\left(t\right)/T\left(t\right)\geq 0,$$
which results in a new Clausius inequality
(99)$$\oint d_{e}Q\left(t\right)/T\left(t\right)\leq 0;$$
compare with the original Clausius inequality in Equation (Equation 96).

## 8. Extended State Space and $M_{\mathrm{nieq}}$

### 8.1. Choice of Z

We first discuss how to choose a particular state space for a unique description of a macrostate M depending on the experimental setup. To understand the procedure for this, we begin by considering a set $\xi_{n}$ of internal variables $\left(\xi_{1},\xi_{2},\cdots ,\xi_{n}\right)$ and $Z_{n}≐X\cup \xi_{n}$ to form a sequence of state spaces $S_{Z}^{\left(n\right)}$. In general, one may need many internal variables, with the value of *n* increasing as M is more and more out of EQ [45] relative to $M_{\mathrm{eq}}$. We will take $n^{*}$ to be the maximum *n* in this study, even though *n*$<<n^{*}$ needed for $S_{Z}^{\left(n\right)}$ will usually be a small number in most cases. We refer to Section 6.4, where the choice of $n^{*}$ is determined by the mathematical identity in Equation ([Disp-formula FD74b-entropy-23-01584]) in $S_{Z}^{\left(n^{*}\right)}$. The two most important but distinct time scales are $\tau_{\mathrm{obs}}$, the time to make observations, and $\tau_{\mathrm{eq}}$, the equilibration time for a macrostate M to turn into $M_{\mathrm{eq}}$. For $\tau_{\mathrm{obs}}<\tau_{\mathrm{eq}}$, the system will be in a NEQ macrostate. Let $\tau_{i}\;$denote the relaxation time of $\xi_{i}$ needed to come to its equilibrium value so that its affinity $A_{i}\to 0$ [3,16,45,48,49,50]. For convenience, we order $\xi_{i}$ so that
$$\tau_{1}>\tau_{2}>\cdots ;$$
we assume distinct $\tau_{i}$’s for simplicity without affecting our conclusions. For $\tau_{1}<\tau_{\mathrm{obs}}$, all internal variables have equilibrated so they play no role in equilibration except thermodynamic forces $T-T_{0},P-P_{0}$, etc. associated with X that still drive the system towards Equation We choose *n* satisfying $\tau_{n}>\tau_{\mathrm{obs}}>\tau_{n+1}$ so that all of $\xi_{1},\xi_{2},\cdots ,\xi_{n}$ have not equilibrated (their affinities are nonzero). They play an important role in the NEQT, while $\xi_{n+1},\xi_{n+2},\cdots$ need not be considered as they have all equilibrated. This specify M
*uniquely* in $S_{Z}^{\left(n\right)}$, which was earlier identified as in IEquation

Note that NEQ macrostates with $\tau_{n+1}>\tau_{\mathrm{obs}}>\tau_{n+2}$ are not uniquely identifiable in $S_{Z}^{\left(n\right)}$, even though they are uniquely identifiable in $S_{Z}^{\left(n+1\right)}$. Thus, there are many NEQ macrostates that are not unique in $S_{Z}^{\left(n\right)}$. The unique macrostates $M_{\mathrm{ieq}}$ are special in that its Gibbs entropy $S(Z_{n})$ is a state function of $Z_{n}$ in $S_{Z}^{\left(n\right)}$. Thus, given $\tau_{\mathrm{obs}}$, we look for the window $\tau_{n}>\tau_{\mathrm{obs}}>\tau_{n+1}$ to choose the particular value of *n*. This then determines $S_{Z}^{\left(n\right)}$ in which the macrostates are in IEquation From now onward, we assume that *n* has been found and $S_{Z}^{\left(n\right)}$ has been identified. We now suppress *n* and simply use $S_{Z}$ below.

**Remark** **11.**
*The linear sizes of various subsystems introduced in Section 6.4 must be larger than the correlation length $\lambda_{\mathrm{corr}}$ as discussed elsewhere [41] for the first time, and briefly revisited in Section 2 to ensure entropy additivity; see also Section 15. Therefore, it is usually sufficient to take the linear size of Σ to be a small multiple (for example, 10 to 20) of the correlation length to obtain a proper thermodynamics, which is extensive. This means that we will usually need a theoretically manageable but small number of internal variables n.*


### 8.2. Microstate Probabilities for $M_{\mathrm{ieq}}$

As $M_{\mathrm{ieq}}$ is unique in $S_{Z}$, we need to identify the unique set $\left\{p_{k}\right\}$. If we keep W fixed in $M_{\mathrm{ieq}}$ as the parameter, then $F_{wk}$ are fluctuating microforces in $S_{Z}$ as we have seen in Section 5.2. In additions, we have microstate energies $E_{k}$ also fluctuating. We need to maximize the entropy S(Z) at fixed
$$E=\sum_{k}E_{k}p_{k},F_{w}=\sum_{k}F_{wk}p_{k}$$
by varying $p_{k}$ without changing $\left\{m_{k}\right\}$ , i.e. $E_{k}$ and $F_{wk}$. This variation has nothing to do with $dp_{k}$ in a physical process. Using the Lagrange multiplier technique, it is easy to show that the condition for this in terms of three Lagrange multipliers with obvious definitions is
(100)$$\eta_{k}=\lambda_{1}+\lambda_{2}E_{k}+\lambda_{3}\cdot F_{wk},$$
from which follows the statistical entropy $S=-(\lambda_{1}+\lambda_{2}E+\lambda_{3}\cdot F_{w})$; we have reverted back to the original symbol for the statistical entropy here. It is now easy to identify $\lambda_{2}=-\beta ,\lambda_{3}=-\beta W$ by comparing dS with dS in Equation ([Disp-formula FD71a-entropy-23-01584]) by varying *E* and W so we finally have
(101)$$p_{k}=\exp\left[\beta \left(\hat{G}-E_{k}-W\cdot F_{wk}\right)\right],$$
where $\lambda_{1}=\beta \hat{G}$ with $\hat{G}\left(t\right)$ is a normalization constant and defines a NEQ partition function
(102a)$$\exp\left(-\beta \hat{G}\right)\equiv \sum_{k}\exp\left[-\beta \left(E_{k}+W\cdot F_{wk}\right)\right].$$

It is easy to verify that
(102b)$$\hat{G}\left(T,W\right)=E+W\cdot F_{w}-TS,$$
so that if we neglect the fluctuations $E_{k}-E$ and $F_{wk}-F_{w}$ and replace $E_{k}$ by *E* and $F_{wk}$ by $F_{w}$, then $p_{k}$ reduces to the flat distribution $p_{k}=1/W\left(E,W\right)=\exp\left[\beta \left(\hat{G}-E-W\cdot F_{w}\right)\right]=\exp\left(-S\right)$ in Remark 5, which can be identified as the microstate probability in the NEQ microcanonical ensemble. It should be stressed that this is consistent with the well-known fact that thermodynamics does not describe fluctuations; the latter require using statistical mechanics [14].

It should be remarked that the Lagrange multipliers in $p_{k}$ are determined by comparing the resulting entropy to match exactly the Gibbs fundamental relation, a thermodynamic relation. This then proves that S is the same as the thermodynamic entropy *S* up to a constant [52], which can be fixed by appeals to the third law, according to which *S* vanishes at absolute zero. We avoid considering here the issue of a residual entropy, which is discussed elsewhere [45,58]. The $p_{k}$ above clearly shows the effect of irreversibility and is very different from its equilibrium analog $p_{k}^{\mathrm{eq}}$
$$p_{k}^{\mathrm{eq}}=\exp\left[\beta_{0}\left(\hat{G}\left(T_{0},w\right)-E_{k}-w\cdot f_{wk}\right)\right],$$
see Equation (Equation 28), obtained by replacing W by w, $F_{wk}$ by $f_{w0k}$, and β by $\beta_{0}$. The fluctuating $E_{k},f_{wk}$ satisfy
$$E=\sum_{k}E_{k}p_{k}^{\mathrm{eq}},f_{w0}=\sum_{k}f_{w0k}p_{k}^{\mathrm{eq}}.$$

The observation time $\tau_{\mathrm{obs}}$ is determined by the way *T* and W are changed during a process. Thus, during each change, $\tau_{\mathrm{obs}}$ must be compared with the time needed for Σ to come to the next IEQ macrostate, and for the microstate probabilities to be given by Equation (Equation 101) with the new values of *T* and W.

### 8.3. $M_{\mathrm{nieq}}$ in $S_{Z}$


We now focus on a non-unique macrostate $M_{\mathrm{nieq}}$ in $S_{Z}$. This will be needed if $\tau_{\mathrm{obs}}$ is reduced to make the process faster so that instead of falling in the window ($\tau_{n},\tau_{n+1}$), it now falls in a higher window such as ($\tau_{n+1},\tau_{n+2}$). As said above, M can now be treated as a unique macrostate in a larger state space $S_{Z^{\prime }}⊃S_{Z}$. Let $\xi^{\prime }\left(t\right)$ denote the set of additional internal variables needed over $S_{Z}$ so that
$$Z^{\prime }\left(t\right)=\left(Z\left(t\right),\xi^{\prime }\left(t\right)\right).$$

The entropy $S\left(Z^{\prime }\left(t\right)\right)=S\left(Z\left(t\right),t\right)$ for $M_{\mathrm{ieq}}\left(t\right)$ in $S_{Z^{\prime }}$ satisfies the Gibbs fundamental relation
$$dS\left(Z^{\prime }\left(t\right)\right)=\frac{\partial S}{\partial E}dE+\frac{\partial S}{\partial W}\cdot dW+\frac{\partial S}{\partial \xi^{\prime }}\cdot d\xi^{\prime },$$
where W is the work variable in $S_{Z}$. Expressing the last term as
$$\frac{\partial S}{\partial \xi^{\prime }}\cdot \frac{d\xi^{\prime }}{dt}dt,$$
we obtain the following generalization of the Gibbs fundamental relation for $M_{\mathrm{nieq}}\left(t\right)$ in $S_{Z}$:
(103a)$$dS\left(Z\left(t\right),t\right)=\frac{\partial S}{\partial E}dE+\frac{\partial S}{\partial W}\cdot dW+\frac{\partial S}{\partial t}dt,$$
where
(103b)$$\frac{\partial S}{\partial t}≐\frac{\partial S}{\partial \xi^{\prime }}\cdot \frac{d\xi^{\prime }}{dt}\geq 0.$$

In $S_{Z^{\prime }}$, we can identify the temperature *T* as the thermodynamic temperature in $S_{Z^{\prime }}$ by the standard definition. But, it is clear from the above discussion that $\partial S(Z^{\prime }\left(t\right))/\partial E$ in $S_{Z^{\prime }}$ has the same value as ∂S(Z(t),t)/∂E in $S_{Z}$. Therefore, we are now set to identify $T_{\mathrm{arb}}$ in Equation (Equation 62) as a thermodynamic temperature.

**Remark** **12.**
*$T_{\mathrm{arb}}$ in Equation (Equation 62) in $S_{Z}$ is identified by the same derivative in the Gibbs fundamental relation in $S_{Z^{\prime }}$ as follows*

(104)
$$\beta_{\mathrm{arb}}=1/T_{\mathrm{arb}}=\partial S\left(Z^{\prime }\left(t\right)\right)/\partial E=\partial S\left(Z\left(t\right),t\right)/\partial E.$$



**Definition** **19.***As the presence of ∂S/∂t above in $S_{Z}$ is due to "hidden" internal variables in $\xi^{\prime }$, we will call it the hidden entropy generation rate, and*(105a)$$d_{i}S^{\mathrm{hid}}\left(t\right)=\frac{\partial S}{\partial t}dt=\frac{\partial S}{\partial \xi^{\prime }}\cdot d\xi^{\prime }\geq 0,$$
the hidden entropy generation. It results in a hidden irreversible macrowork
(105b)$$d_{i}W^{\mathrm{hid}}≐Td_{i}S^{\mathrm{hid}}=A^{\prime }\cdot d\xi^{\prime },$$
*in $S_{Z}$ due to the hidden internal variable with affinity $A^{\prime }$.*

**Remark** **13.**
*A macrostate $M_{\mathrm{nieq}}\left(t\right)$ with S(Z(t),t) can be converted to $M_{\mathrm{ieq}}\left(t\right)$ with a state function $S(Z^{\prime }\left(t\right))$ in an appropriately chosen state space $S_{Z^{\prime }}⊃S_{Z}$ by finding the appropriate window in which $\tau_{\mathrm{obs}}$ lies. The needed additional internal variable $\xi^{\prime }$ determines the hidden entropy generation rate ∂S/∂t in Equation ([Disp-formula FD103b-entropy-23-01584]) due to the non-IEQ nature of $M_{\mathrm{nieq}}\left(t\right)$ in $S_{Z}$, and ensures validity of the Gibbs relation in Equation ([Disp-formula FD103a-entropy-23-01584]) for it, thereby providing not only a new interpretation of the temporal variation of the entropy due to hidden variables but also extends the MNEQT to $M_{\mathrm{nieq}}\left(t\right)$ in $S_{Z}$.*


The above discussion strongly points towards the possible.

**Proposition** **2.**
*The MNEQT provides a very general framework to study any $M_{\mathrm{nieq}}\left(t\right)$ in $S_{Z}$, since it can be converted into a $M_{\mathrm{ieq}}\left(t\right)$ in an appropriately chosen state space $S_{Z^{\prime }}$, with $d_{i}S^{\mathrm{hid}}\left(t\right)$ originating from hidden internal variable $\xi^{\prime }$.*


**Remark** **14.**
*In a process P resulting in $M_{\mathrm{nieq}}\left(t\right)$ in $S_{Z}$, it is natural to assume that the terminal macrostates in P are $M_{\mathrm{ieq}}$ so the affinity corresponding to $\xi^{\prime }$ must vanish in them.*


**Remark** **15.**
*By replacing Z by X, and $Z^{\prime }$ by Z, we can also express the Gibbs fundamental relation for any NEQ macrostate in $S_{X}$ as*

(106)
$$dS\left(X\left(t\right),t\right)=\frac{\partial S}{\partial E}dE+\frac{\partial S}{\partial w}\cdot dw+\frac{\partial S}{\partial t}dt,$$

*by treating M as $M_{\mathrm{ieq}}$ in $S_{Z}$. In a NEQ process $\bar{P}$ between two EQ macrostates but resulting in $M_{\mathrm{ieq}}\left(t\right)$ between them in $S_{Z}$, the affinity corresponding to ξ must vanish in the terminal EQ macrostates of $\bar{P}$.*


Equation (Equation 106) proves extremely useful to describe M in $S_{X}$ as it may not be easy to identify ξ in all cases.

**Remark** **16.**
*The explicit time dependence in the entropy for $M_{\mathrm{neq}}$ in $S_{X}$ or $M_{\mathrm{nieq}}\left(t\right)$ in $S_{Z}$ is solely due to the internal variables, which do not affect $dQ=dE_{s}$, Equation ([Disp-formula FD5a-entropy-23-01584]) remains valid, with T defined as the inverse of ∂S/∂E at fixed w,t or W,t in the two state spaces, respectively; see Equation (Equation 72).*


### 8.4. External and Internal Variations of $dp_{k}\left(t\right)$

We focus on $N_{k}$ in Section 4.2, and partition its change $dN_{k}$ in accordance with the micropartition rule; see Definition 11. We take N fixed. Then, the macropartition results in the partition for $dp_{k}$ given in Equation ([Disp-formula FD17a-entropy-23-01584]), where $d_{e}p_{k}$ is the change due to exchanges with the medium and $d_{i}p_{k}$ the change due to internal processes. It follows from the partition that
(107)$$d_{e}Q\left(t\right)\equiv \sum_{k}E_{k}d_{e}p_{k}\left(t\right),\;d_{i}Q\left(t\right)\equiv \sum_{k}E_{k}d_{i}p_{k}\left(t\right),$$
where we have replaced *d* by $d_{\alpha }$ in Equation (Equation 61). As $d_{\alpha }Q\left(t\right)$ are thermodynamic quantities, they must not change their values if we change $E_{k}$ by adding a constant to H. This requires
(108)$$\sum_{k}d_{\alpha }p_{k}\left(t\right)=0,\forall \alpha ,$$
and put a limitation on the possible variations $d_{\alpha }p_{k}$. We will assume this to be true. Using this fact, we similarly have
(109)$$d_{e}S\left(t\right)\equiv -\sum_{k}\eta_{k}d_{e}p_{k}\left(t\right),\;d_{i}S\left(t\right)\equiv -\sum_{k}\eta_{k}d_{i}p_{k}\left(t\right).\;$$

The relation $d_{e}Q\left(t\right)=T_{0}d_{e}S\left(t\right)$ can be expressed in terms of $d_{e}p_{k}\left(t\right)$
$$\sum_{k}\left(\eta_{k}-\beta_{0}E_{k}\right)d_{e}p_{k}=0.$$

Similarly, the relation $d_{i}Q\left(t\right)=T\left(t\right)dS\left(t\right)-T_{0}d_{e}S\left(t\right)$ can be written as
$$\sum_{k}\left(E_{k}-T_{0}\eta_{k}\right)d_{i}p_{k}=\left(T\left(t\right)-T_{0}\right)\sum_{k}\eta_{k}dp_{k},$$
which acts as a constraint on possible variations $d_{i}p_{k}$, given that $dp_{k}$ can be directly obtained from Equation (Equation 101).

## 9. A Model Entropy Calculation

We consider a gas of non-interacting identical structureless particles with no spin, each of mass *m*, in a fixed region confined by impenetrable walls (infinite potential well). Initially, the gas is in a NEQ macrostate, and is isolated in that region. In time, the gas will equilibrate and the microstate probabilities change in a way that the entropy increases. We wish to understand how the increase happens.

### 9.1. 1-Dimensional Ideal Gas

In order to be able to carry out an *exact calculation*, we consider the gas in a 1-dimensional box of initial size $L_{\mathrm{in}}$. As there are no interactions between the particles, the wavefunction Ψ for the gas is a product of individual particle wavefunctions ψ. Thus, we can focus on a single particle to study the nonequilibrium behavior of the gas [110,111,112]. The simple model of a particle in a box has been extensively studied in the literature but with a very different emphasis. Refs. [113,114,115,116,117] The particle only has non-degenerate eigenstates whose energies are determined by *L*, and a quantum number *k*. We use the energy scale $\varepsilon_{1}=\pi^{2}\hbar^{2}/2mL_{\mathrm{in}}^{2}$ to measure the eigenstate energies, and $\alpha =L/L_{\mathrm{in}}$ so that
(110)$$\varepsilon_{k}\left(L\right)=k^{2}/\alpha^{2};$$
the corresponding eigenfunctions are given by
(111)$$\psi_{k}\left(x\right)=\sqrt{2/L}\sin\left(k\pi x/L\right),\;\;k=1,2,3,\cdots .$$

The pressure generated by the eigenstate on the walls is given by [118]
(112)$$P_{k}\left(L\right)\equiv -\partial \varepsilon_{k}/\partial L=2\varepsilon_{k}\left(L\right)/L.$$

In terms of the eigenstate probability $p_{k}\left(t\right)$, the average energy and pressure are given by
(113a)$$\begin{array}{cc} \varepsilon (t,L) & \equiv \sum_{k}p_{k}\left(t\right)\varepsilon_{k}\left(L\right), \end{array}$$
(113b)$$\begin{array}{cc} P(t,L) & \equiv \sum_{k}p_{k}\left(t\right)P_{k}\left(L\right)=2\varepsilon \left(t,L\right)/L. \end{array}$$

The entropy follows from Equation (Equation 47) and is given for the single particle case by
$$s\left(t,L\right)\equiv -\sum_{k}p_{k}\left(t\right)\ln p_{k}\left(t\right).$$

The time dependence in ε(t) or P(t) is due to the time dependence in $p_{k}$ and $\varepsilon_{k}\left(L\right)$. Even for an isolated system, for which ε remains constant, $p_{k}$ cannot remain constant as follows directly from the second law [53] and creates a conceptual problem because the eigenstates are mutually orthogonal and there can be no transitions among them to allow for a change in $p_{k}$.

As the gas is isolated, its energy, volume and the number of particles remain constant. As it is originally not in equilibrium, it will eventually reach equilibrium in which its entropy must increase. This requires the introduction of some internal variables even in this system whose variation will give rise to entropy generation by causing internal variations $d_{i}p_{k}\left(t\right)$ in $p_{k}\left(t\right)$. Here, we will assume a single internal variable ξ(t). What is relevant is that the variation in ξ(t) is accompanied by changes $dp_{k}\left(t\right)$ occurring within the isolated system. According to our identification of heat with changes in $p_{k}\left(t\right)$, these variations must be associated with heat, which in this case will be associated with irreversible heat $d_{i}Q\left(t\right)$.

### 9.2. Chemical Reaction Approach

A way to change $p_{k}$ in an isolated system is to require the presence of some stochastic interactions, whose presence allows for transitions among eigenstates [51]. As these transitions are happening within the system, we can treat them as “chemical reactions” between different eigenstates [1,3,16] by treating each eigenstate *k* as a chemical species. During the transition, these species undergo chemical reactions to allow for the changes in their probabilities.

We follow this analogy further and extend the traditional approach [1,3,16] to the present case. For the sake of simplicity, our discussion will be limited to the ideal gas in a box; the extension to any general system is trivial. Therefore, we will use microstates $\left\{m_{k}\right\}$ instead of eigenstates in the following to keep the discussion general. Let there be $N_{k}\left(t\right)$ particles in $m_{k}$ at some instant *t* so that
$$N=\sum_{k}N_{k}\left(t\right)$$
at all times, and $p_{k}\left(t\right)=N_{k}\left(t\right)/N$. We will consider the general case that also includes the case in which final microstates refer to a box size $L^{\prime }$ different from its initial value *L*. Let us use $A_{k}$ to denote the reactants (initial microstates) and $A_{k}^{\prime }$ to denote the products (final microstates). For the sake of simplicity of argument, we will assume that transitions between microstates is described by a single chemical reaction, which is expressed in stoichiometry form as
(114)$$\sum_{k}a_{k}A_{k}⟶\sum_{k}a_{k}^{\prime }A_{k}^{\prime }.$$

Let $N_{k}$ and $N_{k}^{\prime }$ denote the population of $A_{k}$ and $A_{k}^{\prime }$, respectively, so that $N=\sum_{k}N_{k}=\sum_{k}N_{k}^{\prime }$. Accordingly, $p_{k}\left(t\right)=N_{k}\left(t\right)/N$ for the reactant and $p_{k}\left(t+dt\right)=N_{k}^{\prime }\left(t\right)/N$ for the product. The single reaction is described by a single extent of reaction ξ and we have
$$d\xi \left(t\right)\equiv -dN_{k}\left(t\right)/a_{k}\left(t\right)\equiv dN_{k^{\prime }}^{\prime }\left(t\right)/a_{k^{\prime }}^{\prime }\left(t\right)\;\;\;\mathrm{for}\;\mathrm{all}\;k,k^{\prime }.$$

It is easy to see that the coefficients satisfy an important relation
$$\sum_{k}a_{k}\left(t\right)=\sum_{k}a_{k}^{\prime }\left(t\right),$$
which reflects the fact that the change $\left|dN\right|$ in the reactant microstates is the same as in the product microstates. The *affinity* in terms of the chemical potentials μ is given by
$$A\left(t\right)=\sum a_{k}\left(t\right)\mu_{A_{k}}\left(t\right)-\sum a_{k}^{\prime }\left(t\right)\mu_{A_{k}^{\prime }}\left(t\right),$$
and will vanish only in "equilibrium," i.e. only when $p_{k}$’ s attain their equilibrium values. Otherwise, A(t) will remain non-zero. It acts as the thermodynamic force in driving the chemical reaction [1,3,16]. But we must wait long enough for the reaction to come to completion, which happens when A(t) and dξ/dt both vanish. The extent of reaction ξ is an example of an internal variable. There may be other internal variables depending on the initial NEQ macrostate. This will be discussed in the following section.

## 10. Simple Applications

### 10.1. Isothermal Expansion

Let us first consider an isothermal expansion of an ideal gas in which the temperature *T* of the gas remains constant and equal to that of the medium $T_{0}$. During an irreversible isothermal expansion, energy is pumped into the gas isothermally from outside so E(t) remains constant. The pumping of energy will result in the change $d_{e}p_{k}\left(t\right)$. This will determine $d_{e}S\left(t\right)=d_{e}Q\left(t\right)/T_{0}$. In addition, the gas may undergo transitions among various energy levels, as discussed in Section 9.2, without any external energy input, which will determine the change $d_{i}p_{k}\left(t\right)$. From Equation ([Disp-formula FD76a-entropy-23-01584]), we determine $d_{i}Q\left(t\right)=T_{0}d_{i}S\left(t\right)$, and consequently $d_{i}W\left(t\right)$. Thus,
$$\left[P\left(t\right)-P_{0}\right]dV\left(t\right)+A\left(t\right)d\xi \left(t\right)=T_{0}dS\left(t\right)-d_{e}Q\left(t\right).$$

Such a calculation will not be possible using the first law in Equation ([Disp-formula FD23b-entropy-23-01584]) in the $\overset{{}^{\circ}}{M}$NEQT.

### 10.2. Intrinsic Adiabatic Expansion

In a nonequilibrium intrinsic adiabatic process, we have $d_{i}W\left(t\right)=-$ $d_{e}Q\left(t\right)$ so the heat exchange $\left|d_{e}Q\left(t\right)\right|=$$T_{0}\left|d_{e}S\left(t\right)\right|$ is converted into the irreversible work. We can use this to determine the work $d_{i}W_{\xi }\left(t\right)$ due to the single internal variable
$$A\left(t\right)d\xi \left(t\right)=-d_{e}Q\left(t\right)-\left(P\left(t\right)-P_{0}\right)dV>0.$$

The identification $d_{i}W\left(t\right)=-$$d_{e}Q\left(t\right)$ and the calculation of A(t)dξ(t) and of $d_{i}S\left(t\right)$ cannot be done in the traditional formulation of the first law in the $\overset{{}^{\circ}}{M}$NEQT, in which $d_{e}Q\left(t\right)=0$ for the traditional adiabatic process so that $dE=-d_{e}W\left(t\right)$.

### 10.3. Composite Σ with Temperature Inhomogeneity

Here, we will show by examples that the thermodynamic temperature *T* of Σ allows us to treat it as a "black box" $\Sigma_{B}$ without knowing its detailed internal structure such as its composition in terms of two subsystems $\Sigma_{1}$ and $\Sigma_{2}$. Alternatively, we can treat Σ as a combination $\Sigma_{C}$ of $\Sigma_{1}$ at $T_{1}$ and $\Sigma_{2}$ at at $T_{2}<T_{1}$, and obtain same thermodynamics. Thus, both approaches are equivalent, which justifies the usefulness and uniqueness (see Theorem 6) of *T* as a thermodynamically appropriate global temperature.

In the following, we will consider various cases that can be obtained as special cases of the following general situation (Figure 2): $\Sigma_{1}$ in thermal contact with the medium ${\tilde{\Sigma }}_{h1}$ at temperature $T_{01}$, and $\Sigma_{2}$ in thermal contact with the medium ${\tilde{\Sigma }}_{h2}$ at temperature $T_{02}$, with the two media having no mutual interaction.

We will consider the two realizations for Σ: $\Sigma_{B}$ and $\Sigma_{C}$ to compare their predictions. As discussed for the case (b) in Section 3, $\Sigma_{1}$ and $\Sigma_{2}$ are always taken to be in EQ, but Σ in IEquation The entropies in the two realizations are
(115)$$S_{B}\left(t\right)=S\left(E\left(t\right),\xi \left(t\right)\right);S_{C}=S_{1}\left(E_{1}\left(t\right)\right)+S_{2}\left(E_{2}\left(t\right)\right),$$
and have the same value; recall that $E\left(t\right)=E_{1}\left(t\right)+E_{2}\left(t\right)$, and $\xi \left(t\right)=E_{1}\left(t\right)-E_{2}\left(t\right)$ for Σ(t); see Equation (Equation 36). For clarity, we will often use the argument *t* to emphasize the variations in time *t* in this section. In general, the irreversible entropy generation is given by
(116)$$d_{i}S\left(t\right)=d{\tilde{S}}_{1}\left(t\right)+d{\tilde{S}}_{2}\left(t\right)+dS\left(t\right),$$
where dS should be replaced by $dS_{B}$ or $dS_{C}$ as the case may be:(117)$$\begin{array}{c} \left[c\right]ldS_{B}\left(t\right)=\beta \left(t\right)dE\left(t\right)+\beta \left(t\right)A\left(t\right)d\xi \left(t\right),\\ dS_{C}\left(t\right)=\beta_{1}\left(t\right)dE_{1}\left(t\right)+\beta_{2}\left(t\right)dE_{2}\left(t\right), \end{array}\;\;\;\;\;$$
where we are using the inverse temperatures for various bodies. Let $d_{e}Q_{l}\left(t\right),l=1,2$ be the energy or macroheat transferred to $\Sigma_{l}\left(t\right)$ from ${\tilde{\Sigma }}_{h}^{\left(l\right)}$, and $dE_{\mathrm{in}}\left(t\right)=d_{e}Q_{\mathrm{in}}\left(t\right)$ the energy or macroheat transferred from $\Sigma_{1}\left(t\right)$ to $\Sigma_{2}\left(t\right)$. We have, using $\delta_{1}=+1$ and $\delta_{2}=-1,$
(118a)$$\begin{array}{cc} dE_{l}\left(t\right) & =d_{e}Q_{l}\left(t\right)+\delta_{l}dE_{\mathrm{in}}\left(t\right),\\ dE(t) & =d_{e}Q_{1}\left(t\right)+d_{e}Q_{2}\left(t\right),\\ d{\tilde{S}}_{l}\left(t\right) & =-d_{e}S_{l}\left(t\right)=-\beta_{0l}d_{e}Q_{l}\left(t\right). \end{array}$$

We see that dE(t) is unaffected by the internal energy transfer $dE_{\mathrm{in}}\left(t\right)$, while
(118b)$$d\xi \left(t\right)=d_{e}Q_{1}\left(t\right)-d_{e}Q_{2}\left(t\right)+2dE_{\mathrm{in}}\left(t\right),$$
is affected by the macroheat exchange disparity $d_{e}Q_{1}\left(t\right)-d_{e}Q_{2}\left(t\right)$ along with $dE_{\mathrm{in}}\left(t\right)$.

We finally have
(119)$$d_{i}S\left(t\right)=-\sum_{l}\beta_{0l}d_{e}Q_{l}\left(t\right)+dS.$$

We now consider various cases to make our point.

#### 10.3.1. Isolated Σ

We first consider the realization $\Sigma_{B}$. Using $dE\left(t\right)=dE_{1}\left(t\right)+dE_{2}\left(t\right),d\xi \left(t\right)=dE_{1}\left(t\right)-dE_{2}\left(t\right)$, see Equations (Equation 36) and (Equation 117) for $dS_{B}\left(t\right)$ above, we obtain
(120a)$$\beta \left(t\right)=\frac{\beta_{1}\left(t\right)+\beta_{2}\left(t\right)}{2},\;\beta \left(t\right)A\left(t\right)=\frac{\beta_{1}\left(t\right)-\beta_{2}\left(t\right)}{2}.$$

This identifies $T\left(t\right)$ in terms of $T_{1}\left(t\right)$ and $T_{2}\left(t\right)$. As EQ is attained, $T\left(t\right)\to T_{0}$, the EQ temperature between $\Sigma_{1}$ and $\Sigma_{2}$, and $A\left(t\right)\to A_{0}=0$ as expected. In the following, we will use $A^{\prime }\left(t\right)$ for β(t)A(t) for simplicity. In terms of β and $A^{\prime }$, we also have
(120b)$$\beta_{1}=\beta +A^{\prime },\;\beta_{2}=\beta -A^{\prime }.$$

We now justify that in this simple example, $A^{\prime }\left(t\right)d\xi \left(t\right)$ determines $d_{i}S\left(t\right)$ due to irreversibilty in Σ(t); see Equation (Equation 80). Setting dE(t)=0 in $dS_{B}\left(t\right)$, we have by direct evaluation,
(121)$$d_{i}S\left(t\right)=A^{\prime }\left(t\right)d\xi \left(t\right)=\beta \left(t\right)d_{i}W\left(t\right).$$

The last equation follows from the general result in Equation (Equation 81). It should be emphasized that the existence of $d_{i}S\left(t\right)\geq 0$ due to ξ in $M_{\mathrm{ieq}}$ is consistent with $M_{\mathrm{ieq}}$ as a NEQ macrostate, even though its entropy is a state function in the extended state space.

We now consider $\Sigma_{C}$, which is also very instructive to understand the origin of $d_{i}S\left(t\right)$ in a different way. Considering internal energy or macroheat transfer $dE_{\mathrm{in}}\left(t\right)=d_{e}Q_{\mathrm{in}}\left(t\right)$ between $\Sigma_{1}\left(t\right)$ and $\Sigma_{2}\left(t\right)$ at some instant *t*, we have
(122a)$$dS_{1}\left(t\right)=\frac{dE_{\mathrm{in}}\left(t\right)}{T_{1}\left(t\right)},\;dS_{2}\left(t\right)=-\frac{dE_{\mathrm{in}}\left(t\right)}{T_{2}\left(t\right)},$$
due to this transfer. This results in
(122b)$$d_{i}S\left(t\right)=\left[\beta_{1}\left(t\right)-\beta_{2}\left(t\right)\right]dE_{\mathrm{in}}\left(t\right)=A^{\prime }d\xi \left(t\right),$$
since $d\xi \left(t\right)=dE_{1}\left(t\right)-dE_{2}\left(t\right)=2dE_{\mathrm{in}}\left(t\right)$. Thus, the physical origin of $d_{i}S\left(t\right)$ is the internal entropy change of the subsystems.

#### 10.3.2. Σ Interacting with ${\tilde{\Sigma }}_{h}$

To further appreciate the physical significance of the NEQ T(t) of the above composite system Σ(t), we allow it to interact with ${\tilde{\Sigma }}_{h}$, a heat bath, at the EQ temperature $T_{0}$. For this, we take ${\tilde{\Sigma }}_{h1}$ and ${\tilde{\Sigma }}_{h2}$ at the same common temperature $T_{0}=T_{01}=T_{02}$ so that we can treat them as a single medium ${\tilde{\Sigma }}_{h}$ with macroheat exchange $d_{e}Q\left(t\right)$. We thus obtain from Equation (Equation 119)
$$d_{i}S\left(t\right)=-\beta_{0}d_{e}Q\left(t\right)+dS.$$

We will consider two different kinds of interaction below:

(i) We first consider $\Sigma_{B}\left(t\right)$ in $M_{\mathrm{ieq}}$ at T(t) so we use $dS_{B}\left(t\right)$ above. We thus have (using the identity $d_{e}S\left(t\right)=\beta_{0}d_{e}Q\left(t\right)$)
(123)$$d_{i}S\left(t\right)=\left[\beta \left(t\right)-\beta_{0}\right]d_{e}Q\left(t\right)+A^{\prime }\left(t\right)d\xi \left(t\right),$$
which is consistent with the general identity given by the top equation in Equation ([Disp-formula FD76a-entropy-23-01584]), a result which was derived for a single system at temperature $T\left(t\right)$. This confirms that the composite $\Sigma_{C}$ here can be treated as a noncomposite $\Sigma_{B}$ at $T\left(t\right)$. To be convinced that the above $d_{i}S\left(t\right)$ includes the internally generated irreversibility in Equation (Equation 121) due to macroheat transfer between $\Sigma_{1}\left(t\right)$ and $\Sigma_{2}\left(t\right)$, we only have to set $d_{e}S\left(t\right)=0$ to ensure the isolation of Σ. We reproduce Equation (Equation 121) as $d_{i}Q\left(t\right)=d_{i}W\left(t\right)$. The remaining source of irreversibility $T\left(t\right)d_{i}S^{Q}\left(t\right)$ given by the first term above is due to external macroheat exchange between Σ and ${\tilde{\Sigma }}_{h}$
(124)$$d_{i}S^{Q}\left(t\right)=\left[T_{0}\beta \left(t\right)-1\right]d_{e}S\left(t\right),$$
as expected; see the first term on the right in Equation ([Disp-formula FD85b-entropy-23-01584]).

(ii) We take treat Σ(t) as $\Sigma_{C}\left(t\right)$ in contact with ${\tilde{\Sigma }}_{h}$. We deal directly with the two macroheat exchanges $d_{e}Q_{l}\left(t\right),l=1,2$ to $\Sigma_{l}\left(t\right)$ from ${\tilde{\Sigma }}_{h}$, and the internal energy transfer $dE_{\mathrm{in}}\left(t\right)$. Using $dE_{l}\left(t\right)$ from Equation ([Disp-formula FD118a-entropy-23-01584]) in $dS_{C}$ given in Equation (Equation 119), we find that
$$d_{i}S\left(t\right)=\sum_{l}\left[\beta_{l}\left(t\right)-\beta_{0}\right]d_{e}Q_{l}\left(t\right)+\left[\beta_{1}\left(t\right)-\beta_{2}\left(t\right)\right]dE_{\mathrm{in}}\left(t\right).$$

Using Equation ([Disp-formula FD120b-entropy-23-01584]) to express $\beta_{l}$, we can rewrite the above equation as
$$d_{i}S\left(t\right)=\left[\beta \left(t\right)-\beta_{0}\right]d_{e}Q\left(t\right)+A^{\prime }d\xi ,$$
where we have used the identity
(125)$$d_{e}Q\left(t\right)=d_{e}Q_{1}+d_{e}Q_{2},$$
and have found
(126)$$d\xi =d_{e}Q_{1}-d_{e}Q_{2}+2dE_{\mathrm{in}}$$
using its general definition $d\xi \left(t\right)=dE_{1}\left(t\right)-dE_{2}\left(t\right)$. We thus see that $d_{i}S\left(t\right)$ obtained by both realizations are the same as they must. However, the realization $\Sigma_{C}\left(t\right)$ allows us to also identify dξ.

Each exchange generates irreversible entropy following Equation (Equation 124). Using $d_{e}Q\left(t\right)=d_{e}Q_{1}\left(t\right)+d_{e}Q_{2}\left(t\right)$ in dQ(t)=T(t)dS(t) to determine $d_{i}Q\left(t\right)$, we find the generalization of Equation (Equation 123):(127)$$d_{i}S\left(t\right)=\left[\beta_{1}\left(t\right)-\beta_{2}\left(t\right)\right]d_{e}Q_{\mathrm{in}}\left(t\right)+\sum_{l}\left[T_{0}\beta_{l}\left(t\right)-1\right]d_{e}S_{l}\left(t\right).$$

It is easy to see that the last term above gives nothing but the sum of the irreversible entropies due to external exchanges of macroheat by $\Sigma_{1}\left(t\right)$ and $\Sigma_{2}\left(t\right)$ with ${\tilde{\Sigma }}_{h}$:(128)$$d_{i}S^{Q}\left(t\right)=d_{i}S_{1}^{Q}\left(t\right)+d_{i}S_{2}^{Q}\left(t\right),$$
where
(129)$$d_{i}S_{l}^{Q}\left(t\right)=\left[T_{0}\beta_{l}\left(t\right)-1\right]d_{e}S_{l}\left(t\right),l=1,2$$
is the external entropy exchange of $\Sigma_{l}\left(t\right)$ with ${\tilde{\Sigma }}_{h}$.

Thus, whether we treat Σ as a system $\Sigma_{B}$ at temperature T(t) or a collection $\Sigma_{C}$ of $\Sigma_{1}\left(t\right)$ and $\Sigma_{2}\left(t\right)$ at temperatures $T_{1}\left(t\right)$ and $T_{2}\left(t\right)$, respectively, we obtain the same irreversibility. In other words, T(t) is a sensible thermodynamic temperature even in the presence of inhomogeneity.

### 10.4. Σ Interacting with ${\tilde{\Sigma }}_{h1}$ and
${\tilde{\Sigma }}_{h2}$

We now consider our composite Σ in thermal contact with two distinct and mutually noninteracting stochastic media ${\tilde{\Sigma }}_{h1}$ and ${\tilde{\Sigma }}_{h2}$ at temperatures $T_{01}$ and $T_{02}$. We will again discuss the two different realizations as above.

(i) We first consider $\Sigma_{B}\left(t\right)$ at temperature T(t), which interacts with the two ${\tilde{\Sigma }}_{h}$’s, and use the general result in Equation (Equation 119). A simple calculation using $dS_{B}$ generalizes Equation (Equation 123) and yields
(130a)$$d_{i}S\left(t\right)=\sum_{l}\left[\beta \left(t\right)-\beta_{0l}\right]d_{e}Q_{l}\left(t\right)+A^{\prime }\left(t\right)d\xi \left(t\right),$$
since this reduces to that result when we set $\beta_{01}=\beta_{02}=\beta_{0}$. As above, $d_{i}Q\left(t\right)=d_{i}W\left(t\right)=A\left(t\right)d\xi \left(t\right)$; see Equation (Equation 121), which gives rise to the last term above. Thus, setting $d_{e}Q_{l}\left(t\right)=0,l=1,2$ to make Σ isolated, we retrieve $d_{i}S\left(t\right)$ in Equation (Equation 121) as expected. The first sum above gives the external entropy exchanges with the two stochastic media as above.

(ii) We now consider $\Sigma_{C}$, and allow ${\tilde{\Sigma }}_{h1}$ to directly interact with $\Sigma_{1}\left(t\right)$ at temperature $T_{1}\left(t\right)$ and ${\tilde{\Sigma }}_{h2}$ to directly interact with $\Sigma_{2}\left(t\right)$ at temperature $T_{2}\left(t\right)$. Using $dS_{C}$ generalizes Equation (Equation 123) and yields
(130b)$$d_{i}S\left(t\right)=\sum_{l}\left[\beta_{l}\left(t\right)-\beta_{0l}\right]d_{e}Q_{l}\left(t\right)+\left[\beta_{1}\left(t\right)-\beta_{2}\left(t\right)\right]dE_{\mathrm{in}}\left(t\right).$$

Again using Equation ([Disp-formula FD120b-entropy-23-01584]) to express $\beta_{l}$, we can rewrite the above $d_{i}S\left(t\right)$ as the $d_{i}S\left(t\right)$ in Equation ([Disp-formula FD130a-entropy-23-01584]) for $\Sigma_{B}$, and also find that dξ is given by Equation (Equation 126).

It should be emphasized that the determination of $d_{i}S\left(t\right)$ in Equations ([Disp-formula FD130a-entropy-23-01584]) and ([Disp-formula FD130b-entropy-23-01584]) is valid for all cases of Σ interacting with ${\tilde{\Sigma }}_{h1}$ and ${\tilde{\Sigma }}_{h2}$ as we have not imposed any conditions on $T_{1}\left(t\right)$ and $T_{2}\left(t\right)$ with respect to $T_{01}$ and $T_{02}$, respectively. Thus it is very general. The derivation also applies to the NEQ stationary state, which happens when $T_{1}\left(t\right)\to T_{01}$ and $T_{2}\left(t\right)\to T_{02}$. For the stationary case, using Equation ([Disp-formula FD130b-entropy-23-01584]), we have
(131)$$d_{i}S^{\mathrm{st}}=\left[\beta_{01}-\beta_{02}\right]dE_{\mathrm{in}},$$
where all quantities on the right have their steady values. Thus, $d_{i}S^{\mathrm{st}}$ is only determined by the stationary value of the internal energy exchange $dE_{\mathrm{in}}$. The reader can easily verify that $d_{i}S\left(t\right)$ in Equation ([Disp-formula FD130a-entropy-23-01584]) also reduces to the above result in the stationary limit.

From the above examples, we see that we can consider Σ in any of the two realization $\Sigma_{B}$ and $\Sigma_{C}$ as we obtain the same thermodynamics in that $d_{i}S\left(t\right)$ is identical. We emphasize this important observation by summarizing it in the following conclusion.

**Conclusion** **11.**
*If we consider Σ(t) as a single system $\Sigma_{B}$ with an uniform temperature T(t) and with an internal variable ξ(t), we do not need to consider the energy transfer $dE_{\mathrm{in}}\left(t\right)$ explicitly to obtain $d_{i}S\left(t\right)$. If we consider Σ(t) as a composite system $\Sigma_{C}$ formed of $\Sigma_{1}\left(t\right)$ and $\Sigma_{2}\left(t\right)$ at their specific temperatures, then we specifically need to consider the energy transfer $dE_{\mathrm{in}}\left(t\right)$ to obtain $d_{i}S\left(t\right)$ but no internal variable.*


This conclusion emphasizes the most important fact of the MNEQT that the homogeneous thermodynamic temperature T(t) of $\Sigma_{B}$ can also describe an inhomogeneous system $\Sigma_{C}$. This observation justifies using the thermodynamic temperature T(t) for treating Σ(t) as a single system $\Sigma_{B}$, a black box, without any need to consider the internal energy transfers.

The above discussion can be easily extended to also include inhomogeneities such as two different work media ${\tilde{\Sigma }}_{w}^{\left(1\right)}$ and ${\tilde{\Sigma }}_{w}^{\left(2\right)}$ corresponding to different pressures $P_{01\;}$and $P_{02}$. We will not do that here.

### 10.5. Σ Interacting with ${\tilde{\Sigma }}_{w}$ and
${\tilde{\Sigma }}_{h}$

In this case, Σ is specified by two observables *E* and *V* so to describe any inhomogeneity will require considering at least two subsystems $\Sigma_{1}$ and $\Sigma_{2}$ specified by $E_{1},V_{1}$ and $E_{2,}V_{2}$, respectively. From these four observables, we construct the following four combinations
$$\begin{array}{cc} E_{1}+E_{2} & =E,\xi_{E}=E_{1}-E_{2},\\ V_{1}+V_{2} & =V,\xi_{V}=V_{1}-V_{2}, \end{array}$$
to express the entropy of the system
$$S\left(E,V,\xi_{E},\xi_{V}\right)=S_{1}\left(E_{1},V_{1}\right)+S_{2}\left(E_{2},V_{2}\right)$$
in terms of
$$E_{1,2}=\frac{E\pm \xi_{E}}{2},V_{1,2}=\frac{V\pm \xi_{V}}{2}.$$

Note that we have assumed that $\Sigma_{1}$ and $\Sigma_{2}$ are in EQ (no internal variables for them). We now follow the procedure carried out in Section 10.3 to identify thermodynamic temperature *T*, pressure *P*, and affinities:(132)$$\begin{array}{c} \beta =\frac{\left(\beta_{1}+\beta_{2}\right)}{2},\beta P=\frac{\left(\beta_{1}P_{1}+\beta_{2}P_{2}\right)}{2},\\ \beta A_{E}=\frac{\left(\beta_{1}-\beta_{2}\right)}{2},\beta A_{V}=\frac{\left(\beta_{1}P_{1}-\beta_{2}P_{2}\right)}{2}. \end{array}$$

All these quantities are SI-quantities and have the same values regardless of whether Σ is isolated or interacting. A more complicated inhomogeneities will require more internal variables.

**Remark** **17.**
*We now make an important remark about Equation ([Disp-formula FD85b-entropy-23-01584]) that contains only a single internal variable. From what is said above, it must include at least two internal variables if Σ contains inhomogeneity. in both E and V. If it contains inhomogeneity. in only one variable, then and only then we will have at least one internal variable. Thus, either we will $\xi_{E}$ or $\xi_{V}$ as the case may be.*


## 11. Tool-Narayanaswamy Equation

We consider a simple NEQ laboratory problem to model the situation in a glass [55]. It is a system consisting of two "interpenetrating" parts at different temperatures $T_{1}$ and $T_{2}>T_{1}$, but insulated from each other so that they cannot come to equilibrium. The two parts are like slow and fast motions in a glass, and the insulation allows us to treat them as independent, having different temperatures. This is a very simple model for a glass. A more detailed discussion is given elsewhere [55], where each part was assumed to be in EQ macrostates. Here, we go beyond the earlier discussion, and assume that the two parts are in some IEQ macrostates $M_{1}$ and $M_{2}$ with temperatures $T_{1}$ and $T_{2}$, respectively; we have suppressed "ieq" in the subscripts for simplicity. Thus, there are irreversible processes going on within each part so that there are nonzero irreversible macroheat $d_{i}Q_{1}$ and $d_{i}Q_{2}$ generated within each part. We wish to identify the temperature of the system that we treat as a black box $\Sigma_{B}$. This will require introducing its global temperature *T*. However, we also need to relate it to $T_{1}$ and $T_{2}$ so that we need to treat Σ as $\Sigma_{C}$. We now imagine that each part is added a certain *infinitesimal* amount of exchange macroheat from outside, which we denote by $d_{e}Q_{1}$ and $d_{e}Q_{2}$ so that $dQ_{1}=d_{e}Q_{1}+d_{i}Q_{1}$ and $dQ_{2}=d_{e}Q_{2}+d_{i}Q_{2}$. This does not affect their temperatures. We assume the entropy changes to be $dS_{1}$ and $dS_{2}$. Then, we have for the net macroheat and entropy change
$$dQ=dQ_{1}+dQ_{2},\;dS=dS_{1}+dS_{2}.$$

We introduce the temperature *T* by dQ=TdS. This makes it a thermodynamic temperature of the black box. Using $dQ_{1}=T_{1}dS_{1},\;dQ_{2}=T_{2}dS_{2}$, we immediately find
$$dQ\left(1/T-1/T_{2}\right)=dQ_{1}\left(1/T_{1}-1/T_{2}\right).$$

By introducing $x=dQ_{1}/dQ$, which is determined by the setup, we find that *T* is given by
(133)$$\frac{1}{T}=\frac{x}{T_{1}}+\frac{1-x}{T_{2}}.$$

As *x* is between 0 and 1, it is clear that *T* lies between $T_{1}$ and $T_{2}$ depending on the value of *x*. For x=1/2, this heuristic model calculation reduces to that in Equation ([Disp-formula FD120a-entropy-23-01584]) as expected. The derivation also shows that the thermodynamic temperature *T* is not affected by having two nonoverlapping parts or overlapping parts. A similar relation also exists for the pressure *P* of a composite system; see Equation (Equation 156).

If the insulation between the parts is not perfect, there is going to be some energy transfer between the two parts, which would result in maximizing the entropy of the system. As a consequence, their temperatures will eventually become the same. During this period, *T* will also change until all the three temperatures become equal. This will require additional internal variable or variables as in Section 10.3.

## 12. Irreversible Carnot Cycle

Let us consider an “irreversible” Carnot engine running between two heat sources ${\tilde{\Sigma }}_{h1}$ and ${\tilde{\Sigma }}_{h2}$ as shown in Figure 5 that are always maintained at fixed temperatures $T_{01}$ and $T_{02}$, respectively, during each cyclic process $P_{\mathrm{cyc}}$. As Σ needs to perform work, we also need to consider it to be in constant with a work source ${\tilde{\Sigma }}_{w}$. We first observe the following features of a reversible Carnot cycle. The system, which we take to be formed by an ideal gas, starts in thermal contact with ${\tilde{\Sigma }}_{h1}$ in $A_{\mathrm{eq}}=M_{\mathrm{eq}}\left(T_{01},V_{1}\right)$ as it expands to $V_{2}>V_{1}$, and ends in $M_{\mathrm{eq}}\left(T_{01},V_{2}\right)$ through an isothermal process $P_{1\mathrm{eq}}$ resulting in $\Delta_{e}Q_{1\mathrm{eq}}=\Delta_{e}W_{1\mathrm{eq}}>0$. It is then detached from ${\tilde{\Sigma }}_{h1}$ so no heat is exchanged ($\Delta_{e}Q_{2\mathrm{eq}}=0$) but exchanges work $\Delta_{e}W_{2\mathrm{eq}}>0$ during the process $P_{2\mathrm{eq}}$ as it expands to $V_{3}$ and ends in $M_{\mathrm{eq}}\left(T_{02},V_{3}\right)$ at temperature $T_{02}$. The system is brought in thermal contact with ${\tilde{\Sigma }}_{h2}$ now, and the volume is compressed to $V_{4}$ isothermally during the process $P_{3\mathrm{eq}}$ and ends in $M_{\mathrm{eq}}\left(T_{02},V_{4}\right)$. During $P_{3\mathrm{eq}}$, $\Delta_{e}Q_{3\mathrm{eq}}=\Delta_{e}W_{3\mathrm{eq}}<0$. The choice of $V_{4}$ is chosen so that Σ comes back to $A_{\mathrm{eq}}=M_{\mathrm{eq}}\left(T_{01},V_{1}\right)$ along a process $P_{4\mathrm{eq}}$ after detaching it from ${\tilde{\Sigma }}_{h2}$ during which $\Delta_{e}Q_{4\mathrm{eq}}=0$, but $\Delta_{e}W_{4\mathrm{eq}}<0$. The four segments bring back Σ to its starting state $A_{\mathrm{eq}}$, and form a cycle $P_{\mathrm{eq},\mathrm{cyc}}$. It is well known that the EQ efficiency $\epsilon_{\mathrm{eq}}$ of the Carnot cycle is
(134a)$$\epsilon_{\mathrm{eq}}=1-T_{02}/T_{01},$$
so that
(134b)$$\Delta_{e}W_{\mathrm{eq}}=\epsilon_{\mathrm{eq}}\Delta_{e}Q_{1},$$
the equilibrium macrowork obtained from the cycle for a given $\Delta_{e}Q_{1}$.

We now consider an irreversible cyclic process P, which consists of the same four segments except that some or all may be irreversible. We have discussed such a process in Section 6.8. However, to have a cyclic process, the system must start and end in $M_{\mathrm{eq}}\left(T_{01},V_{1}\right)$, which does not require any internal variable. Being an irreversible process, there is no guarantee that Σ would be in EQ macrostates at the end of $P_{1},P_{2}$, and $P_{3}$; $P_{4}$ must bring Σ to the EQ initial macrostate $M_{\mathrm{eq}}\left(T_{01},V_{1}\right)$. However, we will simplify the calculation here by assuming that the end states in $P_{1},P_{2}$, and $P_{3}$ are $M_{\mathrm{eq}}\left(T_{01},V_{2}\right),M_{\mathrm{eq}}\left(T_{02},V_{3}\right)$, and $M_{\mathrm{eq}}\left(T_{02},V_{4}\right)$, respectively. However, relaxing this condition does not change the results below.

Being a cyclic process, we have
(135a)$$\Delta_{c}E=\Delta_{c}S=0$$
over P. Thus, over P,
(135b)$$\Delta_{e}Q=\Delta_{e}W.$$

In the MNEQT, we also have over P,
(135c)ΔQ=ΔW.

Let $\Delta_{e}Q_{1}=T_{01}\Delta_{e}S_{1}$ and $\Delta_{e}Q_{3}=T_{02}\Delta_{e}S_{3}$ be the macroheat exchanges during $P_{1}$ and $P_{3}$, and $\Delta_{e}Q=\Delta_{e}Q_{1}+\Delta_{e}Q_{3}$. Similarly, let $\Delta_{e}W_{l}$ be the macrowork exchanges during $P_{l},l=1,2,3,4$, and $\Delta_{e}W=\sum_{l}\Delta_{e}W_{l}$, the net exchange macrowork. From $\Delta_{c}S=0$ follows
$$\Delta_{e}Q_{1}/T_{01}+\Delta_{e}Q_{3}/T_{02}+\Delta_{i}S=0,$$
which can be rewritten by simple manipulation as
(136a)$$\Delta_{e}W=\Delta_{e}Q_{1}\left(1-T_{02}/T_{01}\right)-T_{02}\Delta_{i}S,$$
where we have used the identity in Equation ([Disp-formula FD135b-entropy-23-01584]). We can also express it as
(136b)$$\Delta_{e}W=\epsilon_{\mathrm{eq}}\Delta_{e}Q_{1}-T_{02}\Delta_{i}S\leq \Delta_{e}W_{\mathrm{eq}}.$$

The efficiency of the irreversible Carnot cycle is given by
(137)$$\epsilon_{\mathrm{irr}}=\epsilon_{\mathrm{eq}}-T_{02}\Delta_{i}S/\Delta_{e}Q_{1}\leq \epsilon_{\mathrm{eq}}.$$

We remark that $\epsilon_{\mathrm{irr}}$ above is similar to the result obtained by Eu ([109] (see Equation (7.139))) in which the numerator of the last term is identified as "the total dissipation" by Eu; however, the analysis is tedious compared to the one given here.

The determination of $\Delta_{i}S$ requires the extended state space $S_{Z}$ needed for the four process segments. We will focus on Equation ([Disp-formula FD85b-entropy-23-01584]). Let $T_{l}\left(t\right)$ and $P_{l}\left(t\right)$ be the temperature and pressure of Σ along $P_{l}$, respectively, with l=1,⋯,4 for $P_{l}$. As seen from Equation (Equation 132), we need at least two internal variables $\xi_{El}$ and $\xi_{Vl}$ along $P_{l}$ that are usually different along the four segments. The corresponding affinities must vanish at the end points of each segment because they are EQ macrostates. We will assume this to be the case, and introduce
$$A_{l}\left(t\right)\cdot d\xi_{l}\left(t\right)=A_{El}\left(t\right)d\xi_{El}\left(t\right)+A_{Vl}\left(t\right)d\xi_{Vl}\left(t\right)$$
for each segment. Then,
(138)$$\begin{array}{cc} \Delta_{i}S_{l} & =\int_{P_{l}}[\left(\beta_{l}\left(t\right)-\beta_{0l}\right)d_{e}Q_{l}\left(t\right)+\left(P_{l}\left(t\right)-P_{0l}\right)dV\left(t\right)\\ \; & +\beta_{l}\left(t\right)A_{l}\left(t\right)\cdot d\xi_{l}\left(t\right)], \end{array}$$
where $d_{e}Q_{l}\left(t\right)=0$ for l=2,4, and where $P_{0l}$ is the external pressures of ${\tilde{\Sigma }}_{w}$ along $P_{l}$ and must be the same as for the reversible Carnot engine. This then determines $\Delta_{i}S$ so that $\epsilon_{\mathrm{irr}}$ is determined.

Using $\Delta_{i}Q=\Delta Q-\Delta_{e}Q$, we also have
(139)$$\Delta_{i}W=\sum_{l=1}^{4}\int_{P_{l}}\left[T_{l}\left(t\right)d_{i}S_{l}\left(t\right)+\left(T_{l}\left(t\right)-T_{0l}\right)d_{e}S_{l}\left(t\right)\right]$$

Recognizing that $d_{e}S_{l}\left(t\right)$ is nonzero only for l=1,3, we can also rewritten $\Delta_{i}W$ as
(140)$$\Delta_{i}W=\sum_{l=1,3}\int_{P_{l}}\left(\beta_{0l}/\beta_{l}\left(t\right)-1\right)d_{e}Q_{l}\left(t\right)+\int_{P}T\left(t\right)d_{i}S\left(t\right).$$

It should be noted that nowhere did we use the vanishing of the affinities in the EQ states at the end of $P_{1},P_{2}$, and $P_{3}$ so the calculation above is not limited by this requirement. Thus, the results in this section are general.

## 13. Origin of Friction and Brownian Motion

It is well known, see for example Kestin ([12] Sections 4.7 and 5.12), that there are several ways to incorporate friction in a system in thermodynamics. This has to do with the difficulties in making an unambiguous distinction between various possibilities of exchange macroheat in a process P. We overcome this problem by using the MNEQT in which this is not an issue as both dQ and $d_{e}Q$ are uniquely defined. We identify the origin of friction in our approach [41,51,52,91,92] by considering relative motion between parts of a system or between the system and the medium; see also [14,32]. Such a situation arises during sudden mixing of fluids or in a Couette flow or when friction is involved between two bodies. The origin of friction is also applicable to NEQ terminal states of the process in the MNEQT.

### 13.1. Piston-Gas System

We consider the piston gas system in Figure 4a. As discussed in Section 3.5, the system entropy in $M_{\mathrm{ieq}}$ is $S(E,V,P_{\mathrm{gc}},P_{p})$. Hence, the corresponding Gibbs fundamental relation becomes
$$dS=\beta [dE+PdV-V_{\mathrm{gc}}\cdot dP_{\mathrm{gc}}-V_{p}\cdot dP_{p}],$$
where we have used the conventional conjugate fields
(141)$$\begin{array}{c} {}[c]c\beta ≐\partial S/\partial E,\beta P≐\partial S/\partial V,,\\ \beta V_{\mathrm{gc}}≐-\partial S/\partial P_{\mathrm{gc}},\beta V_{p}≐-\partial S/\partial P_{p} \end{array}\;\;\;\;\;\;\;\;\;\;\;\;\;\;$$
as shown by Landau and Lifshitz [119] and by us elsewhere ([41] and references theirin). Using Equation (Equation 40), we can rewrite this equation as
(142)$$dS=\beta [dE+PdV-V\cdot dP_{p}]$$
in terms of the relative
*velocity*, also known as the *drift velocity*
$V≐V_{p}-V_{\mathrm{gc}}$ of the piston with respect to $\Sigma_{\mathrm{gc}}$. We can cast the drift velocity term as $V\cdot dP_{p}\equiv F_{p}\cdot dR$, where $F_{p}≐dP_{p}/dt$ is the *force* (a macroforce in nature) and dR=Vdt is the *relative displacement* of the piston.

The internal motions of $\Sigma_{\mathrm{gc}}$ and $\Sigma_{p}$ is not controlled by any external agent so the relative motion described by the relative displacement R represents an *internal variable* [12] so that the corresponding affinity $F_{p0}=0$ for $\tilde{\Sigma }$. Because of this, the first law $dE=T_{0}d_{e}S-P_{0}dV$ as given in Equation ([Disp-formula FD23b-entropy-23-01584]) does not involve the relative displacement R. We now support this claim using our approach in the following. This also shows how $H(\left.x\right|V,P_{\mathrm{gc}},P_{p})$ develops a dependence on the internal variable R. We manipulate dS in Equation (Equation 142) by using the above first law for dE so that
$$TdS=T_{0}d_{e}S+\left(P-P_{0}\right)dV-F_{p}\cdot dR,$$
which reduces to
$$T_{0}d_{i}S=\left(T_{0}-T\right)dS+\left(P-P_{0}\right)dV-F_{p}\cdot dR.$$

This equation expresses the irreversible entropy generation as sum of three distinct and independent irreversible entropy generations. To comply with the second law, we conclude that for $T_{0}>0$,
(143)$$\left(T_{0}-T\right)dS\geq 0,\left(P-P_{0}\right)dV\geq 0,F_{p}\cdot dR\leq 0,$$
which shows that each of the components of $d_{i}S$ is nonnegative. In equilibrium, each irreversible component vanishes, which happens when
(144)$$T\to T_{0},P\to P_{0},\;\mathrm{and}\;V\to 0\;\mathrm{or}\;F_{p}\to 0.$$

The inequality $F_{p}\cdot dR\leq 0$ shows that $F_{p}$ and dR are antiparallel, which is what is expected of a *frictional* force $F_{\mathrm{fr}}$. Thus, we can identify $F_{p}$ with $F_{\mathrm{fr}}$, and the irreversible frictional macrowork $d_{i}W_{\mathrm{fr}}$ done during this motion by
(145)$$d_{i}W_{\mathrm{fr}}≐-V\left(t\right)\cdot dP\left(t\right)=-dR\left(t\right)\cdot F_{\mathrm{fr}}\left(t\right)>0.$$

This macrowork, which appears as part of $d_{i}W$ in our approach, vanishes as the motion ceases so that the equilibrium value $V_{0}$ of V(t) is $V_{0}=0$ just as the equilibrium affinity $A_{0}=0$ for ξ. This causes the piston to finally come to rest. As $F_{\mathrm{fr}}$ and V vanish together, we can express this force as
(146)$$F_{\mathrm{fr}}=-\mu Vf\left(V^{2}\right),$$
where μ>0 and *f* is an *even* function of V. The medium $\tilde{\Sigma }$ is specified by $T=T_{0},P=P_{0}$ and $V_{0}=0$ or $F_{p}=0$. We will take $F_{\mathrm{fr}}$ and dR to be colinear and replace $F_{\mathrm{fr}}\cdot dR$ by $-F_{\mathrm{fr}}dx$ ($F_{\mathrm{fr}}dx\geq 0$), where dx is the magnitude of the relative displacement dR. The sign convention is that $F_{\mathrm{fr}}$ and increasing *x* point in the same direction. We can invert Equation (Equation 142) to obtain
(147)$$dE=TdS-PdV-F_{\mathrm{fr}}dx$$
in which dQ=TdS from our general result in Equation ([Disp-formula FD5a-entropy-23-01584]). Comparing the above equation with the first law in Equation ([Disp-formula FD23a-entropy-23-01584]), we conclude that
(148)$$dW=PdV+F_{\mathrm{fr}}dx.$$

The important point to note is that the friction term $F_{\mathrm{fr}}dx$ properly belongs to dW. As $d_{e}W=P_{0}dV$, we have
(149)$$d_{i}W=\left(P-P_{0}\right)dV+F_{\mathrm{fr}}dx.$$

Both contributions in $d_{i}W$ are separately nonnegative.

We can determine the exchange macroheat $d_{e}Q=dQ-d_{i}W$
(150)$$d_{e}Q=TdS-\left(P-P_{0}\right)dV-F_{\mathrm{fr}}dx$$

It should be emphasized that in the above discussion, we have not considered any other internal motion such as between different parts of the gas besides the relative motion between $\Sigma_{\mathrm{gc}}$ and $\Sigma_{p}$. These internal motions within $\Sigma_{g}$ can be considered by following the approach outlined elsewhere [41]. We will not consider such a complication here.

### 13.2. Particle-Spring-Fluid System

It should be evident that by treating the piston as a mesoscopic particle such as a pollen or a colloid, we can treat its thermodynamics using the above procedure. This allows us to finally make a connection with the system depicted in Figure 4b in which the particle (a pollen or a colloid) is manipulated by an external force $F_{0}$. We need to also consider two additional forces $F_{s}$ and $F_{\mathrm{fr}}$, both pointing in the same direction as increasing *x*; the latter is the frictional force induced by the presence of the fluid in which the particle is moving around. The analog of Equation (Equation 149) for this case becomes
(151)$$d_{i}W=\left(F_{s}+F_{0}\right)dx+F_{\mathrm{fr}}dx=F_{t}dx,$$
where $F_{t}=F_{s}+F_{0}+F_{\mathrm{fr}}$ is the net force. The other two macroworks are $dW=(F_{s}+F_{\mathrm{fr}})dx$ and $d\tilde{W}=F_{0}dx=-d_{e}W$. In EQ, $F_{\mathrm{fr}}=0$ and $F_{s}+F_{0}=0$ ($F_{0}\neq 0$) to ensure $d_{i}W=0$.

### 13.3. Particle-Fluid System

In the absence of a spring in the previous subsection, we must set $F_{s}=0$ so
(152)$$dW=F_{\mathrm{fr}}dx,d\tilde{W}=F_{0}dx=-d_{e}W,d_{i}W=\left(F_{0}+F_{\mathrm{fr}}\right)dx.$$

In EQ, $F_{0}+F_{\mathrm{fr}}=0$ so that $F_{\mathrm{fr}}=-F_{0}$. This means that in EQ, the particle’s nonzero terminal velocity is determined by $F_{0}$ as expected.

## 14. Free Expansion

What makes NEQ thermodynamics complicated than EQ thermodynamics is the evaluation of nonzero irreversible entropy generation $d_{i}S\left(t\right)\geq 0$. As $d_{i}S^{Q}\left(t\right)$ and $d_{i}S^{W}\left(t\right)$ are independent contributions, it is simpler to consider an isolated system for which $d_{i}S^{Q}\left(t\right)\equiv 0$ so that we only deal with $d_{i}S^{W}\left(t\right)$ in Equation (Equation 87). Then the use of Equation (Equation 81) allows us to determine the temperature of the system in any arbitrary macrostate. In free expansion, there is no exchange of any kind so $d=d_{i}$. This simplifies our notation as we do not need to use $d_{i}$ when referring to Σ, which we will do in this section.

### 14.1. Classical Free Expansion in $S_{Z}$

The gas, which forms Σ, expands freely in a vacuum ($\tilde{\Sigma }$) from $V_{\mathrm{in}}$, the volume of the left chamber, to $V_{\mathrm{fin}}=2V_{\mathrm{in}}$, the volume of $\Sigma_{0}$; the volume of the right chamber is $V_{\mathrm{fin}}-V_{\mathrm{in}}=V_{\mathrm{in}}$. The initial and final macrostate are denoted by $A_{\mathrm{eq}}$ and $B_{\mathrm{eq}}$. The vacuum exerts no pressure ($\tilde{P}=P_{\mathrm{vacuum}}=0$). The left (L) and right (R) chambers are initially separated by an impenetrable partition, shown by the solid partition in Figure 6a, to ensure that they are thermodynamically independent regions, with all the *N* particles of Σ in the left chamber, which are initially in an EQ macrostate $A_{\mathrm{eq}}$ with entropy $S_{\mathrm{in}}$. For ideal gas, we have [14]
S(E,V)=Nln(eV/N)+f(E),
where *N* is kept as a suffix for a reason that will become evident below. The initial pressure and temperature of the gas prior to expansion in $A_{\mathrm{eq}}$ at time t=0 are $P_{\mathrm{in}}$ and $T_{\mathrm{in}}=T_{0}$, respectively, that are related to $E_{0}=E_{\mathrm{in}}$ and $V_{\mathrm{in}}$ by its EQ equation of state. A similar set of quantities also pertain to $B_{\mathrm{eq}}$. As $\Sigma_{0}$ is isolated, the expansion occurs at *constant* energy $E_{0}$, which is also the energy of Σ.

It should be stated, which is also evident from Figure 6b, that while the removal of the partition is instantaneous, the actual process of gas expanding in the right chamber is continuous and gradually fills it. This is obviously a very complex internal process in a highly inhomogeneous macrostate. As thus, it will require many internal variables to describe different number of particles, different energies, different pressures, different flow pattern which may be even chaotic, etc. in each of the chambers. For example, we can divide the volume $V_{\mathrm{fin}}$ into many layers of volume parallel to the partition, each layer in equilibrium with itself but need not be with others; see the example in Section 3. As our aim is to show the feasibility of the MNEQT in this investigation, we will simplify the situation by limiting to two internal variables. The first internal variable $\xi_{N}$
(153)$$\xi_{N}≐N_{R}/N$$
is obtained by considering only two layers to describe different numbers $N_{L}=\left(1-\xi_{N}\right)N$ particles to left and $N_{R}=N\xi_{N}$ particles to the right of the chamber partition in Figure 6a as a function of time. Initially, $\xi_{N}=0$ and finally at EQ, $\xi_{N}=1/2$. At each instant, we imagine a front of the expanding gas shown by the solid vertical line in Figure 6b containing all the particles to its left. We denote this volume by a time-dependent V=V(t) to the right of which exists a vacuum. This means that at each instant when there is a vacuum to the right of this front, the gas is expanding against zero pressure so that $d_{e}W=0$. Since we have a NEQ expansion, dW>0. As V(t) cannot be controlled externally, it can be used to determine another internal variable by using $V^{\prime }=V-V_{\mathrm{in}}$:$$\xi_{V}≐V^{\prime }/V=1-V_{\mathrm{in}}/V,$$
so that $V^{\prime }=\xi_{V}V$ and $V_{\mathrm{in}}=\left(1-\xi_{V}\right)V$. Initially, $\xi_{V}=0$ and finally at EQ, $\xi_{V}=1/2$. The choice of the two internal variables $\xi_{N}\left(t\right)$ and $\xi_{V}\left(t\right)$ follows the procedure in Section 11 for two subsystems of different sizes, and allow us to distinguish between $P^{\prime }$ and $P^{\prime \prime }$ as we will see below. We assume that the expansion is isothermal (which it need not be) so there is no additional internal variable associated with temperature variation. As dQ=dW>0, the expansion is irreversible so the entropy continues to increase.

At t=0, the partition is suddenly removed, shown by the broken partition in Figure 6b and the gas expands freely to the final volume $V\left(t^{\prime }\right)=V_{\mathrm{fin}}$ at time $t^{\prime }<\tau_{\mathrm{eq}}$ during $P^{\prime }$. At $t^{\prime }$, the free expansion stops but there is no reason a priori for $\xi_{N}=0$ so the gas is still inhomogeneous ($\xi_{N}\neq 0$). This is in a NEQ macrostate until $\xi_{N}$ achieves its EQ value $\xi_{N}=0$ during $P^{\prime \prime }$, at the end of which at $t=\tau_{\mathrm{eq}}$ the gas eventually comes into $B_{\mathrm{eq}}$ isoenergetically. The complete process is $\bar{P}=P^{\prime }\cup P^{\prime \prime }$ between $A_{\mathrm{eq}}$ and $B_{\mathrm{eq}}$. We briefly review this expansion in the MNEQT [92].

We work in the extended state space with the two internal variables, which we denote simply by S here. Using Equation ([Disp-formula FD23a-entropy-23-01584]), we have
(154a)dS(t)=dW(t)/T(t).

Setting $P_{0}=0$ in Equation ([Disp-formula FD32b-entropy-23-01584]), we have
(154b)$$dW\left(t\right)=\left\{\begin{array}{c} P\left(t\right)dV\left(t\right)+A\left(t\right)\cdot d\xi \left(t\right)\;\mathrm{for}\;t<t^{\prime }<\tau_{\mathrm{eq}},\\ A\left(t\right)d\xi \left(t\right)\;\mathrm{for}\;t^{\prime }<t\leq \tau_{\mathrm{eq}}; \end{array}\right.$$
here, we have used the fact that V(t) does not change for $\tau^{\prime }<t\leq \tau_{\mathrm{eq}}$. Thus,
$$\begin{array}{cc} \Delta S & =\int_{\bar{P}}\frac{dW(t)+A(t)\cdot d\xi (t)}{T(t)}>0,\\ \Delta Q & =\int_{\bar{P}}dW\left(t\right)=\Delta W>0; \end{array}$$
the last equation is the fundamental identity in Equation (Equation 22). The irreversible entropy change ΔS from EQ macrostate from $A_{\mathrm{eq}}$ to $B_{\mathrm{eq}}$ is the EQ entropy change $\Delta_{i}S$ is
(155)$$\Delta S\equiv S_{\mathrm{fin}}-S_{\mathrm{in}},$$
and can be directly obtained since the EQ entropy S(E,V) is known. The above analysis is also valid for any arbitrary free expansion process P and not just $\bar{P}$ as we have not used any information yet about $A_{\mathrm{eq}}$ to $B_{\mathrm{eq}}$.

For $V_{\mathrm{fin}}=2V_{\mathrm{in}}$, $\Delta_{i}S=N\ln2$, a well-known result [16]. Here, we provide a more general result for the entropy for $t\leq t^{\prime }$, which can be trivially determined:$$S\left(\xi_{V},\xi_{N}\right)=N_{L}\ln\left(eV_{\mathrm{in}}/N_{L}\right)+N_{R}\ln\left(eV^{\prime }/N_{R}\right)+f\left(E_{0}\right).$$

Thus, for arbitrary $\xi_{V}$ and $\xi_{N}$, we have $\Delta_{i}S\left(\xi_{V},\xi_{N}\right)=S\left(\xi_{V},\xi_{N}\right)-S_{\mathrm{in}}$. We can determine the two affinities. A simple calculation gives
$$\begin{array}{cc} A_{V}/T & =\partial S/\partial \xi_{V}=1-\xi_{N}-\xi_{V},\\ A_{N}/T & =\partial S/\partial \xi_{N}=\ln\frac{\left(1-\xi_{N}\right)\xi_{V}}{\left(1-\xi_{V}\right)\xi_{N}}. \end{array}$$

We see that $A_{V}$ does not vanish when $V^{\prime }=V_{\mathrm{fin}}$ as discussed above. It is easy to verify that A vanish in $B_{\mathrm{eq}}$. The pressure of the expanding gas is obtained by using the derivative ∂S/∂V as usual. A simple calculation yields
(156)$$\begin{array}{cc} \beta P & =\left(1-\xi_{V}\right)\frac{N_{L}}{V_{\mathrm{in}}}+\xi_{V}\frac{N_{R}}{V^{\prime }}\\ \; & =\left(1-\xi_{V}\right)\beta P_{L}+\xi_{V}\beta P_{R}. \end{array}$$

The last expression for pressure has a close similarity with the Toll-Narayanaswamy Equation (Equation 133), which should not be surprising. Before expansion, we have $\beta P_{\mathrm{in}}=N/V_{\mathrm{in}}$ in $A_{\mathrm{eq}}$ and $\beta P_{\mathrm{fin}}=N/V_{\mathrm{fin}}$ in $B_{\mathrm{eq}}$ as expected. At EQ in $B_{\mathrm{eq}}$, the entropy is given by $S_{\mathrm{fin}}=N\ln\left(2eV_{\mathrm{in}}/N\right)$, which gives $\Delta_{i}S=N\ln2$, as expected. We can also take the initial macrostate to be not an EQ one in P by using one or more additional internal variables. Thus, the approach is very general.

### 14.2. Quantum Free Expansion

The sudden expansion has been studied [110,111,113] quantum mechanically (without any $\xi_{N}$) as a particle in an isolated box $\Sigma_{0}$ of length $L_{\mathrm{fin}}$, which we restrict to $2L_{\mathrm{in}}$ here, with rigid, insulating walls. We briefly revisit this study and expand on it by introducing a $\xi_{N}$ to parallel the study of the classical expansion above but using the μNEQT. Thus, we will closely follow the microstates and follow Ref. [111] closely.

We make the very simplifying assumptions in the previous section to introduce $\xi_{N}$. At time t=0, all the *N* particles (or their wavefunctions) are confined in EQ in the left chamber of length $L_{\mathrm{in}}$ so that $N_{L}=N$ initially. We can think of an intermediate length $L_{\mathrm{fin}}\geq L\left(t\right)>L_{\mathrm{in}}$, in analogy with V(t) in the previous section, so that $N_{R}=N-N_{L}$ particles are simultaneously confined in the intermediate chamber of size L(t), while $N_{L}$ particles are still confined in the left chamber for all t>0. This is slightly different from what we did in the previous section. Eventually, at $t=\tau_{\mathrm{eq}}$, all the $N_{R}=N$ particles are confined in the larger chamber of size $L_{\mathrm{fin}}$ so that there are no particles are confined in the initial chamber. We let $\xi_{N}=N_{R}/N$, which gradually increases from $\xi_{N}=0$ to $\xi_{N}=1$. Note that this definition is different from the previous section but we make this choice for the sake of simplicity. At some intermediate time $\tau^{\prime }<\tau_{\mathrm{eq}}$ that identifies $P^{\prime }$, $L\left(t\right)=L_{\mathrm{fin}}$, but $N_{R}$ is still not equal to *N* ($\xi_{N}\neq 0$). We then follow its equilibration during $P^{\prime \prime }$ as the gas come to EQ in the larger chamber at the end of $\bar{P}$ when $\xi_{N}=1$. Again, there are two internal variables *L* and $\xi_{N}$. The expansion is isoenergetic at each instant. As we will see below, this means that it is also isothermal. However, dQ=dW≠0 ensuring a irreversible process so the microstate probabilities continue to change.

Since we are dealing with an ideal gas, we can focus on a single particle whose energy levels are in appropriate units $E_{k}=k^{2}/l^{2}$, where *l* is the length of the chamber confining it. The single-particle partition function for arbitrary *l* and inverse temperature β=1/T is given by
$$Z\left(\beta ,l\right)=\sum_{k}e^{-\beta E_{k}\left(l\right)},$$
from which we find that the single particle free energy is $\bar{F}=-\left(T/2\right)\ln\left(\pi Tl^{2}/4\right)$ and the average single particle energy is E=1/2β, which depends only on β but not on *l*. Assuming that the gas is in IEQ so that the particles in each of the two chambers are in EQ (see the second example in Section 3) at inverse temperatures $\beta_{L}$ and β, we find that the *N*-particle partition function is given by
$$Z_{N}\left(\beta_{L},\beta \right)={\left[Z(\beta_{L},L_{\mathrm{in}})\right]}^{N(1-\xi_{N})}{\left[Z(\beta ,L)\right]}^{N\xi_{N}}$$
so that the average energy is $E_{N}\left(\beta_{L},\beta ,L_{\mathrm{in}},L,\xi_{N}\right)=N\left(1-\xi_{N}\right)/2\beta_{L}+N\xi_{N}/2\beta$. As this must equal $N/2\beta_{0}$ for all values of *L* and $\xi_{N}$, it is clear that $\beta_{L}=\beta =\beta_{0}$, which proves the above assertion of an isothermal free expansion at $T_{0}$.

To determine $\Delta W_{k}$, we merely have to determine the microenergy change $\Delta E_{k}=E_{k,\mathrm{fin}}-E_{k,\mathrm{in}}$ [54,111].

Below we will show that the quantum calculation here deals with an irreversible $\bar{P}$. The single-particle energy change $\Delta E_{k}$ is
$$\Delta E_{k}=k^{2}\left(1/L^{2}-1/L_{\mathrm{in}}^{2}\right)<0,L>L_{\mathrm{in}}.$$

The micropressure
(157)$$P_{k}=-\partial E_{k}/\partial L=2E_{k}/L\neq 0$$
determines the microwork
(158)$$\Delta W_{k}=\int_{L_{\mathrm{in}}}^{L_{\mathrm{fin}}}P_{k}dL>0.$$

It is easy to see that this microwork is precisely equal to $\left(-\Delta E_{k}\right)$ as expected. It is also evident from Equation (Equation 157) that for each *L* between $L_{\mathrm{in}}$ and $L_{\mathrm{fin}}$,
$$P=\sum_{k}p_{k}P_{k}=2E/L\neq 0,$$

We can use this average pressure to calculate the thermodynamic macrowork
$$\Delta W=\int_{L_{\mathrm{in}}}^{L_{\mathrm{fin}}}PdL=2\sum_{k}\int_{L_{\mathrm{in}}}^{L_{\mathrm{fin}}}p_{k}E_{k}dL/L\neq 0.$$
as expected. As ΔE=0, this means that the irreversible macroheat and macrowork are nonnegative and equal: ΔQ=ΔW>0. This establishes that the expansion we are studying is *irreversible*.

We now turn to the entire system in which the work is done by $N_{R}$ particles occupying the larger box. We need to think of the microstate index *k* as an *N*-component vector $k=\left\{k_{i}\right\}$ denoting the indices for the single-particle microstates. For a given $\xi_{N}$, we have $\Delta W_{k}\left(L,\xi_{N}\right)=-\sum_{i}\Delta E_{k_{i}}$, where *i* runs over the $N_{R}$ particles. We can compute the macrowork, which turns out to be $\Delta W_{N}\left(\xi_{N}\right)=N\xi_{N}\Delta W>0$. The corresponding change in the free energy is
$$\begin{array}{cc} \Delta {\bar{F}}_{N}\left(L,\xi_{N}\right) & =N\xi_{N}\left[\bar{F}\left(\beta_{0},L\right)-\bar{F}\left(\beta_{0},L_{\mathrm{in}}\right)\right]\\ \; & =-\Delta W_{N}\left(\xi_{N}\right), \end{array}$$
which is consistent with Equation ([Disp-formula FD84a-entropy-23-01584]) for an isolated system for any $\xi_{N}$.

At the end of $P_{0}$, $\Delta W_{N}\left(0\right)=N\Delta W>0$, and $\Delta {\bar{F}}_{N}\left(0\right)=N\left[\bar{F}\left(\beta_{0},L_{\mathrm{fin}}\right)-\bar{F}\left(\beta_{0},L_{\mathrm{in}}\right)\right]$. We find that for the isothermal expansion
(159)$$\Delta W_{N}=-\Delta {\bar{F}}_{N}=T_{0}\Delta_{i}S_{N}>0.$$
after using Equation (Equation 81). The same result is also obtained from the classical isothermal expansion; see Equation ([Disp-formula FD84b-entropy-23-01584]). All this is in accordance with Theorem 7 in the MNEQT, as expected.

## 15. Discussion and Conclusions

As we noted in Section 1, thermodynamics is a science of entropy and temperature. As these macroquantities should uniquely describe the system, we have required them to SI-quantities in developing the new NEQT, called the MNEQT, to go beyond the EQ thermodynamics. We will now briefly summarize and discuss our conclusions form this thermodynamics. We will consider them separately.

### 15.1. Unique NEQ *S* in $S_{Z}$


We first point out the important consequence of the restriction imposed by quasi-independence discussed in Section 2 and Section 6.4. By always dealing with the SI-entropy *S*, which we have shown to be identical to the statistical quantity S in all cases, we can appreciate the concept of quasi-independence by considering $p_{k}$ that appears in Equation (Equation 47). Considering Σ to consist of two subsystems $\Sigma_{1}$ and $\Sigma_{2}$, which are in macrostates $M_{1}≐\left\{m_{k_{1}},p_{k_{1}}\right\}$ and $M_{2}≐\left\{m_{k_{2}},p_{k_{2}}\right\}$. If $M_{1}$ and $M_{2}$ are quasi-independent and form M for Σ, then
$$p_{k}≃p_{k_{1}}p_{k_{2}}.$$

As a consequence, the entropy additivity
$$S\left(X\left(t\right),t\right)≃S_{1}\left(X_{1}\left(t\right),t\right)+S_{2}\left(X_{2}\left(t\right),t\right)$$
is approximately satisfied. This has a generalization to many subsystems $\left\{\Sigma_{i}\right\}$ given in Equations ([Disp-formula FD74a-entropy-23-01584])–([Disp-formula FD74c-entropy-23-01584]) so that they are *all* quasi-independent. In terms of the volume $\Delta V_{i}$ of $\Sigma_{i}$ so that $V=\sum_{i}\Delta V_{i}$, the generalization can be simply written in the form of the entropy additivity requirement over $\Delta V_{i}$
(160a)$$S\left(X\left(t\right),t\right)=\sum_{i}S_{i}\left(X_{i}\left(t\right),t\right),$$
in accordance with quasi-independence. The requirement of quasi-independence forces the linear size $\Delta l_{i}$ of $\Sigma_{i}$ to be not less than the correlation length $\lambda_{\mathrm{corr}}$ as discussed in Section 2 and Section 6.4; see also [41]. Thus, there will be no nonlocal effects ([120,121] (for example)) to consider in the MNEQT as they are subsumed within each subsystem. Each subsystem has its own Hamiltonian H containing all the information regarding interactions between its constituent particles and internal variables (see how H in Section ) so its microstates will contain the effects of all the interactions in $\left\{E_{k}\right\}$.

By a proper choice of $S_{Z}$, S(X(t),t) can be replaced by a unique state function S(Z(t)). Similarly, by a proper choice of $S_{Z_{i}}$, a subspace of $S_{Z}$, $S_{i}\left(X_{i}\left(t\right),t\right)$ can be replaced by a state function $S_{i}\left(Z_{i}\left(t\right)\right)$ in $S_{Z_{i}}$. By matching the number of independent variables $n^{*}$ on both sides in Equation ([Disp-formula FD160a-entropy-23-01584]) as discussed in Section 6.4, we ensures that S(Z(t)) is uniquely determined as a sum of $S_{i}\left(Z_{i}\left(t\right)\right)$ in accordance with Equation ([Disp-formula FD74c-entropy-23-01584]). By replacing $S_{Z_{i}}$ by $S_{X},\forall i$, and using Theorem 6 and Corollary 1, we know that S(Z(t)) uniquely exists in the MNEQT so there is no freedom to choose any other variables on which S(Z(t)) can depend on. But the actual choice of $n<<n^{*}$ for a given $M_{\mathrm{ieq}}$ is determined by the experimental setup. It is this $n>n_{\mathrm{obs}}$ that is physically relevant for $M_{\mathrm{ieq}}$ unless we are dealing with a $M_{\mathrm{eq}}$, where $n_{\mathrm{obs}}$ is the number of independent variables in X. The remaining $n^{*}-n$ internal variables have equilibrated so their affinities vanish.

We turn back to S(X(t),t) and $S_{i}\left(X_{i}\left(t\right),t\right)$. If $\Delta l_{i}$ is less than the correlation length, then Equation ([Disp-formula FD160a-entropy-23-01584]) must be replaced by
(160b)$$S\left(X\left(t\right),t\right)=\sum_{i}S_{i}\left(X_{i}\left(t\right),t\right)+S_{\mathrm{corr}}\left(X\left(t\right),t\right)$$
due to the correlations present among various $\Sigma_{i}$’s; cf. Equation (Equation 2). In a continuum NEQT introduced as CNEQT in Section 6.3, we are in the limit of "infinitesimal volume element" over which $S_{i}\left(X_{i}\left(t\right),t\right)$ can be expressed as s(r∣x(r,t),t)dr; here, x(r,t) is the local analog of $X_{i}\left(t\right)$ over this volume element. In this limit, Equation ([Disp-formula FD160b-entropy-23-01584]) reduces to
(160c)$$S\left(X\left(t\right),t\right)=\int s\left(r|x\left(r,t\right),t\right)dr+S_{\mathrm{corr}}\left(X\left(t\right),t\right),$$
where $S_{\mathrm{corr}}$ cannot be converted to a volume integral as it is a nonlocal quantity over at least the correlation length. Unfortunately, $S_{\mathrm{corr}}$ is almost invariably overlooked in the CNEQT [20,40], which allows the function s(r∣x(r,t),t) to be commonly identified as the entropy density. This is most probably a misleading nomenclature as $S_{\mathrm{corr}}\left(X\left(t\right),t\right)$ has been neglected. Even in EQ, the correct entropy density should be precisely S(X(t),t)/V(t), which is not s(r∣x(r,t),t). With this approximation, the SI-entropy is replaced by $S_{\mathrm{CT}}\left(X\left(t\right),t\right)$ given by the integral on the right side (CT for the CNEQT),
(160d)$$S\left(X\left(t\right),t\right)\overset{?}{≃}S_{\mathrm{CT}}\left(X\left(t\right),t\right)=\int s\left(r|x\left(r,t\right),t\right)dr;$$
the questionmark is because it is hard to estimate the error due to the neglect of $S_{\mathrm{corr}}$. Thus, the additivity of *s* in the integral is not the postulated additivity of *S* even in EQ thermodynamics. To ensure that *s* satisfies the second law, it is postulated that *s* shares this property [20]. Because of these issues, we do not focus on *s* in this review as the volume elements are not usually quasi-independent unless we are at high enough temperatures so that the correlation lengths become small enough to make them quasi-independent.

The above limitations also distinguish the MNEQT with all CNEQT theories, which fall under the category of the $\overset{{}^{\circ}}{M}$NEQT. Here, we will briefly comment on two successful theories.

The first one is the extended irreversible thermodynamics (ENEQT) [20], a well-known CNEQT, which also neglects $S_{\mathrm{corr}}$ but treats the corresponding entropy density $s_{\mathrm{ET}\;}$(ET for the ENEQT)as a state function involving various dissipative fluxes such as the heat flux. As said above, one needs to be careful to incorporate nonlocal effects ([120,121], (for example.)) in the CNEQT. In addition, the total entropy $S_{\mathrm{ET}}$ ([20] (see Equation (5.66) and the discussion thereafter)) is also a state function involving the same fluxes for Σ, which violates Corollary 1 about requiring a larger state space relative to $s_{\mathrm{ET}}$: Both $S_{\mathrm{ET}}$ and $s_{\mathrm{ET}}$ cannot have the same state space for the additivity of entropy to be an identity; see also Remark 11. As the fluxes determine MI-macroquantities $d_{e}Q$ and $d_{e}W$, $S_{\mathrm{ET}}$ is not a SI-entropy as *S* is in the MNEQT.

The second one is the MNET [122] that is based on the idea of *internal* dof (dof${}_{\mathrm{in}}$) proposed by Prigogine and Mazur [123] for a Σ in contact with a $\tilde{\Sigma }$. The authors provide a very good comparison of the MNET with other important theories to which we direct the reader. Here, we only compare it with the MNEQT. The emphasis in the MNET is to study slow relaxation in Σ (cf. Section 11) caused by the dof, that we denote here by dof${}_{\mathrm{slow}}$ or by observables $X_{\mathrm{slow}}$, and the corresponding part of Σ by $\Sigma_{\mathrm{slow}}$; the remainder of the system is denoted by $\Sigma_{\mathrm{fast}}$ with observables $X_{\mathrm{fast}}$. In addition to $\tilde{\Sigma }$, $\Sigma_{\mathrm{slow}}$ is also allowed to interact with another work medium, which we denote by ${\tilde{\Sigma }}^{\prime \left(w\right)}$ with an extra prime, with which it exchanges macrowork only; see ([14] (Section 20)). This makes As is well-known from the Tool-Narayanaswamy equation, see Equation (Equation 133), and other works [41,42,43,124,125], $\Sigma_{\mathrm{slow}}$ and $\Sigma_{\mathrm{fast}}$ usually have different temperatures, see Equation (Equation 133), pressures, see for example Equation (Equation 132), etc. and they need not be equal to those of $\tilde{\Sigma }$. This is not considered in the MNET, where it is assumed that $T=T_{0},P=P_{0}$, etc. so Σ is assumed to be in EQ with $\tilde{\Sigma }$, so there cannot be any internal exchanges between $\Sigma_{\mathrm{slow}}$ and $\Sigma_{\mathrm{fast}}$. The main focus in the MNET is only on $\Sigma_{\mathrm{slow}}$ and not the entire Σ. This makes Σ as the realization $\Sigma_{C}$; see Section 10. The entropy in the MNET in our discrete notation is given by
(161)$$S_{\mathrm{MNET}}\left(X,t\right)=S_{\mathrm{eq}}\left(X\right)-H\left(X_{\mathrm{slow}},t\right),$$
where
$$H\left(X_{\mathrm{slow}},t\right)≐\sum_{k}P_{k}\left(t\right)\ln\frac{P_{k}\left(t\right)}{P_{k}^{\mathrm{eq}}}$$
is the net contribution from $X_{\mathrm{slow}}$ ([122] (Equation (Equation 7))); here, $P_{k}\left(t\right)$ is the probability of $m_{k}^{c}$, the microstate in the *internal configurational space* (c) formed by dof${}_{\mathrm{slow}}$ [123], with $X_{\mathrm{slow}k}$ denoting its configuration.As $\Sigma_{\mathrm{slow}}$ equilibrates with ${\tilde{\Sigma }}^{\prime }$, $P_{k}\left(t\right)\to P_{k}^{\mathrm{eq}}$ so that H(t)→0. Consequently, $S_{\mathrm{MNET}}\left(X,t\right)\to S_{\mathrm{eq}}\left(X\right)$ so that $S_{\mathrm{MNET}}\left(X,t\right)-S_{\mathrm{eq}}\left(X\right)=-H\left(X_{\mathrm{slow}},t\right)\;$is the contribution from the NEQ dof${}_{\mathrm{slow}}$ [126]. The presence of $P_{k}^{\mathrm{eq}}$, which surely depends on the conjugate fields of the medium ${\tilde{\Sigma }}_{\mathrm{slow}}$ controlling $X_{\mathrm{slow}}$, makes $S_{\mathrm{MNET}}\left(X,t\right)$ an MI-quantity. Thus, it is different from our SI-entropy S(X,t). Moreover, as $\Sigma_{\mathrm{slow}}$ interacts with ${\tilde{\Sigma }}^{\prime \left(w\right)}$, there is an exchange work $\Delta {\tilde{W}}^{\prime }=-\Delta_{e}W_{\mathrm{slow}}$ done by ${\tilde{\Sigma }}^{\prime \left(w\right)}$ [14]. As $X_{\mathrm{slow}}$ in the MNET is controlled externally, it does not represent an internal variable in the sense used in the MNEQT, which explains the use of $X_{\mathrm{slow}}$ and not ξ to represent dof${}_{\mathrm{slow}}$. This is also consistent with Conclusion 11 since there is no need to consider any internal variable for the realization $\Sigma_{C}$. This is further clarified in the next paragraph. It is easy to see that $\Delta_{e}W_{\mathrm{slow}}$ satisfies Equation ([Disp-formula FD23b-entropy-23-01584]):(162)$$\Delta E=T_{0}\Delta_{e}S-P_{0}\Delta V-\Delta_{e}W_{\mathrm{slow}};$$
see ([14] (see the derivation leading to Equation (20.1))). Thus, MNET belongs to the $\overset{{}^{\circ}}{M}$NEQT as pointed out above. A configuration temperature for $m_{k}^{c}$ is also introduced in the MNET by using $s_{ck}=-\ln P_{k}\left(t\right)$, which is not considered in the MNEQT, where only a global thermodynamic temperature is defined.

As the examples in Section 10.3 have revealed, we can treat Σ either as $\Sigma_{B}$ or $\Sigma_{C}$. We need internal variables to specify $\Sigma_{B}$ that help to describe whatever is going on within Σ without knowing these processes. While we do not need internal variables to specify $\Sigma_{C}$, we need to know internal processes such as the internal transfer $dE_{\mathrm{in}}\left(t\right)=d_{e}Q_{\mathrm{in}}\left(t\right)$. Both realizations are equivalent in the MNEQT. As the entropy is a unique function in $S_{Z}$, there is no room for any extra dependence such as external fluxes in either realization; see Theorem 6. The internal fluxes such as $dE_{\mathrm{in}}\left(t\right)$ are needed for $\Sigma_{C}$, but they are not controlled by the medium (they are present even if Σ is isolated). Thus, the MNEQT always deals with a SI-entropy.

Thus, the two entropies, $S_{\mathrm{ET}}$ and *S*, are very distinct in many ways, and cannot be compared as their predictions will be very different.

We now turn to the significance of the new NEQT (MNEQT) in the enlarged state space, which is a SI-thermodynamics. As macroheat and macrowork are two independent quantities in the MNEQT, it is clear that the notion of temperature can be understood by merely focusing of the relationship between dQ and dS for any arbitrary process; dW plays no role in it. This is what makes the MNEQT a very useful thermodynamic approach. It should be stressed that the generalized macrowork dW (the generalized macroheat dQ) is not the same as the exchanged macrowork $d_{e}W$ (exchanged macroheat $d_{e}Q$) with the medium unless $d_{i}W=0$ ($d_{i}Q=0$), i.e., unless there is no *internal irreversibility* caused by internal processes.

Thus, any deviation of dW from $d_{e}W$ or dQ from $d_{e}Q$ in a process is the result of irreversibility due to internal processes alone. Indeed, $d_{i}W$ is the macrowork done internally by the system against all *dissipative forces within the system*, see Equation (Equation 82), which explains why $d_{i}W$ is a measure of dissipative irreversibility (Definition 17) within the system. In a similar manner, $d_{i}Q$ is the macroheat generated internally by the system, which from $d_{i}Q=d_{i}W$ is also due to all dissipative forces within the system. It must be emphasized that irreversible macroheat transfer due to temperature difference between a system and a medium does not affect $d_{i}W=d_{i}Q$; see Equation (Equation 86). On the other hand, irreversible macroheat transfer between different *internal* parts of Σ will be part of $d_{i}W=d_{i}Q$ as seen from the discussion of $\Sigma_{C}$ in Section 10.

The use of SI-quantities to specify a macrostate and microstates of a system allows us to determine a statistical definition of the entropy of any $M_{\mathrm{arb}}$ in Section 4, which is based on the ideas of Boltzmann. Using the flatness hypothesis, see Remark 5, known to be valid for macroscopic systems, we provide a simple proof of the second law in Section 4.3; see also [53] (Theorem 4).

### 15.2. Unique NEQ *T* in $S_{Z}$

As dQ and dS are SI-macroquantities, their extensivity requires a linear relation between them for any $M_{\mathrm{arb}}$ as discussed in Section 5.2; see Equation (Equation 62). The proportionality parameter is identified as the temperature *T* in Equation (Equation 104). With this extension to deal with $M_{\mathrm{nieq}}$ in $S_{Z^{\prime }}⊃S_{Z}$, the definition is applicable to any $M_{\mathrm{arb}}$. Thus, there is no need to differentiate between $T_{\mathrm{arb}}$ and *T* in the MNEQT as said earlier. The same definition also applies to an isolated system in an arbitrary macrostate. Determining other fields related to dW such as the pressure do not pose any new complications in the new approach as they are mechanical in nature as discussed in Section 6.2. We have thus proposed a novel approach to define the unique temperature (see Theorem 6) that is applicable to any $M_{\mathrm{arb}}$ by selecting the particular $S_{Z^{\prime }}$ where $M_{\mathrm{nieq}}$ is convert to $M_{\mathrm{ieq}}$. Then the changes in $\left\{dp_{k}\right\}$ and $\left\{E_{k}\right\}$ identify dS,dQ and dW for given *T* and W in $S_{Z^{\prime }}$. All these SI-macroquantities have the same values also in $S_{Z}$ for $M_{\mathrm{nieq}}$. All of this strongly supports Proposition 2 as the fundamental axiom of the MNEQT that can explain the behavior of any $M_{\mathrm{arb}}$.

It should be pointed out that the statistical definition of temperature in Equations ([Disp-formula FD5a-entropy-23-01584]) or (Equation 81) is not limited to extensive systems only. The discussion and the conclusions are also valid for systems for which dQ and dS scale the same way with *N*. We have only considered a linear scaling between the two SI-macroquantities in this work. It should be pointed out that our concept of temperature has some similarity with the idea of a contact temperature for a system in thermal contact with a medium. The latter is introduced [23,100,127] by the inequality in Equation (Equation 88) for a system in thermal contact with a medium (but not when the system is isolated). We, instead, define the temperature as an equality in Equation ([Disp-formula FD5a-entropy-23-01584]) for any arbitrary macrostate, which works even for an isolated system. Equation (Equation 88) is a consequence of our definition.

We have seen that this definition of *T* satisfies the four requirements, see Criterion 5, listed in Section 6. This thus solves the dreams of Planck and Landau [97,98]. For example, we need to ensure that *macroheat*$d_{e}Q$*, if it is transferred at different temperatures, always flows from hot to cold*. Indeed, this is a fundamental requirement for a consistent notion of temperature due to the second law; see Criterion C3. To the best of our knowledge, this question has not been answered satisfactorily [93,99,100,103,104,105] for an arbitrary nonequilibrium macrostate. The question is not purely academic as it arises in various contexts of current interest in applying nonequilibrium thermodynamics to various fields such as the Szilard engine [128,129,130,131], Jarzynski process [34], stochastic thermodynamics [36], Maxwell’s demon [132,133], thermogalvanic cells, corrosion, chemical reactions, biological systems [134,135,136], etc. to name a few. Our approach thus finally *solves* the long-standing unsolved problem of defining the temperature for an arbitrary macrostate in a consistent way that satisfy the stringent criteria C1–C4 as proven in Theorem 9. Thus, *T* must be treated as a genuine unique temperature of the system in any macrostate M.

Our definition of the temperature in Equations ([Disp-formula FD5a-entropy-23-01584]) and (Equation 81) introduces T(t) as a global quantity, see C4 in Criterion 5, for the entire system and should not be confused as a local quantity, which varies from region to region within the system. This is true even if the system is inhomogeneous. Recall that we have not imposed any requirement for the system to be homogeneous in our discussion in Section 6. One may wonder if it makes any sense to call T(t) the temperature of the system even if it is inhomogeneous. It is possible to think of an inhomogeneous system to be composed of a number of homogeneous subsystems $\Sigma_{1},\Sigma_{2},\cdots$, each macroscopic in its own right. In that case, we can assign a temperature $T_{1}\left(t\right),T_{2}\left(t\right),\cdots$ to $\Sigma_{1},\Sigma_{2},\cdots$, respectively. It is then possible to relate T(t) to $T_{1}\left(t\right),T_{2}\left(t\right),\cdots$. We have explicitly shown this here by considering only two subsystems in Section 10 when they are of identical sizes, and Section 11 when they are of different sizes, and treating the system as $\Sigma_{B}$; see Equations ([Disp-formula FD120a-entropy-23-01584]) and (Equation 133), respectively. For example, we can divide Σ into four subsystems $\Sigma_{1},\Sigma_{2},\Sigma_{3}$, and $\Sigma_{4}$ of equal volumes and numbers of particles, but of different energies. We can assume them in their own EQ macrostate $M_{i}\left(E_{i},V/4,N/4\right)$ at temperature $T_{i}$. Then, we will obtain for $\Sigma =\Sigma_{B}$
$$\beta \left(t\right)=[\beta_{1}\left(t\right)+\beta_{2}\left(t\right)+\beta_{3}\left(t\right)+\beta_{4}\left(t\right)]/4,$$
which will require three internal variables as shown in Equation (Equation 37). It is easy to generalize the above relation to many subsystems and allowing the possibility of different sizes. We can also allow for volumes to be different for different subsystems as was done in deriving Equation (Equation 132).

The possibility to study the formation of internal structures in $\Sigma_{i}$ in a NEQ Σ should prove very useful to understand what drives their formation. A very simple example of this is the pattern formation of Rayleigh-Bérnard cells and their competition [137] in a fluid system. This pattern formation has received a lot of attention recently ([138,139] (for example)), where stable cells are studied in nonturbulent convection in steady state. It is found that each cell can be described in one of its EQ macrostate to a very good approximation with its own temperature $T_{i}$. What our approach shows is that the stable convection here can also be described by a thermodynamic constant (steady) temperature *T* associated with the steady macrostate of the entire fluid.

Having a global temperature for an inhomogeneous system does not mean that if we insert a thermometer in it anywhere, we will measure *T*. This is because the act of "inserting" a thermometer amounts to looking at the "internal" structure of the system, so we will be probing it as $\Sigma_{C}$. Thus, if we insert it in $\Sigma_{1}$, we will record $T_{1}$; and if we insert it in $\Sigma_{2}$, we will record $T_{2}$, and so on. This should not be a surprise. We refer the reader to an interesting discussion of this issue in [127].

As far as fields such as the pressure that are associated with dW are concerned, they do not pose the same kind of problem as they are purely mechanical. All one needs to do is to take their instantaneous averages over microstate probabilities for any arbitrary macrostate; see, for example, Equation ([Disp-formula FD64a-entropy-23-01584]) involving such an average. This is possible because W is a parameter, which makes $F_{wk}$ fluctuating quantities over $m_{k}$. This cannot be done for the temperature as *E* is not a parameter in the Hamiltonian. In this sense, we are considering a NEQ version of the canonical ensemble in the MNEQT, which makes $E_{k}$ fluctuating over $m_{k}$. Thus, *T* plays the role of a "parameter." For this reason, there is no way to define a temperature $T_{k}$ for $m_{k}$ and then take its average. What we can do in the MNEQT is to use the temperature of various subsystems to obtain *T* as is done in Equations ([Disp-formula FD120a-entropy-23-01584]) or (Equation 133).

We have shown that the definition of the irreversible macrowork $d_{i}W$ is always *nonnegative* as required by the second law; see Equation ([Disp-formula FD77c-entropy-23-01584]). Various consequences of the second law are discussed in Section 6.5. We have shown that, once a model for a system is given, we can identify the required number and nature of internal variables as a computational scheme in Section 3 and Section 10, and later sections in the second half of the review. These applications provide a clear strategy, once a model has been created, for computation for an arbitrary thermodynamic process and should prove useful in the field.

We have mostly alluded to $M_{\mathrm{ieq}}$’s above to highlight the importance of internal variables in $S_{Z}$, and to $M_{\mathrm{nieq}}$’s for memory-effects with respect to $M_{\mathrm{ieq}}$’s in $S_{Z}$. In the absence of a reliable model, finding $S_{Z}$ in many cases may not be easy to do. Compared to this, the identification of the state space $S_{X}$ is almost trivial based on the experimental setup. Therefore, it is much more convenient to work with $S_{X}$, with respect to which all NEQ states possess memory. Thus, *the novel approach we develop here is extremely useful as it does not require knowing the internal variables* as discussed in Section 8.3. However, for completeness, we have developed the MNEQT in $S_{Z}$, which can be easily adapted to $S_{X}$ by the procedure outlined in Remark 13.

It should be stressed, as noted in Remarks 6 and 16 that both dQ and dS exist, and so does their relation in Equation ([Disp-formula FD5a-entropy-23-01584]), regardless of the speed of the arbitrary process $P_{\mathrm{arb}}$. This makes the Clausius equality extremely important and useful as there is no restriction on its validity. It is a genuine equality even in the presence of irreversibility without any restriction on the process. This should be contrasted with the conventional form of Clausius’s inequality in Equation (Equation 94); the equality here holds only in the absence of irreversibility.

The existence of a unique *T* also appears in the microstate probabilities, see Section 8.2 that can be used to determine various fluctuations of interest. These probabilities for $M_{\mathrm{ieq}}$ also give a generalization of the EQ partition function to a NEQ partition function in Equation ([Disp-formula FD102a-entropy-23-01584]). Because of the space limitation, we did not cover its consequences.

### 15.3. Applications

We now come to the various applications of the MNEQT in the later half of the review. The main lesson here is that several applications cannot be carried out in the $\overset{{}^{\circ}}{M}$NEQT. Apart from the many applications in Section 10 that we have already discussed above, we have applied it to glasses when we derive the famous Tool-Narayanaswamy equation in Section 11. It is a phenomenological equation for which we provide a theoretical justification within the MNEQT. We study an irreversible Carnot cycle in Section 12 and derive its efficiency in terms of the entropy generation $\Delta_{i}S$ and show how it differs from the that of a reversible Carnot cycle. We also show how to compute $\Delta_{i}S$ for a simple case in which each segment is irreversible but between EQ macrostates; see Equation (Equation 138).

The next important application is about friction and the Brownian motion in Section 13. By considering the relative motion between Σ and $\tilde{\Sigma }$, we theoretically predict the well-know empirical fact that friction is caused by the relative motion. We apply the approach to a system of piston in a cylinder, a moving particle-spring system in a fluid, and just a particle fluid system.

The last application is on free expansion in Section 14. Here, we consider classical and quantum expansion. In both cases, we make a simple model of the process and show how it can used to determine $\Delta_{i}S$ between not only two $M_{\mathrm{eq}}$ macrostates but also between two $M_{\mathrm{nieq}}$; the latter cannot be determined in the $\overset{{}^{\circ}}{M}$NEQT.

### 15.4. Summary

To summarize, we have given a detailed review of the MNEQT in an extended state space that was initiated a while back [41,52,92]. Its main attraction is the variety of new applications, many of which cannot be investigated in the $\overset{{}^{\circ}}{M}$NEQT in which internal variables play no direct role. The approach is applicable to a system in any arbitrary macrostate $M_{\mathrm{arb}}$ and is used to provide a unique but very sensible definition of the temperature, which satisfies all of its important requirements. The useful aspect of the statistical approach needed for the MNEQT is that it provides a *unique* definition of generalized macroheat and macrowork dQ and dW, respectively, that are independent contributions in the generalized first law in Equation ([Disp-formula FD23a-entropy-23-01584]); both quantities are system intrinsic and obey the conventional partitioning in Equation (Equation 4) valid for any process. These macroquantities differ from the exchange macroheat and macrowork $d_{e}Q$ and $d_{e}W$, respectively. Therefore, the MNEQT directly considers the irreversible components $d_{i}Q$ and $d_{i}W$ that originate from all *internal* dissipation within the system and satisfy an important identity $d_{i}Q\equiv d_{i}W>0$, see Corollary 2, for any arbitrary irreversible process. The irreversible macroquantities vanish for a reversible process. The identification of a global and unique temperature *T* is the most significant aspect of the MNEQT in that it allows us to deal with Σ as a blackbox so that we do not need to know its interior. This requires a certain number of internal variables, which explains the extended state space. We similarly define other fields like the pressure, etc. statistically in terms of generalized “mechanical” forces; these also include generalized forces for internal variables. All these definitions are instantaneous and are not affected by how slow or fast any arbitrary process is. The latter only determines the time window of relaxations of the internal processes, and the choice of the state space. We believe that our novel approach provides a first-ever definition of the temperature, pressure, etc. and of dQ and dW for any arbitrary macrostate, whether the system is isolated on in a medium. Our approach is also valid to investigate nonequilibrium macrostates with respect to $S_{X}$, which brings memory effects in the investigation. Thus, the approach is applicable in a wide variety of situations, and fulfils Planck’s dream. 

## Figures and Tables

**Figure 1 entropy-23-01584-f001:**
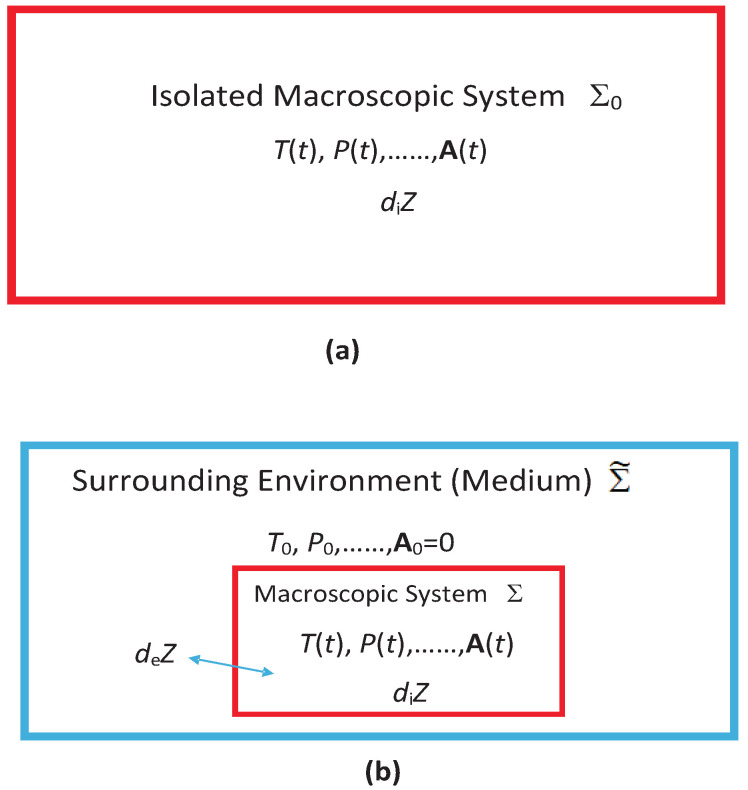
(**a**) An isolated nonequilibrium system $\Sigma_{0}$ with internally generated $d_{i}Z$ driving it towards equilibrium, during which its SI-fields T(t),P(t),⋯,A(t) continue to change to their equilibrium values; $d_{i}Z_{k}$ denote the microanalog of $d_{i}Z$. The sign of $d_{i}Z$ is determined by the second law. (**b**) A nonequilibrium systen Σ in a surrounding medium $\tilde{\Sigma }$, both forming an isolated system $\Sigma_{0}$. The macrostates of the medium and the system are characterized by their fields $T_{0},P_{0},...,A_{0}=0$ and T(t),P(t),...,A(t), respectively, which are different when the two are out of equilibrium. Exchange quantities ($d_{e}Z$) carry a suffix “e” and irreversibly generated quantities ($d_{i}Z$) within the system by a suffix "i" by extending the Prigogine notation. Their sum $d_{e}Z+d_{i}Z$ is denoted by dZ, which is a system-intrinsic quantity (see text).

**Figure 2 entropy-23-01584-f002:**
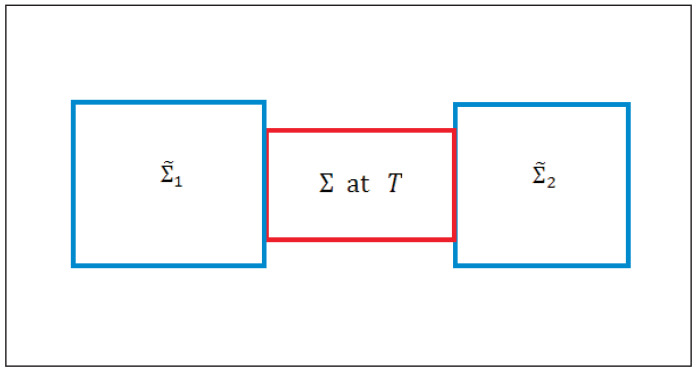
A system driven between two sources that are different in their fields; see Figure 1. If they are the same, the situation reduces to that in Figure 1a. Later in Section 10.3 we consider this situation between two heat sources ${\tilde{\Sigma }}_{h1}$ and ${\tilde{\Sigma }}_{h2}$, where we treat Σ as a composite system as an application of our approach.

**Figure 3 entropy-23-01584-f003:**
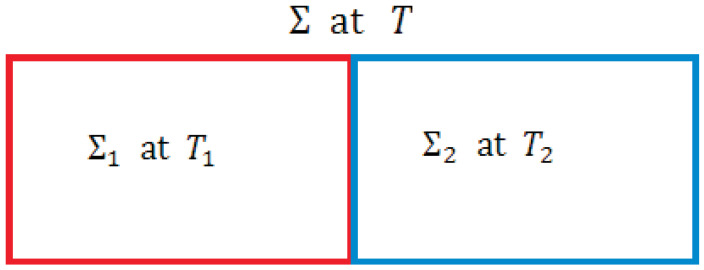
A composite system Σ consisting of two identical subsystems $\Sigma_{1}$ at temperature $T_{1}$ and $\Sigma_{2}$ at temperature $T_{2}$. It will be seen later in Section 10.3 that the thermodynamic temperature of Σ can be defined as *T* given by Equation ([Disp-formula FD120a-entropy-23-01584]). The irreversibility in Σ requires one internal variable ξ given in Equation (Equation 36).

**Figure 4 entropy-23-01584-f004:**
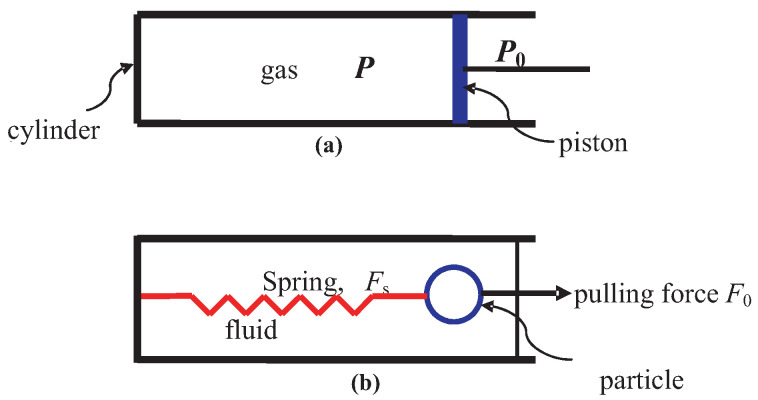
We schematically show a system of (**a**) gas in a cylinder with a movable piston under an external pressure $P_{0\;}$controlling the volume *V* of the gas, and (**b**) a particle attached to a spring in a fluid being pulled by an external force $F_{0}$, which causes the spring to stretch or compress depending on its direction. In an irreversible process, the internal pressure *P* (the spring force $F_{s}$) is different in magnitude from the external pressure $P_{0}$ (external force $F_{0}$).

**Figure 5 entropy-23-01584-f005:**
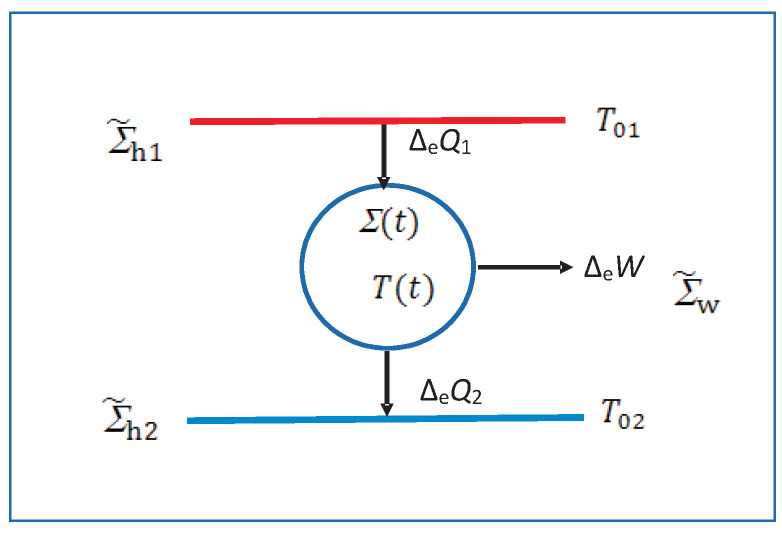
An irreversible Carnot cycle running between two heat reservoirs ${\tilde{\Sigma }}_{h1}$ and ${\tilde{\Sigma }}_{h2}$.

**Figure 6 entropy-23-01584-f006:**
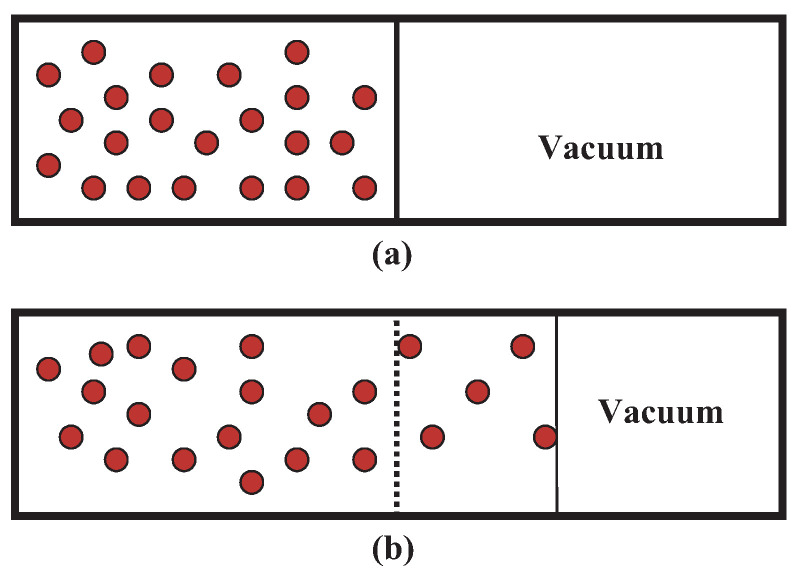
Free expansion of a gas. The gas is confined to the left chamber, which is separated by a hard partition (shown by a solid black vertical line) from the vacuum in the right chamber as shown in (**a**). At time t=0, the partition is removed abruptly as shown by the broken line in its original place in (**b**). The gas expands in the empty space, devoid of matter and radiation, on the right but the expansion is gradual as shown by the solid front, which separates it from the vacuum on its right. We can also think of the hard partition in (**a**) as a piston, which maintains the volume of the gas on its left. The piston can be moved slowly or rapidly to the right within the right chamber with a pressure $P_{0}<P$ to change this volume. The free expansion occurs when the piston moves extremely (infinitely) fast by letting $P_{0}\to 0$.

## Data Availability

Not applicable.

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
