# Peer review of "A Review of the System-Intrinsic Nonequilibrium Thermodynamics in Extended Space (MNEQT) with Applications"

_entropy, 2021, doi:10.3390/e23121584_

Round 1
Reviewer 1 Report
This is a well written exhaustive review paper on some of the active topics of interest in non-equilibrium thermodynamics. I don't particularly have any criticisms on the contents of the paper but I would like to point out that the exploratory examples that the author uses are fairly simple (but of course in a complex setting), such as, the 1D gas, piston-cylinder systems, free expansions etc. I would be more interested in systems that are more complex; like, how dissipation, thermodynamic forces and internal gradients in the system give rise to complex structures. Especially, in the conclusion section of the paper (pg. 34), the author discusses about inhomogeneous systems with different (local) temperatures. There are some interesting results that have been shown in this specific context, see for example Sci Rep 9 (1), 1-11, 2019; Physica A 547 123867, 2020; arXiv:2107.03678, 2021 along with others such as, Entropy 22 (8), 800 2020; PRL 61 (8) (1988) 947; MNRAS 88 493, 1928; and arXiv:2010.14241, 2020 which I think the author should discuss in this review as well. With these considerations the paper, in my opinion should be accepted.
Reviewer 2 Report
In this work the author presents an extensive development of the fundamentals behind an internal-variable description of non-equilibrium thermodynamics, based on his previous work. He shows how through this abstract development a consistent to Clausius inequality definition for the nonequilibrium thermodynamic temperature can be provided. The foundation of the theory is based on the analysis of reversible and irreversible heat and work changes in a nonequilibrium system. The connection to the microstates giving rise to the macrostates is also investigated. Some examples are also provided.
As the foundation of nonequilibrium thermodynamics and in particular of proper nonequilibrium extensions for the temperature and entropy remain still unexplained, a work that tries t address those issues and at a fundamental generic level is in principle welcome. However, as several investigations are taking place, it is important for any one to at least try to present a connection through similarities or draw a difference and thus explain better its particular advantage and benefit. Regarding the present work, there are many of these issues that the author leaves untouched. Especially two of them are of high importance (as outlined below in the detailed comments) to require his careful consideration in a major revision, before his work can be recommended for publication. In addition, some minor errors are pointed out in the editorial comments.
Detailed comments
- The concept of temperature is considered here exclusively based on a macroscopic point of view through Clausius inequality and Gibbs-type relations. However, it is also well known that temperature enters in the evaluation of fluctuations, as well as though the thermal energy. All of these concepts can be used to extend the definition of the temperature under nonequilibrium conditions---see, the recent work [1]. Although it seems unlikely that any single definition of the nonequilibrium temperature will allow to preserve all the equilibrium-based relations involving the equilibrium temperature, there is interest to know why one approach should be preferred over another and how those different extensions are related to each other. The author therefore has to address those issues much more extensively in a revised version of his work.
- The concept of nonequilibrium temperature is connected closely to that of nonequilibrium entropy. Still, not much about the latter quantity is mentioned here. In particular, it has been proposed (see, for example, reference [19] provided in this manuscript) that the gradient of various state variables may play a role in the definitions of nonequilibrium thermodynamic quantities. Next to that proposal is that of involving nonlocal effects altogether. How can then one possibly define microstates and suitable internal variables? The author is asked to provide some examples where those effects have been mentioned and show how they can be effectively dealt within his theory.
- Editorial remarks
- a) line 17: Use “examples” instead of “example”
- b) Heading after line 882: Use “derivation” instead of “derive”
- c) Reference 17: Boltzmann’s name is misspelled.
Reference
[1] U. Lucia and G. Grisolia, Nonequilibrium temperature: An approach from irreversibility, Materials, 14 (2021) 2004.
Reviewer 3 Report
The paper is interesting. It is well written and well organized.
I have only a doubt on the equation (85a). If we consider an engine which exchange heat with the environment the first term delta_eQ (T_0 - T)/T_0 is related to the efficiency: - eta/(1-eta) which is always negative. It means that the work results delta_iW > eta/(1-eta) delta_eQ. Is this a lower value of work that can be obtained? It means that there is an upper and lower limit of work obtainable in any process?
I suggest to improve only this consideration.
Round 2
Reviewer 2 Report
I have found that in this revised version the author has satisfactorily responded to my previous comments raised upon review of the original manuscript and that he made adequate changes in the revised manuscript. As a result, I have no further objections and I recommend its publication without any further changes needed.
Author Response
I am pleased that the referee is satisfied now.